# Federated Natural Policy Gradient and Actor Critic Methods for Multi-task Reinforcement Learning

**Tong Yang**[*]  **Shicong Cen**[†]  **Yuting Wei**[‡]  **Yuxin Chen**[§]  **Yuejie Chi**[¶]
CMU  CMU  UPenn  UPenn  CMU

## Abstract

Federated reinforcement learning (RL) enables collaborative decision making of multiple distributed agents without sharing local data trajectories. In this work, we consider a multi-task setting, in which each agent has its own private reward function corresponding to different tasks, while sharing the same transition kernel of the environment. Focusing on infinite-horizon Markov decision processes, the goal is to learn a globally optimal policy that maximizes the sum of the discounted total rewards of all the agents in a decentralized manner, where each agent only communicates with its neighbors over some prescribed graph topology. We develop federated vanilla and entropy-regularized natural policy gradient (NPG) methods in the tabular setting under softmax parameterization, where gradient tracking is applied to estimate the global Q-function to mitigate the impact of imperfect information sharing. We establish non-asymptotic global convergence guarantees under exact policy evaluation, where the rates are nearly independent of the size of the state-action space and illuminate the impacts of network size and connectivity, and further establish its robustness against inexact policy evaluation. We further propose a federated natural actor critic (NAC) method for multi-task RL with function approximation and stochastic policy evaluation, and establish its finite-time sample complexity taking the errors of function approximation into account. To the best of our knowledge, this is the first time that near dimension-free global convergence is established for federated multi-task RL using policy optimization.

## 1 Introduction

Federated reinforcement learning (FRL) is an emerging paradigm that combines the advantages of federated learning (FL) and reinforcement learning (RL) [QZLZ21, ZFL+19], allowing multiple agents to learn a shared policy from local experiences, without exposing their private data to a central server nor other agents. FRL is poised to enable collaborative and efficient decision making in scenarios where data is distributed, heterogeneous, and sensitive, which arise frequently in applications such as edge computing, smart cities, and healthcare [WHM+23, WKNL20, ZFL+19], to name just a few. As has been observed [LZZ+17], decentralized training can lead to performance improvements in FL by avoiding communication congestions at busy nodes such as the server, especially under high-latency scenarios. This motivates us to design algorithms for the *fully decentralized* setting, a

---

[*]Department of Electrical and Computer Engineering, Carnegie Mellon University; email: `tongyang@andrew.cmu.edu`.

[†]Department of Electrical and Computer Engineering, Carnegie Mellon University; email: `shicongc@andrew.cmu.edu`.

[‡]Department of Statistics and Data Science, Wharton School, University of Pennsylvania; email: `ytwei@wharton.upenn.edu`.

[§]Department of Statistics and Data Science, Wharton School, University of Pennsylvania; email: `yuxinc@wharton.upenn.edu`.

[¶]Department of Electrical and Computer Engineering, Carnegie Mellon University; email: `yuejiechi@cmu.edu`.

38th Conference on Neural Information Processing Systems (NeurIPS 2024).

scenario where the agents can only communicate with their local neighbors over a prescribed network topology.[6]

In this work, we study the problem of *federated multi-task RL* [AR21, QZLZ21, YLS+20], where each agent collects its own reward — possibly unknown to other agents — corresponding to the local task at hand, while having access to the same dynamics (i.e., transition kernel) of the environment. The collective goal is to learn a shared policy that maximizes the total rewards accumulated from all the agents; in other words, one seeks a policy that performs well in terms of overall benefits, rather than biasing towards any individual task, achieving the Pareto frontier in a multi-objective context. There is no shortage of application scenarios where federated multi-task RL becomes highly relevant. For instance, in healthcare [ZBW+20], different hospitals may be interested in finding an optimal treatment for all patients without disclosing private data, where the effectiveness of the treatment can vary across different hospitals due to demographic differences. See Appendix B.1 for more application scenarios of our setting.

Nonetheless, despite the promise, provably efficient algorithms for federated multi-task RL remain substantially under-explored, especially in the fully decentralized setting. The heterogeneity of local tasks leads to a higher degree of disagreements between the global value function and local value functions of individual agents. Due to the lack of global information sharing, care needs to be taken to judiciously balance the use of neighboring information (to facilitate consensus) and local data (to facilitate learning) when updating the policy. To the best of our knowledge, very few algorithms are currently available to find the global optimal policy with non-asymptotic convergence guarantees even for tabular infinite-horizon Markov decision processes.

Motivated by the connection with decentralized optimization, it is tempting to take a policy optimization perspective to tackle this challenge. Policy gradient (PG) methods, which seek to learn the policy of interest via first-order optimization methods, play an eminent role in RL due to their simplicity and scalability. In particular, natural policy gradient (NPG) methods [Ama98, Kak01] are among the most popular variants of PG methods, underpinning default methods used in practice such as trust region policy optimization (TRPO) [SLA+15] and proximal policy optimization (PPO) [SWD+17]. On the theoretical side, it has also been established recently that the NPG method enjoys fast global convergence to the optimal policy in an almost dimension-free manner [AKLM21, CWC21], where the iteration complexity is nearly independent of the size of the state-action space. These benefits can be translated to their sample-based counterparts such as the natural actor critic (NAC) method [BSGL09, XWL20, KDRM22], where the policies are evaluated via stochastic samples. It is natural to ask:

> Can we develop **federated** NPG and NAC methods with **non-asymptotic global convergence** guarantees for multi-task RL in the fully decentralized setting?

## 1.1 Our contributions

Focusing on infinite-horizon Markov decision processes (MDPs), we provide an affirmative answer to the above question, by developing federated NPG (FedNPG) methods for solving both the vanilla and entropy-regularized multi-task RL problems with finite-time global convergence guarantees. While entropy regularization is often incorporated as an effective strategy to encourage exploration during policy learning, solving the entropy-regularized RL problem is of interest in its own right, as the optimal regularized policy possesses desirable robust properties with respect to reward perturbations [EL21, MP95]. Due to the multiplicative update nature of NPG methods under softmax parameterization, it is more convenient to work with the logarithms of local policies in the decentralized setting. In each iteration of the proposed FedNPG method, the logarithms of local policies are updated by a weighted linear combination of two terms (up to normalization): a gossip mixing [NO09] of the logarithms of neighboring local policies, and a local estimate of the global Q-function tracked via the technique of dynamic average consensus [ZM10], a prevalent idea in decentralized optimization that allows for the use of large constant learning rates [DLS16, NOS17, QL17] to accelerate convergence. We further develop sample-efficient federated NAC (FedNAC) methods that allow for both stochastic policy evaluation and function approximation. Our contributions are as follows.

- We propose FedNPG methods for both the vanilla and entropy-regularized multi-task RL problems, where each agent only communicates with its neighbors and performs local computation using its own reward or task information.

---

[6]Our work seamlessly handles the server-client setting as a special case, by assuming the network topology as a fully connected network.

| setting | algorithms | iteration complexity | optimality criteria |
|---------|-----------|---------------------|---------------------|
| unregularized | NPG [AKLM21] | $\mathcal{O}\left(\frac{1}{(1-\gamma)^2\varepsilon} + \frac{\log|\mathcal{A}|}{\eta\varepsilon}\right)$ | $V^\star - V^{\pi^{(t)}} \leq \varepsilon$ |
| | FedNPG (ours) | $\mathcal{O}\left(\frac{\sigma\sqrt{N}\log|\mathcal{A}|}{(1-\gamma)^{\frac{9}{2}}(1-\sigma)\varepsilon^{\frac{3}{2}}} + \frac{1}{(1-\gamma)^2\varepsilon}\right)$ | $\frac{1}{T}\sum_{t=0}^{T-1}\left(V^\star - V^{\bar{\pi}^{(t)}}\right) \leq \varepsilon$ |
| regularized | NPG [CWC21] | $\mathcal{O}\left(\frac{1}{\tau\eta}\log\left(\frac{1}{\varepsilon}\right)\right)$ | $V_\tau^\star - V_\tau^{\pi^{(t)}} \leq \varepsilon$ |
| | FedNPG (ours) | $\mathcal{O}\left(\max\left\{\frac{1}{\tau\eta}, \frac{1}{1-\sigma}\right\}\log\left(\frac{1}{\varepsilon}\right)\right)$ | $V_\tau^\star - V_\tau^{\bar{\pi}^{(t)}} \leq \varepsilon$ |

Table 1: Iteration complexities of NPG and FedNPG (ours) methods to reach $\varepsilon$-accuracy of the vanilla and entropy-regularized problems, where we assume exact gradient evaluation, and only keep the dominant terms w.r.t. $\varepsilon$. The policy estimates in the $t$-iteration are $\pi^{(t)}$ and $\bar{\pi}^{(t)}$ for NPG and FedNPG, respectively, where $T$ is the number of iterations. Here, $N$ is the number of agents, $\tau \leq 1$ is the regularization parameter, $\sigma \in [0, 1]$ is the spectral radius of the network, $\gamma \in [0, 1)$ is the discount factor, $|\mathcal{A}|$ is the size of the action space, and $\eta > 0$ is the learning rate. The iteration complexities of FedNPG reduce to their centralized counterparts when $\sigma = 0$. For vanilla FedNPG, the learning rate is set as $\eta = \eta_1 = \mathcal{O}\left(\frac{(1-\gamma)^9(1-\sigma)^2\log|\mathcal{A}|}{TN\sigma}\right)^{1/3}$; for entropy-regularized FedNPG, the learning rate satisfies $0 < \eta < \eta_0 = \mathcal{O}\left(\frac{(1-\gamma)^7(1-\sigma)^2\tau}{\sigma N}\right)$.

- Assuming access to exact policy evaluation, we establish that the average iterate of vanilla FedNPG converges globally at a rate of $\mathcal{O}(1/T^{2/3})$ in terms of the sub-optimality gap for the multi-task RL problem, and that the last iterate of entropy-regularized FedNPG converges globally at a linear rate to the regularized optimal policy. Our convergence theory highlights the impacts of all salient problem parameters (see Table 1 for details), such as the size and connectivity of the communication network. In particular, the iteration complexities of FedNPG are again almost independent of the size of the state-action space, which recover prior results on the centralized NPG methods when the network is fully connected.
- We further demonstrate the stability of the proposed FedNPG methods when policy evaluations are only available in an inexact manner. To be specific, we prove that their convergence rates remain unchanged as long as the approximation errors are sufficiently small in the $\ell_\infty$ sense.
- We go beyond the tabular setting and black-box policy evaluation by proposing FedNAC— a federated actor critic method for multi-task RL with function approximation and stochastic policy evaluation — and establish a finite-sample sample complexity on the order of $\mathcal{O}(1/\varepsilon^{7/2})$ for each agent in terms of the expected sub-optimality gap for the fully decentralized setting.

To the best of our knowledge, the proposed federated NPG and NAC methods are the first policy optimization methods for multi-task RL that achieve near dimension-free global convergence guarantees in terms of iteration and sample complexities, allowing for fully decentralized communication without any need to share local reward/task information. We conduct numerical experiments in a multi-task GridWorld environment to corroborate the efficacy of the proposed methods (see Appendix H). We defer the readers to Appendix A for more related work, and Appendix B.2 for additional discussions on our theoretical contributions.

**Notation.** Boldface small and capital letters denote vectors and matrices, respectively. Sets are denoted with curly capital letters, e.g., $\mathcal{S}, \mathcal{A}$. We let $(\mathbb{R}^d, \|\cdot\|)$ denote the $d$-dimensional real coordinate space equipped with norm $\|\cdot\|$. The $\ell_p$-norm of $\boldsymbol{v}$ is denoted by $\|\boldsymbol{v}\|_p$, where $1 \leq p \leq \infty$, and the spectral norm and the Frobenius norm of a matrix $\boldsymbol{M}$ are denoted by $\|\boldsymbol{M}\|_2$ and $\|\boldsymbol{M}\|_F$, resp. We let $[N]$ denote $\{1, \ldots, N\}$, use $\mathbf{1}_N$ to represent the all-one vector of length $N$, and denote by $\mathbf{0}$ a vector or a matrix consisting of all 0's. We allow the application of functions such as $\log(\cdot)$ and $\exp(\cdot)$ to vectors or matrices, with the understanding that they are applied in an element-wise manner.

## 2 Model and backgrounds

**Markov decision processes.** We consider an infinite-horizon discounted Markov decision process (MDP) denoted by $\mathcal{M} = (\mathcal{S}, \mathcal{A}, P, r, \gamma)$, where $\mathcal{S}$ and $\mathcal{A}$ denote the state space and the action space, respectively, $\gamma \in [0, 1)$ indicates the discount factor, $P : \mathcal{S} \times \mathcal{A} \to \Delta(\mathcal{S})$ is the transition kernel, and $r : \mathcal{S} \times \mathcal{A} \to [0, 1]$ stands for the reward function. To be more specific, for each state-action pair $(s, a) \in \mathcal{S} \times \mathcal{A}$ and any state $s' \in \mathcal{S}$, we denote by $P(s'|s, a)$ the transition probability from state

$s$ to state $s'$ when action $a$ is taken, and $r(s, a)$ the instantaneous reward received in state $s$ when action $a$ is taken. Furthermore, a policy $\pi : \mathcal{S} \to \Delta(\mathcal{A})$ specifies an action selection rule, where $\pi(a|s)$ specifies the probability of taking action $a$ in state $s$ for each $(s, a) \in \mathcal{S} \times \mathcal{A}$.

For any given policy $\pi$, we denote by $V^\pi : \mathcal{S} \mapsto \mathbb{R}$ the corresponding value function, which is the expected discounted cumulative reward with an initial state $s_0 = s$, given by

$$\forall s \in \mathcal{S} : \quad V^\pi(s) := \mathbb{E}\left[\sum_{t=0}^\infty \gamma^t r(s_t, a_t)|s_0 = s\right], \tag{1}$$

where the randomness is over the trajectory generated following the policy $a_t \sim \pi(\cdot|s_t)$ and the MDP dynamic $s_{t+1} \sim P(\cdot|s_t, a_t)$. We also overload the notation $V^\pi(\rho)$ to indicate the expected value function of policy $\pi$ when the initial state follows a distribution $\rho$ over $\mathcal{S}$, namely, $V^\pi(\rho) := \mathbb{E}_{s \sim \rho}[V^\pi(s)]$. Similarly, the Q-function $Q^\pi : \mathcal{S} \times \mathcal{A} \mapsto \mathbb{R}$ of policy $\pi$ is defined by

$$Q^\pi(s, a) := \mathbb{E}\left[\sum_{t=0}^\infty \gamma^t r(s_t, a_t)|s_0 = s, a_0 = a\right] \tag{2}$$

for all $(s, a) \in \mathcal{S} \times \mathcal{A}$, which measures the expected discounted cumulative reward with an initial state $s_0 = s$ and an initial action $a_0 = a$, with expectation taken over the randomness of the trajectory. The optimal policy $\pi^\star$ refers to the policy that maximizes the value function $V^\pi(s)$ for all states $s \in \mathcal{S}$, which is guaranteed to exist [Put14]. The corresponding optimal value function and Q-function are denoted as $V^\star$ and $Q^\star$, respectively.

**Entropy-regularized RL.** Entropy regularization [WP91, ALRNS19] is a popular technique in practice that encourages stochasticity of the policy to promote exploration, as well as robustness against reward uncertainties. Mathematically, this can be viewed as adjusting the instantaneous reward based the current policy in use as

$$\forall(s, a) \in \mathcal{S} \times \mathcal{A} : \quad r_\tau(s, a) := r(s, a) - \tau \log \pi(a|s), \tag{3}$$

where $\tau \geq 0$ denotes the regularization parameter. Typically, $\tau$ should not be too large to outweigh the actual rewards; for ease of presentation, we assume $\tau \leq \min\left\{1, \frac{1}{\log|\mathcal{A}|}\right\}$ [CCDX22]. Equivalently, this amounts to the entropy-regularized (also known as "soft") value function, defined as

$$\forall s \in \mathcal{S} : \quad V_\tau^\pi(s) := V^\pi(s) + \tau \mathcal{H}(s, \pi), \tag{4}$$

where

$$\mathcal{H}(s, \pi) := \mathbb{E}\left[\sum_{t=0}^\infty -\gamma^t \log \pi(a_t|s_t)|s_0 = s\right]. \tag{5}$$

Analogously, for all $(s, a) \in \mathcal{S} \times \mathcal{A}$, the regularized (or soft) Q-function $Q_\tau^\pi$ of policy $\pi$ is related to the soft value function $V_\tau^\pi(s)$ as

$$Q_\tau^\pi(s, a) = r(s, a) + \gamma \mathbb{E}_{s' \in P(\cdot|s,a)}\left[V_\tau^\pi(s')\right], \tag{6a}$$
$$V_\tau^\pi(s) = \mathbb{E}_{a \sim \pi(\cdot|s)}\left[-\tau\pi(a|s) + Q_\tau^\pi(s, a)\right]. \tag{6b}$$

The optimal regularized policy, the optimal regularized value function, and the Q-function are denoted by $\pi_\tau^\star$, $V_\tau^\star$, and $Q_\tau^\star$, respectively.

**Natural policy gradient methods.** Natural policy gradient (NPG) methods lie at the heart of policy optimization, serving as the backbone of popular heuristics such as TRPO [SLA$^+$15] and PPO [SWD$^+$17]. Instead of directly optimizing the policy over the probability simplex, one often adopts the softmax parameterization, which parameterizes the policy as $\pi_\theta := \mathrm{softmax}(\theta)$ or

$$\pi_\theta(a|s) := \frac{\exp \theta(s, a)}{\sum_{a' \in \mathcal{A}} \exp \theta(s, a')} \tag{7}$$

for any $\theta : \mathcal{S} \times \mathcal{A} \to \mathbb{R}$ and $(s, a) \in \mathcal{S} \times \mathcal{A}$.

In the tabular setting, the update rule of vanilla NPG at the $t$-th iteration can be concisely represented as

$$\pi^{(t+1)}(a|s) \propto \pi^{(t)}(a|s) \exp\left(\frac{\eta Q^{(t)}(s, a)}{1 - \gamma}\right), \tag{8}$$

Turning to the regularized problem, we note that the update rule of entropy-regularized NPG becomes

$$\pi^{(t+1)}(a|s) \propto (\pi^{(t)}(a|s))^{1-\frac{\eta\tau}{1-\gamma}} \exp\left(\frac{\eta Q_\tau^{(t)}(s,a)}{1-\gamma}\right), \tag{9}$$

where $\eta \in (0, \frac{1-\gamma}{\tau}]$ is the learning rate, and $Q_\tau^{(t)} = Q_\tau^{\pi^{(t)}}$ is the soft Q-function of policy $\pi^{(t)}$.

## 3  Federated NPG methods for multi-task RL

In this paper, we consider the federated multi-task RL setting, where a set of agents learn collaboratively a single policy that maximizes its average performance over all the tasks using only local computation and communication.

**Multi-task RL.** Each agent $n \in [N]$ has its own private reward function $r_n(s,a)$ — corresponding to different tasks — while sharing the same transition kernel of the environment. The goal is to collectively learn a single policy $\pi$ that maximizes the global value function given by $V^\pi(s) = \frac{1}{N}\sum_{n=1}^N V_n^\pi(s)$, where $V_n^\pi$ is the value function of agent $n \in [N]$, defined by

$$V_n^\pi(s) := \mathbb{E}\left[\sum_{t=0}^\infty \gamma^t r_n(s_t, a_t)|s_0 = s\right].$$

Clearly, the global value function corresponds to using the average reward of all agents $r(s,a) = \frac{1}{N}\sum_{n=1}^N r_n(s,a)$. The global Q-function $Q^\pi(s,a)$ and the agent Q-functions $Q_n^\pi(s,a)$ can be defined in a similar manner obeying $Q^\pi(s,a) = \frac{1}{N}\sum_{n=1}^N Q_n^\pi(s,a)$.

In parallel, we are interested in the entropy-regularized setting, where each agent $n \in [N]$ is equipped with a regularized reward function given by $r_{\tau,n}(s,a) := r_n(s,a) - \tau\log\pi(a|s)$. And we define similarly the regularized value functions as

$$V_{\tau,n}^\pi(s) := \mathbb{E}\left[\sum_{t=0}^\infty \gamma^t r_{\tau,n}(s_t, a_t)|s_0 = s\right]$$

for all $n \in [N]$ and $V_\tau^\pi(s) = \frac{1}{N}\sum_{n=1}^N V_{\tau,n}^\pi(s)$, $\forall s \in \mathcal{S}$. The soft Q-function of agent $n$ is given by

$$Q_{\tau,n}^\pi(s,a) = r_n(s,a) + \gamma\mathbb{E}_{s'\in P(\cdot|s,a)}\left[V_{\tau,n}^\pi(s')\right], \tag{10}$$

and the global soft Q-function is given by $Q_\tau^\pi(s,a) = \frac{1}{N}\sum_{n=1}^N Q_{\tau,n}^\pi(s,a)$.

**Federated policy optimization in the fully decentralized setting.** We consider a federated setting with fully decentralized communication, that is, all the agents are synchronized to perform information exchange over some prescribed network topology denoted by an undirected weighted graph $\mathcal{G}([N], E)$. Here, $E$ stands for the edge set of the graph with $N$ nodes — each corresponding to an agent — and two agents can communicate with each other if and only if there is an edge connecting them. The information sharing over the graph is best described by a mixing matrix [NO09], denoted by $W = [w_{ij}] \in [0,1]^{N\times N}$, where $w_{ij}$ is a positive number if $(i,j) \in E$ and 0 otherwise. We also make the following standard assumptions on the mixing matrix.

**Assumption 3.1** (double stochasticity). The mixing matrix $W = [w_{ij}] \in [0,1]^{N\times N}$ is symmetric (i.e., $W^\top = W$) and doubly stochastic (i.e., $W\mathbf{1}_N = \mathbf{1}_N$, $\mathbf{1}_N^\top W = \mathbf{1}_N^\top$).

The following standard metric measures how fast information propagates over the graph.

**Definition 3.2** (spectral radius). The spectral radius of $W$ is given as $\sigma := \|W - \frac{1}{N}\mathbf{1}_N\mathbf{1}_N^\top\|_2 \in [0,1)$.

The spectral radius $\sigma$ determines how fast information propagate over the network. For instance, in a fully-connected network, we can achieve $\sigma = 0$ by setting $W = \frac{1}{N}\mathbf{1}_N\mathbf{1}_N^\top$. For control of $1/(1-\sigma)$ regarding different graphs, we refer the readers to [NOR18]. In an Erdös-Rényi random graph, as long as the graph is connected, one has with high probability $\sigma \asymp 1$. Another immediate consequence is that for any $x \in \mathbb{R}^N$, letting $\bar{x} = \frac{1}{N}\mathbf{1}_N^\top x$ be its average, we have

$$\|Wx - \bar{x}\mathbf{1}_N\|_2 \le \sigma\|x - \bar{x}\mathbf{1}_N\|_2, \tag{11}$$

where the consensus error contracts by a factor of $\sigma$.

---

**Algorithm 1** Federated NPG (FedNPG)

---

1: **Input:** learning rate $\eta > 0$, iteration number $T \in \mathbb{N}_+$, mixing matrix $\boldsymbol{W} \in \mathbb{R}^{N \times N}$.
2: **Initialize:** $\boldsymbol{\pi}^{(0)}$, $\boldsymbol{T}^{(0)} = \boldsymbol{Q}^{(0)}$.
3: **for** $t = 0, 1, \cdots T - 1$ **do**
4:     Update the policy for each $(s, a) \in \mathcal{S} \times \mathcal{A}$:

$$\log \boldsymbol{\pi}^{(t+1)}(a|s) = \boldsymbol{W}\left(\log \boldsymbol{\pi}^{(t)}(a|s) + \frac{\eta}{1-\gamma}\boldsymbol{T}^{(t)}(s, a)\right) - \log \boldsymbol{z}^{(t)}(s), \qquad (15)$$

    where $\boldsymbol{z}^{(t)}(s) = \sum_{a' \in \mathcal{A}} \exp\left\{\boldsymbol{W}\left(\log \boldsymbol{\pi}^{(t)}(a'|s) + \frac{\eta}{1-\gamma}\boldsymbol{T}^{(t)}(s, a')\right)\right\}$.
5:     Evaluate $\boldsymbol{Q}^{(t+1)}$.
6:     Update the global Q-function estimate for each $(s, a) \in \mathcal{S} \times \mathcal{A}$:

$$\boldsymbol{T}^{(t+1)}(s, a) = \boldsymbol{W}\Big(\boldsymbol{T}^{(t)}(s, a) + \underbrace{\boldsymbol{Q}^{(t+1)}(s, a) - \boldsymbol{Q}^{(t)}(s, a)}_{\text{Q-tracking}}\Big). \qquad (16)$$

7: **end for**

---

### 3.1 Proposed federated NPG algorithms

Assuming softmax parameterization, the problem can be formulated as decentralized optimization,

$$(\text{unregularized}) \quad \max_\theta V^{\pi_\theta}(s) = \frac{1}{N}\sum_{n=1}^{N} V_n^{\pi_\theta}(s), \qquad (12)$$

$$(\text{regularized}) \quad \max_\theta V_\tau^{\pi_\theta}(s) = \frac{1}{N}\sum_{n=1}^{N} V_{\tau,n}^{\pi_\theta}(s), \qquad (13)$$

where $\pi_\theta \coloneqq \mathrm{softmax}(\theta)$ subject to communication constraints. Motivated by the success of NPG methods, we aim to develop federated NPG methods to achieve our goal. For notational convenience, let $\boldsymbol{\pi}^{(t)} \coloneqq \left(\pi_1^{(t)}, \cdots, \pi_N^{(t)}\right)^\top$ be the collection of policy estimates at all agents in the $t$-th iteration. Let

$$\overline{\pi}^{(t)} \coloneqq \mathrm{softmax}\left(\frac{1}{N}\sum_{n=1}^{N}\log \pi_n^{(t)}\right), \qquad (14)$$

which satisfies that $\overline{\pi}^{(t)}(a|s) \propto \left(\prod_{n=1}^{N} \pi_n^{(t)}(a|s)\right)^{1/N}$ for each $(s, a) \in \mathcal{S} \times \mathcal{A}$. Therefore, $\overline{\pi}^{(t)}$ could be seen as the normalized geometric mean of $\{\pi_n^{(t)}\}_{n \in [N]}$. Define the collection of Q-function estimates as $\boldsymbol{Q}^{(t)} \coloneqq \left(Q_1^{\pi_1^{(t)}}, \cdots, Q_N^{\pi_N^{(t)}}\right)^\top$ and $\boldsymbol{Q}_\tau^{(t)} \coloneqq \left(Q_{\tau,1}^{\pi_1^{(t)}}, \cdots, Q_{\tau,N}^{\pi_N^{(t)}}\right)^\top$. We shall often abuse the notation and treat $\boldsymbol{\pi}^{(t)}, \boldsymbol{Q}_\tau^{(t)}$ as matrices in $\mathbb{R}^{N \times |\mathcal{S}||\mathcal{A}|}$, and treat $\boldsymbol{\pi}^{(t)}(a|s), \boldsymbol{Q}_\tau^{(t)}(a|s)$ as vectors in $\mathbb{R}^N$, for all $(s, a) \in \mathcal{S} \times \mathcal{A}$.

**Vanilla federated NPG methods.** To motivate the algorithm development, observe that the NPG method (cf. (8)) applied to (12) adopts the update rule $\pi^{(t+1)}(a|s) \propto \pi^{(t)}(a|s)\exp\left(\frac{\eta \sum_{n=1}^{N} Q_n^{\pi^{(t)}}(s,a)}{N(1-\gamma)}\right)$ for all $(s, a) \in \mathcal{S} \times \mathcal{A}$. Two challenges arise when executing this update rule: the policy estimates are maintained locally without consensus, and the global Q-function are unavailable in the decentralized setting. To address these challenges, we apply the idea of dynamic average consensus [ZM10], where each agent maintains its own estimate $T_n^{(t)}(s, a)$ of the global Q-function, which are collected as vector $\boldsymbol{T}^{(t)} = \left(T_1^{(t)}, \cdots, T_N^{(t)}\right)^\top$. At each iteration, each agent updates its policy estimates based on its neighbors' information via gossip mixing, in addition to a correction term that tracks the difference $Q_n^{\pi_n^{(t+1)}}(s, a) - Q_n^{\pi_n^{(t)}}(s, a)$ of the local Q-functions between consecutive policy updates. Note that the mixing is applied linearly to the logarithms of local policies, which translates into a multiplicative mixing of the local policies. Algorithm 1 summarizes the detailed procedure of the proposed algorithm written in a compact matrix form, which we dub as federated NPG (FedNPG). Note that the agents do not need to share their reward functions with

others, and agent $n \in [N]$ will only be responsible to evaluate the local policy $\pi_n^{(t)}$ using the local reward $r_n$.

**Entropy-regularized federated NPG methods.** Moving onto the entropy regularized case, we adopt similar algorithmic ideas to decentralize (9), and propose the federated NPG (FedNPG) method with entropy regularization, summarized in Algorithm 2 (see Appendix C.1). Clearly, the entropy-regularized FedNPG method reduces to vanilla FedNPG in the absence of the regularization (i.e., when $\tau = 0$).

### 3.2 Theoretical guarantees

**Global convergence of FedNPG with exact policy evaluation.** We begin with the global convergence of FedNPG (cf. Algorithm 1), stated in the following theorem. The formal statement and proof can be found in Appendix D.3, and see Appendix B.2 for discussions on the technical challenges.

**Theorem 3.3** (Global sublinear convergence of exact FedNPG (informal)). *Suppose $\pi_n^{(0)}, n \in [N]$ are set as the uniform distribution. Then when $T \geq \frac{128\sqrt{N}\log|\mathcal{A}|\sigma^4}{(1-\sigma)^4}$ and $\eta = \left(\frac{(1-\gamma)^9(1-\sigma)^2\log|\mathcal{A}|}{32TN\sigma^2}\right)^{1/3}$, we have*

$$\frac{1}{T}\sum_{t=0}^{T-1}\left(V^\star(\rho) - V^{\overline{\pi}^{(t)}}(\rho)\right) \lesssim \frac{V^\star(d_\rho^{\pi^\star})}{(1-\gamma)T} + \frac{N^{1/3}\sigma^{2/3}}{(1-\gamma)^3(1-\sigma)^{2/3}}\left(\frac{\log|\mathcal{A}|}{T}\right)^{2/3}. \tag{17a}$$

$$\left\|\log\pi_n^{(t)} - \log\overline{\pi}^{(t)}\right\|_\infty \lesssim \frac{N^{2/3}\sigma^{1/3}}{(1-\gamma)(1-\sigma)^{1/3}}\left(\frac{\log|\mathcal{A}|}{T}\right)^{1/3}. \tag{17b}$$

Theorem 3.3 characterizes the average-iterate convergence of the average policy $\overline{\pi}^{(t)}$ (cf. (14)) across the agents, which depends logarithmically on the size of the action space, and independently on the size of the state space. Theorem 3.3 indicates that in the server-client setting with $\sigma = 0$, the convergence rate of FedNPG recovers the $\mathcal{O}(1/T)$ rate, matching that of the centralized NPG established in [AKLM21]; on the other end, in the decentralized setting where $\sigma > 0$, FedNPG slows down and eventually converges at the slower $\mathcal{O}(1/T^{2/3})$ rate.

We state the iteration complexity in Corollary 3.4.

**Corollary 3.4** (Iteration complexity of exact FedNPG). *To reach $\frac{1}{T}\sum_{t=0}^{T-1}\left(V^\star(\rho) - V^{\overline{\pi}^{(t)}}(\rho)\right) \leq \varepsilon$, the iteration complexity of FedNPG is at most $\mathcal{O}\left(\left(\frac{\sigma}{(1-\gamma)^{9/2}(1-\sigma)\varepsilon^{3/2}} + \frac{\sigma^2}{(1-\sigma)^4}\right)\sqrt{N}\log|\mathcal{A}| + \frac{1}{\varepsilon(1-\gamma)^2}\right)$.*

**Global convergence of FedNPG with inexact policy evaluation.** In practice, the policies need to be evaluated using samples collected by the agents, where the Q-functions are only estimated approximately. We are interested in gauging how the approximation error impacts the performance of FedNPG, as demonstrated in the following theorem. The formal statement, detailed discussions, and proof of this result is given in Appendix D.4.

**Theorem 3.5** (Global sublinear convergence of inexact FedNPG (informal)). *Suppose that an estimate $q_n^{\pi_n^{(t)}}$ are used in replace of $Q_n^{\pi_n^{(t)}}$ in Algorithm 1. Under the assumptions of Theorem 3.3, when $T \gtrsim \frac{\sqrt{N}\log|\mathcal{A}|\sigma^4}{(1-\sigma)^4}$ and $\eta = \left(\frac{(1-\gamma)^9(1-\sigma)^2\log|\mathcal{A}|}{32TN\sigma^2}\right)^{1/3}$, we have*

$$\frac{1}{T}\sum_{t=0}^{T-1}\left(V^\star(\rho) - V^{\overline{\pi}^{(t)}}(\rho)\right) \lesssim \frac{V^\star(d_\rho^{\pi^\star})}{(1-\gamma)T} + \frac{N^{1/3}\sigma^{2/3}}{(1-\gamma)^3(1-\sigma)^{2/3}}\left(\frac{\log|\mathcal{A}|}{T}\right)^{2/3}$$

$$+ \frac{1}{(1-\gamma)^2}\max_{n\in[N],t\in[T]}\left\|Q_n^{\pi_n^{(t)}} - q_n^{\pi_n^{(t)}}\right\|_\infty. \tag{18}$$

Equipped with existing sample complexity bounds on policy evaluation, e.g. using a simulator as in [LWCC23a], this immediate leads to the sample complexity per state-action pair at each agent to find an $\varepsilon$-optimal policy is at most

$$\widetilde{\mathcal{O}}\left(\frac{\sqrt{N}}{(1-\gamma)^{11.5}(1-\sigma)\varepsilon^{3.5}}\right) \tag{19}$$

for sufficiently small $\varepsilon$.

**Global convergence of entropy-regularized FedNPG with exact policy evaluation.** Next, we present our global convergence guarantee of entropy-regularized FedNPG with exact policy evaluation (cf. Algorithm 2).

**Theorem 3.6** (Global linear convergence of exact entropy-regularized FedNPG (informal)). *For any $\gamma \in (0,1)$ and $0 < \tau \leq 1$, there exists $\eta_0 = \min\left\{\frac{1-\gamma}{\tau}, \mathcal{O}\left(\frac{(1-\gamma)^7(1-\sigma)^2\tau}{\sigma^2 N}\right)\right\}$, such that if $0 < \eta \leq \eta_0$, then we have*

$$\left\|\overline{Q}_\tau^{(t)} - Q_\tau^\star\right\|_\infty \leq 2\gamma C_1 \rho(\eta)^t \qquad \left\|\log \pi_\tau^\star - \log \overline{\pi}^{(t)}\right\|_\infty \leq \frac{2C_1}{\tau}\rho(\eta)^t, \qquad (20)$$

*where $\overline{Q}_\tau^{(t)} := Q_\tau^{\overline{\pi}^{(t)}}$, $\rho(\eta) \leq \max\{1 - \frac{\tau\eta}{2}, \frac{3+\sigma}{4}\} < 1$, and $C_1$ is some problem-dependent constant. Furthermore, the consensus error satisfies*

$$\forall n \in [N]: \quad \left\|\log \pi_n^{(t)} - \log \overline{\pi}^{(t)}\right\|_\infty \leq 2C_1\rho(\eta)^t. \qquad (21)$$

The exact expressions of $C_1$ and $\eta_0$ are specified in Appendix D.1. Theorem 3.6 confirms that entropy-regularized FedNPG converges at a linear rate to the optimal regularized policy, which is almost independent of the size of the state-action space, highlighting the positive role of entropy regularization in federated policy optimization. When the network is fully connected, i.e. $\sigma = 0$, the iteration complexity of entropy-regularized FedNPG reduces to $\mathcal{O}\left(\frac{1}{\eta\tau}\log\frac{1}{\varepsilon}\right)$, matching that of the centralized entropy-regularized NPG established in [CWC21]. When the network is less connected, one needs to be more conservative in the choice of learning rates, leading to a higher iteration complexity, as described in the following corollary.

**Corollary 3.7** (Iteration complexity of exact entropy-regularized FedNPG). *To reach $\left\|\log\pi_\tau^\star - \log\overline{\pi}^{(t)}\right\|_\infty \leq \varepsilon$, the iteration complexity of entropy-regularized FedNPG is at most*

$$\widetilde{\mathcal{O}}\left(\max\left\{\frac{2}{\tau\eta}, \frac{4}{1-\sigma}\right\}\log\frac{1}{\varepsilon}\right) \qquad (22)$$

*up to logarithmic factors. Especially, when $\eta = \eta_0$, the best iteration complexity becomes $\widetilde{\mathcal{O}}\left(\left(\frac{N\sigma^2}{(1-\gamma)^7(1-\sigma)^2\tau^2} + \frac{1}{1-\gamma}\right)\log\frac{1}{\tau\varepsilon}\right)$.*

**Global convergence of entropy-regularized FedNPG with inexact policy evaluation.** Last but not the least, we present the informal convergence results of entropy-regularized FedNPG with inexact policy evaluation, whose formal version can be found in Appendix D.2.

**Theorem 3.8** (Global linear convergence of inexact entropy-regularized FedNPG (informal)). *Suppose that an estimate $q_{\tau,n}^{\pi_n^{(t)}}$ are used in replace of $Q_{\tau,n}^{\pi_n^{(t)}}$ in Algorithm 2. Under the assumptions of Theorem 3.6, we have*

$$\left\|\overline{Q}_\tau^{(t)} - Q_\tau^\star\right\|_\infty \leq 2\gamma\left(C_1\rho(\eta)^t + C_2\varepsilon_q\right), \quad \left\|\log\pi_\tau^\star - \log\overline{\pi}^{(t)}\right\|_\infty \leq \frac{2}{\tau}\left(C_1\rho(\eta)^t + C_2\varepsilon_q\right), \quad (23)$$

*where $\overline{Q}_\tau^{(t)} := Q_\tau^{\overline{\pi}^{(t)}}$, $\varepsilon_q := \max_{n\in[N],t\in[T]}\left\|Q_{\tau,n}^{\pi_n^{(t)}} - q_{\tau,n}^{\pi_n^{(t)}}\right\|_\infty$, $\rho(\eta) \leq \max\{1 - \frac{\tau\eta}{2}, \frac{3+\sigma}{4}\} < 1$, and $C_1, C_2$ are problem-dependent constants.*

## 4 Federated NAC with function approximation and stochastic evaluation

In this section, motivated by the design and analysis of FedNPG, we go beyond the tabular setting and exact policy evaluation, by proposing a federated natural actor-critic (FedNAC) method with function approximation and stochastic policy evaluation. Specifically, we consider the policy with function approximation under softmax parameterization is of the following form:

$$f_\xi(a|s) = \frac{\exp(\phi^\top(s,a)\boldsymbol{\xi})}{\sum_{a'\in\mathcal{A}}\exp(\phi^\top(s,a')\boldsymbol{\xi})}, \qquad (24)$$

for all $(s,a) \in \mathcal{S} \times \mathcal{A}$ and $\boldsymbol{\xi} \in \mathbb{R}^p$, where $\phi : \mathcal{S} \times \mathcal{A} \to \mathbb{R}^p$ is a known feature map. We assume $\phi$ is bounded over $\mathcal{S} \times \mathcal{A}$, i.e., there exists $C_\phi > 0$ such that $\|\phi(s,a)\|_2 \leq C_\phi$ holds for all $(s,a) \in \mathcal{S} \times \mathcal{A}$.

Following [AKLM21, YDG$^+$22], given any $\boldsymbol{w} \in \mathbb{R}^p$, $Q : \mathcal{S} \times \mathcal{A} \to \mathbb{R}$ and probability distribution $\zeta \in \Delta(\mathcal{S} \times \mathcal{A})$ over the state-action space, we define the *function approximation error* $\ell(\boldsymbol{w}, Q, \zeta)$ as follows:

$$\ell(\boldsymbol{w}, Q, \zeta) := \mathbb{E}_{(s,a) \sim \zeta}\left[\left(\boldsymbol{w}^\top \phi(s,a) - Q(s,a)\right)^2\right]. \tag{25}$$

By searching for $\boldsymbol{w}$ that minimizes $\ell(\boldsymbol{w}, Q, \zeta)$, it approximates $Q(s,a)$ using the feature map $\phi$ with respect to the distribution $\zeta$.

**Algorithm design.** Let us now discuss the high-level design of FedNAC, which is presented in Algorithm 3, with more details provided in Appendix C.2. At the $t$-th iteration ($t = 0, \ldots, T-1$), denote the actor (concerning the policies) parameters of all agents as $\boldsymbol{\xi}^{(t)} = (\boldsymbol{\xi}_1^{(t)}, \ldots, \boldsymbol{\xi}_N^{(t)})^\top \in \mathbb{R}^{N \times p}$, and the critic parameters of all agents as $\boldsymbol{w}^{(t)} = (\boldsymbol{w}_1^{(t)}, \ldots, \boldsymbol{w}_N^{(t)})^\top \in \mathbb{R}^{N \times p}$ (concerning the local Q-values) and $\boldsymbol{h}^{(t)} = (\boldsymbol{h}_1^{(t)}, \ldots, \boldsymbol{h}_N^{(t)})^\top \in \mathbb{R}^{N \times p}$ (concerning the global Q-values).

- First, the critic parameter $\boldsymbol{w}_n^{(t)}$ is locally updated at each agent by aiming to minimize $\ell(\boldsymbol{w}, Q_n^{(t)}, \tilde{d}_n^{(t)})$ (cf. (25)) with gradient descent, where $Q_n^{(t)}$ is the local Q-function of the local policy $f_{\boldsymbol{\xi}_n^{(t)}}$, and $\tilde{d}_n^{(t)}$ is the state-action visitation distribution induced by the local policy $f_{\boldsymbol{\xi}_n^{(t)}}$ and an initial state-action distribution $\nu$ (determined from the data sampling mechanism, cf. (30)). However, since $Q_n^{(t)}$ is not directly available, it needs to be estimated from samples. Therefore, the critic update takes $K$ steps of stochastic gradient descent with critic learning rate $\beta$, given by

$$\widetilde{\boldsymbol{w}}_{k+1} = \widetilde{\boldsymbol{w}}_k - \beta\left(\widetilde{\boldsymbol{w}}_k^\top \phi(s_k, a_k) - \widehat{Q}_\xi(s_k, a_k)\right)\phi(s_k, a_k),$$

  for $k = 0, \ldots, K-1$, where $(s_k, a_k)$ is sampled on the local policy $f_{\boldsymbol{\xi}_n^{(t)}}$, and $\widehat{Q}_\xi(s_k, a_k)$ is a careful estimate of the Q-value using a trajectory with expected length $1/(1-\gamma)$ (see Algorithm 5 in Appendix C.2 adopted from [YDG$^+$22, Lemma 4]), and $\widetilde{\boldsymbol{w}}_0 = \boldsymbol{0}$ for simplicity. The final critic is updated as $\boldsymbol{w}_n^{(t)} = \frac{1}{K}\sum_{k=1}^K \widetilde{\boldsymbol{w}}_k$. The total sample complexity of the critic update per iteration is then on the order of $K/(1-\gamma)$.

- Next, the critic parameter $\boldsymbol{h}_n^{(t)}$ for estimating the global Q-function can then be estimated by averaging with the neighbors with the Q-tracking term, given by $\boldsymbol{h}^{(t)} = \boldsymbol{W}\left(\boldsymbol{h}^{(t-1)} + \boldsymbol{w}^{(t)} - \boldsymbol{w}^{(t-1)}\right)$.

- Finally, the actor parameter $\boldsymbol{\xi}_n^{(t)}$ can be updated via averaging with the neighbors along with the policy gradient informed by $\boldsymbol{h}_n^{(t)}$, given by $\boldsymbol{\xi}^{(t+1)} = \boldsymbol{W}\left(\boldsymbol{\xi}^{(t)} + \alpha\boldsymbol{h}^{(t)}\right)$, where $\alpha$ is the learning rate of the actor.

Note that the sample complexity of FedNAC is on the order of $KT/(1-\gamma)$. An important aspect of the FedNAC method is that the policy is updated using trajectory data collected via executing the learned policy, which is closer to practice and more challenging to learn than using the generative model.

**Theoretical guarantees.** We first state the assumptions that are needed to guarantee the convergence of Algorithm 3, which are all commonly used in the literature, e.g., [YDG$^+$22, AKLM21]. To begin, we require the covariance matrix of the feature map induced by the initial state-action distribution $\nu$ satisfies the following assumption to guarantee the convergence of the critic.

**Assumption 4.1** (PSD of the covariance matrix of the feature map). There exists $\mu > 0$ such that $\mathbb{E}_{(s,a) \sim \nu}\left[\phi(s,a)\phi^\top(s,a)\right] = \Sigma_\nu \geq \mu \boldsymbol{I}$.

We also need to ensure that the Q-values can be well approximated by the linear function approximation using feature map $\phi(s,a)$, which is captured next.

**Assumption 4.2** (Bounded approximation error). For each $n \in [N]$, there exists $\varepsilon_{\text{approx}}^n \geq 0$ such that for all $t \in \mathbb{N}$, it holds that $\mathbb{E}\left[\ell\left(\boldsymbol{w}_{\star,n}^{(t)}, Q_n^{(t)}, \tilde{d}_n^{(t)}\right)\right] \leq \varepsilon_{\text{approx}}^n$, where $\boldsymbol{w}_{\star,n}^{(t)} := \arg\min_{\boldsymbol{w}} \ell\left(\boldsymbol{w}_{\star,n}^{(t)}, Q_n^{(t)}, \tilde{d}_n^{(t)}\right)$.

We denote the average approximation error as $\bar{\varepsilon}_{\text{approx}} = \frac{1}{N}\sum_{n=1}^N \varepsilon_{\text{approx}}^n$. Similar as [YDG$^+$22], we need the following assumption that bounds the transfer errors due to distribution shifts.

**Assumption 4.3** (Bounded transfer error). There exists $C_\nu > 0$ such that for all $n \in [N]$ and $t \in \mathbb{N}$, it holds that $\mathbb{E}_{(s,a) \sim \tilde{d}_n^{(t)}} \left[ \left( \frac{h^\pi(s,a)}{\tilde{d}_n^{(t)}(s,a)} \right)^2 \right] \leq C_\nu$, where $h^\pi(s,a)$ is the state-action visitation distribution induced by any policy $\pi$ from initial state distribution $\rho$.

Note that if we choose $\nu(s,a) > 0$ for all $(s,a) \in \mathcal{S} \times \mathcal{A}$, then Assumption 4.3 is guaranteed to hold true (see Lemma E.4 in Appendix E). We are now ready to state the convergence guarantee, whose formal version and proof could be found in Appendix E.

**Theorem 4.4** (Convergence rate of Algorithm 3 (informal)). *Let $\boldsymbol{\xi}_1^{(0)} = \cdots = \boldsymbol{\xi}_N^{(0)}$ in FedNAC. Denoting $\bar{\boldsymbol{\xi}}^{(t)} := \frac{1}{N} \sum_{n=1}^N \boldsymbol{\xi}_n^{(t)}$, and $\bar{f}^{(t)} := f_{\bar{\xi}^{(t)}}$ as the average policy. Then under Assumption 3.1, 4.1, 4.2 and 4.3, with appropriately chosen learning rates $\alpha$ and $\beta$, as long as the number of actor iterations satisfies*

$$T \gtrsim \max \left\{ \frac{\sigma}{\varepsilon^{3/2}(1-\gamma)^{17/4}(1-\sigma)^{3/2}}, \frac{1}{\varepsilon(1-\gamma)}, \frac{\sigma^{1/4}}{\varepsilon^{3/4}(1-\sigma)^{3/8}(1-\gamma)^{7/8}N^{3/8}}, \frac{\sigma^4}{(1-\gamma)^2(1-\sigma)^6} \right\}$$

*and the number of critic iterations satisfies $K = \mathcal{O}\left( \frac{1}{(1-\gamma)^6 \varepsilon^2} \right)$, it holds that*

$$V^\star(\rho) - \frac{1}{T} \sum_{t=0}^{T-1} V^{\bar{f}^{(t)}}(\rho) \lesssim \varepsilon + \frac{\bar{\varepsilon}_{approx}}{1-\gamma}. \tag{26}$$

In the server-client setting when $\sigma = 0$, to reach (26), it suffices to choose $T = \mathcal{O}\left( \frac{1}{(1-\gamma)\varepsilon} \right)$ and $K = \mathcal{O}\left( \frac{1}{(1-\gamma)^6 \varepsilon^2} \right)$, leading to a total sample complexity of $KT/(1-\gamma) = \mathcal{O}\left( \frac{1}{(1-\gamma)^8 \varepsilon^3} \right)$ per agent, and $T = \mathcal{O}\left( \frac{1}{(1-\gamma)\varepsilon} \right)$ rounds of communication. The sample complexity matches that of (centralized) Q-NPG established in [YDG$^+$22] with a single agent. On the other end, in the fully decentralized setting when $\sigma$ is not close to 0, FedNAC requires $\mathcal{O}\left( \frac{1}{(1-\gamma)^{45/4} \varepsilon^{7/2}(1-\sigma)^{3/2}} \right)$ samples for each agent and $\mathcal{O}\left( \frac{1}{\varepsilon^{3/2}(1-\gamma)^{17/4}(1-\sigma)^{3/2}} \right)$ rounds of communication to reach (26), for sufficiently small $\varepsilon$. Encouragingly, the dependency on the accuracy level $\varepsilon$ — the dominating factor — in the sample complexity matches that of FedNPG given in (19) when assuming access to the generative model, which allows query of arbitrary state-action pairs. In contrast, FedNAC only collects on-policy samples, and therefore is much more challenging to guarantee its convergence.

## 5 Conclusions

This work proposes the first provably efficient federated NPG (FedNPG) methods for solving vanilla and entropy-regularized multi-task RL problems in the fully decentralized setting. The established finite-time global convergence guarantees are almost independent of the size of the state-action space up to some logarithmic factor, and illuminate the impacts of the size and connectivity of the network. Furthermore, the proposed FedNPG methods are provably robust vis-a-vis inexactness of local policy evaluations. Last but not least, we also propose FedNAC, which can be viewed as an extension of FedNPG with function approximation and stochastic policy evaluation, and establish its finite-time sample complexity. Future directions include generalizing the framework of federated policy optimization to allow personalized policy learning in a shared environment.

## Acknowledgments and Disclosure of Funding

The work of T. Yang, S. Cen and Y. Chi are supported in part by the grants ONR N00014-19-1-2404, NSF CCF-1901199, CCF-2106778, AFRL FA8750-20-2-0504, and a CMU Cylab seed grant. The work of Y. Wei is supported in part by the the NSF grants DMS-2147546/2015447, CAREER award DMS-2143215, CCF-2106778, and the Google Research Scholar Award. The work of Y. Chen is supported in part by the Alfred P. Sloan Research Fellowship, the Google Research Scholar Award, the AFOSR grant FA9550-22-1-0198, the ONR grant N00014-22-1-2354, and the NSF grants CCF-2221009 and CCF-1907661. S. Cen is also gratefully supported by Wei Shen and Xuehong Zhang Presidential Fellowship, Boeing Scholarship, and JP Morgan Chase PhD Fellowship.

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

## A   Related work

**Global convergence of NPG methods for tabular MDPs.** [AKLM21] first establishes a $\mathcal{O}(1/T)$ last-iterate convergence rate of the NPG method under softmax parameterization with constant step size, assuming access to exact policy evaluation. When entropy regularization is in place, [CWC21] establishes a global linear convergence to the optimal regularized policy for the entire range of admissible constant learning rates using softmax parameterization and exact policy evaluation, which is further shown to be stable in the presence of $\ell_\infty$ policy evaluation errors. The iteration complexity of NPG methods is nearly independent with the size of the state-action space, which is in sharp contrast to softmax policy gradient methods that may take exponential time to converge [LWCC23b, MXSS20]. [Lan23] proposed a more general framework through the lens of mirror descent for regularized RL with global linear convergence guarantees, which is further generalized in [ZCH+23, LLZ23]. Earlier analysis of regularized MDPs can be found in [SEM20]. Besides, [Xia22] proves that vanilla NPG also achieves linear convergence when geometrically increasing learning rates are used; see also [KJVM21, BR21]. [ZLK+22] developed an anchor-changing NPG method for multi-task RL under various optimality criteria in the centralized setting.

**Convergence and sample complexity results of NAC.** The convergence and sample complexity of a variety of natural actor–critic methods (NACs) are extensively studied in the literature [BSGL09, WCYW19, KDRM22, AKLM21, YDG+22]. More pertinent to our work, [AKLM21] introduced Q-NPG—a sample version of the NPG method with function approximation under softmax parameterization —and obtained a convergence rate of $\mathcal{O}(1/\sqrt{T})$. [YDG+22] weakens some of its assumptions and improves the convergence rate to $\mathcal{O}(1/T)$ and gives the $\widetilde{\mathcal{O}}(1/\varepsilon^3)$ sample complexity using a constant actor learning rate. The FedNAC method we propose in this paper can be seen as a decentralized version of Q-NPG, and in the server-client setting where the network is fully connected, our convergence rate and sample complexity match those in [YDG+22].

**Distributed and federated RL.** There have been a variety of settings being set forth for distributed and federated RL. [MBM+16, ESM+18, ARB+19, KSJM22, WJC23] focused on developing federated versions of RL algorithms to accelerate training, assuming all agents share the same transition kernel and reward function; in particular, [KSJM22, WJC23, WSJC24] established the provable benefits of federated learning in terms of linear speedup. More pertinent to our work, [ZRY+23, AR21] considered the federated multi-task framework, allowing different agents having private reward functions. [ZRY+23] proposed an empirically probabilistic algorithm that can seek an optimal policy under the server-client setting, while [AR21] developed new attack methods in the presence of adversarial agents. Recently [LWA+23] discussed how to avoid transmitting the Hessian matrix during communication in the server-client setting where all agents share the same reward function. Different from the FRL framework, [CZGB21, CZC21, OPA+17, KMP12, CFGW22, ZAD+21] considered the distributed multi-agent RL setting where the agents interact with a dynamic environment through a multi-agent Markov decision process, where each agent can have their own state or action spaces. [ZAD+21] developed a decentralized policy gradient method where different agents have different MDPs, where a special case of their setting recovers ours. However, the convergence rate developed in [ZAD+21] has rather pessimistic dependencies with the size of the state-action space, together with other parameters, without leveraging natural policy gradients and gradient tracking techniques.

**Decentralized first-order optimization algorithms.** Early work of consensus-based first-order optimization algorithms for the fully decentralized setting include but are not limited to [LO08, NO09, DAW11]. Gradient tracking, which leverages the idea of dynamic average consensus [ZM10] to track the gradient of the global objective function, is a popular method to improve the convergence speed [QL17, NOS17, DLS16, PN21, LCCC20].

## B   Additional Discussion

### B.1   Application Related to Federated Multi-task RL

In this section, we elaborate more on our motivation and the application scenarios where federated multi-task RL becomes highly relevant.

We first provide some key motivations for our federated multi-task RL setting as follows.

- Efficient knowledge transfer: multi-task RL enables agents to transfer knowledge across related tasks, accelerating learning and improving performance by leveraging experiences gained from one task to another. For instance, in our healthcare example in Section 1, by learning across hospitals with varying demographics, the agent can identify treatment strategies that are effective across diverse patient populations without directly accessing sensitive patient information.

- Generalization and adaptability: agents trained with multi-task RL can generalize their learned policies, adapt to new tasks, and handle diverse environments more effectively, enhancing their robustness and adaptability. In the healthcare example, an optimal treatment over different hospitals better adapts to variations in patient characteristics.

- Resource optimization: training a single policy for multiple tasks optimizes resource usage compared to training separate policies for each task, making it more efficient in scenarios with limited data or computational resources. In the healthcare example, the collaborative approach enhances learning efficiency and scalability while preserving data privacy, particularly in settings where each hospital has limited access to patient information.

Below we provide more application scenarios of our setting.

1. To enhance ChatGPT's performance across different tasks or domains [MA22, RTR$^+$23], one might consult domain experts to chat and rate ChatGPT's outputs for solving different tasks, and train ChatGPT in a federated manner without exposing private data or feedback of each expert.

2. Our setting is especially suitable for the multi-task problems where each agent only have partial access of the "global" task. There are a lot of such problems.

   - An example is the problem we consider in our experiments (see Appendix H), where we distributedly train the agents to learn a shared policy to follow a predetermined trajectory while each agent only has partial information of this trajectory.
   - The above problem could be seen as a simplified version of the Unmanned Aerial Vehicle (UAV) Patrol Mission, each unmanned aerial vehicle (UAV) patrols only in a specific area, and they need to collectively train a strategy utilizing information from the entire patrol range.
   - In the game setting, different agents aim to train a character to perform well in multiple tasks, and each agent trains on one task.

Despite the promise, provably efficient algorithms for federated multi-task RL remain substantially under-explored, especially in the fully decentralized setting. Our work is the first to provide efficient algorithms with global convergence guarantees for federated multi-task RL.

## B.2 Theoretical Contribution

In this section, we stress that while our work is built upon the algorithmic ideas in the distributed learning, reinforcement learning and optimization literature, it is not a strightforward combination and the theoretical analysis is by no means trivial.

One key difficulty is to estimate the global Q-functions using only neighboring information and local data. To address this issue, we invoke the "Q-tracking" step (see Algorithm 1, 2), which is inspired by the gradient tracking method in decentralized optimization. Note that this generalization is highly non-trivial: to the best of our knowledge, the utility of gradient tracking has not been exploited in policy optimization, and the intrinsic nonconcavity issue, together with the use of natural gradients, prevents us from directly using the results from decentralized optimization. It is thus of great value to study if the combination of NPG and gradient tracking could lead to fast globally convergent algorithms as in the standard decentralized optimization literature despite the nonconcavity.

Besides, due to the lack of global information sharing, care needs to be taken to judiciously balance the use of neighboring information (to facilitate consensus) and local data (to facilitate learning) when updating the policy. Compared to the centralized version of our proposed algorithms, a much more delicate theoretical analysis is required to prove our convergence results. For example, the key step to establish the convergence rate of the single-agent exact entropy-regularized NPG is to form the 2nd-order linear system in Eq. (46) in [CCC$^+$22a], while in our corresponding analysis,

a 4th-order linear system in Eq. (49) is needed, where the inequality in each line is non-trivial and requires the introduction of some intricate and novel auxiliary lemmas, see Appendix D.

## C Omitted Algorithms

### C.1 Federated NPG (FedNPG) with entropy regularization

We record the entropy-regularized FedNPG method in Algorithm 2 here due to space limits.

---

**Algorithm 2** Federated NPG (FedNPG) with entropy regularization

---

1: **Input:** learning rate $\eta > 0$, iteration number $T \in \mathbb{N}_+$, mixing matrix $\boldsymbol{W} \in \mathbb{R}^{N \times N}$, regularization coefficient $\tau > 0$.
2: **Initialize:** $\boldsymbol{\pi}^{(0)}, \boldsymbol{T}^{(0)} = \boldsymbol{Q}_\tau^{(0)}$.
3: **for** $t = 0, 1, \cdots$ **do**
4:    Update the policy for each $(s, a) \in \mathcal{S} \times \mathcal{A}$:

$$\log \boldsymbol{\pi}^{(t+1)}(a|s) = \boldsymbol{W}\left(\left(1 - \frac{\eta\tau}{1-\gamma}\right)\log \boldsymbol{\pi}^{(t)}(a|s) + \frac{\eta}{1-\gamma}\boldsymbol{T}^{(t)}(s,a)\right) - \log \boldsymbol{z}^{(t)}(s),$$
(27)

where $\boldsymbol{z}^{(t)}(s) = \sum_{a' \in \mathcal{A}} \exp\left\{\boldsymbol{W}\left(\left(1 - \frac{\eta\tau}{1-\gamma}\right)\log \boldsymbol{\pi}^{(t)}(a'|s) + \frac{\eta}{1-\gamma}\boldsymbol{T}^{(t)}(s,a')\right)\right\}$.

5:    Evaluate $\boldsymbol{Q}_\tau^{(t+1)}$.
6:    Update the global Q-function estimate for each $(s, a) \in \mathcal{S} \times \mathcal{A}$:

$$\boldsymbol{T}^{(t+1)}(s,a) = \boldsymbol{W}\Big(\boldsymbol{T}^{(t)}(s,a) + \underbrace{\boldsymbol{Q}_\tau^{(t+1)}(s,a) - \boldsymbol{Q}_\tau^{(t)}(s,a)}_{\text{Q-tracking}}\Big). \qquad (U_T)$$

7: **end for**

---

### C.2 Development of FedNAC

For any policy $\pi$, we let $d_{s_0}^\pi$ denote the discounted state visitation distribution of $\pi$ given an initial state $s_0 \in \mathcal{S}$, i.e.,

$$\forall s \in \mathcal{S}: \quad d_{s_0}^\pi(s) := (1-\gamma)\sum_{t=0}^\infty \gamma^t \mathbb{P}(s_t = s|s_0). \qquad (28)$$

For a distribution $\rho \in \Delta(\mathcal{S})$, we define $d_\rho^\pi(s) = \mathbb{E}_{s_0 \sim \rho}[d_{s_0}^\pi(s)]$. We also define the *state-action visitation distribution* $\bar{d}_\rho^\pi$ as

$$\bar{d}_\rho^\pi(s,a) := d_\rho^\pi(s)\pi(a|s) = (1-\gamma)\mathbb{E}_{s_0 \sim \rho}\left[\sum_{t=0}^\infty \gamma^t \mathbb{P}(s_t = s, a_t = a|s_0)\right]. \qquad (29)$$

Furthermore, we extend the definition of $\bar{d}_\rho^\pi$ by specifying the initial state-action distribution $\nu \in \Delta(\mathcal{S} \times \mathcal{A})$ and define

$$\tilde{d}_\nu^\pi(s,a) := (1-\gamma) \mathbb{E}_{(s_0,a_0) \sim \nu}\left[\sum_{t=0}^\infty \gamma^t \mathbb{P}(s_t = s, a_t = a|s_0, a_0)\right]. \qquad (30)$$

Our proposed federated NAC method FedNAC could be seen as a decentralized version of Q-NPG method [AKLM21, YDG$^+$22], which we briefly review as follows.

**Q-NPG method.** Q-NPG is a sample version of NPG with function approximation which is suitable for the case where $\mathcal{S}$ or $\mathcal{A}$ is large or infinite. We consider the policy with function approximation under softmax parameterization (24).

Given an approximate solution $\boldsymbol{w}^{(t)}$ for minimizing the function approximation error $\ell(\boldsymbol{w}, Q^{f_{\xi^{(t)}}}, \tilde{d}_\nu^{f_{\xi^{(t)}}})$ (see (25)), the Q-NPG update rule $\boldsymbol{\xi}^{(t+1)} = \boldsymbol{\xi}^{(t)} + \alpha\boldsymbol{w}^{(t)}$, when plugged in

parameterization (24), results in the following policy update rule when we set $\alpha = \eta/(1-\gamma)$:

$$f^{(t+1)}(a|s) \propto f^{(t)}(a|s) \exp\left(\frac{\eta \phi^\top(s,a) \boldsymbol{w}^{(t)}}{1-\gamma}\right), \tag{31}$$

which could be seen as the function approximation version of the update rule (8) of vanilla NPG method.

**Federated NAC method.** *FedNAC* (describe in Section 4) is presented in Algorithm 3, whose subroutines are written in Algorithm 4, 5. In each iteration $t$ of FedNAC, each agent $n$ updates the critic parameter $\boldsymbol{w}_n^{(t)}$ locally using Algorithm 4, which aims to minimize $\ell(\boldsymbol{w}, Q_n^{(t)}, \tilde{d}_n^{(t)})$ by stochastic gradient descent. Note that since we don't know the Q-function $Q_n^{(t)}$ in the gradients, we need to invoke Algorithm 5 [YDG$^+$22, Algorithm 3] to give an unbiased estimate $\widehat{Q}_n^{(t)}(s,a)$, where $(s,a)$ is sampled from $\tilde{d}_n^{(t)}$ (cf. Theorem E.1). As a consequence, in line 4 of Algorithm 4, we have

$$\mathbb{E}\left[\widehat{\nabla}_w \ell(\widetilde{\boldsymbol{w}}_k, \widehat{Q}^\pi, \tilde{d}^{f_\xi})\right] = \nabla_w \ell(\widetilde{\boldsymbol{w}}_k, \widehat{Q}^\pi, \tilde{d}^{f_\xi}). \tag{32}$$

In each actor iteration, agents share with their neighbors actor and critic parameters, where the tracking scheme is also used.

---

**Algorithm 3** Federated Natural Actor-Critic (FedNAC)

---

1: **Input:** number of actor iterations $T$, number of critic iterations $K$, actor learning rate $\alpha$, critic learning rate $\beta$, discounted factor $\gamma \in [0,1)$
2: **Initialization:** initial state-action distribution $\nu$, actor parameter $\boldsymbol{\xi}^{(0)} = (\boldsymbol{\xi}_1^{(0)\top}, \cdots, \boldsymbol{\xi}_N^{(0)\top})^\top \in \mathbb{R}^{N \times p}$, $\boldsymbol{h}^{(-1)} = \boldsymbol{w}^{(-1)} = \boldsymbol{0} \in \mathbb{R}^{N \times p}$
3: **for** $t = 0, \cdots, T-1$ **do**
4:     Critic update: $\boldsymbol{w}_n^{(t)} = $ Critic$(K, \nu, \boldsymbol{\xi}_n^{(t)}, \gamma, \beta, r_n)$, $n \in [N]$ (Algorithm 4)
5:     Update the critic parameter for estimating the global Q-function:

$$\boldsymbol{h}^{(t)} = \boldsymbol{W}\left(\boldsymbol{h}^{(t-1)} + \boldsymbol{w}^{(t)} - \boldsymbol{w}^{(t-1)}\right) \tag{33}$$

6:     Actor update:

$$\boldsymbol{\xi}^{(t+1)} = \boldsymbol{W}\left(\boldsymbol{\xi}^{(t)} + \alpha \boldsymbol{h}^{(t)}\right) \tag{34}$$

7: **end for**

---

---

**Algorithm 4** Critic$(K, \nu, \boldsymbol{\xi}, \gamma, \beta, r)$: sample-based regression solver to minimize $\ell(\boldsymbol{w}, Q_n^{(t)}, \tilde{d}_n^{(t)})$

---

1: **Initialize:** critic parameter $w_0 \in \mathbb{R}^p$
2: **for** $k = 0, \cdots, K-1$ **do**
3:     Sampling: $(s_k, a_k), \widehat{Q}^\pi(s_k, a_k) = $Q-Sampler$(\nu, f_\xi, \gamma, r)$ (Algorithm 5)
4:     Compute the stochastic gradient estimator of $L_Q$:

$$\widehat{\nabla}_w \ell(\widetilde{\boldsymbol{w}}_k, \widehat{Q}^\pi, \tilde{d}^{f_\xi}) = 2\left(\widetilde{\boldsymbol{w}}_k^\top \phi(s_k, a_k) - \widehat{Q}^\pi(s_k, a_k)\right)\phi(s_k, a_k) \tag{35}$$

5:     Critic Update: $\widetilde{\boldsymbol{w}}_{k+1} = \widetilde{\boldsymbol{w}}_k - \beta\widehat{\nabla}_w \ell(\widetilde{\boldsymbol{w}}_k, \widehat{Q}^\pi, \tilde{d}^{f_\xi})$
6: **end for**
7: **Output:** $\boldsymbol{w}_{\text{out}} = \frac{1}{K}\sum_{k=1}^K \widetilde{\boldsymbol{w}}_k$

---

## D Convergence analysis of FedNPG

For technical convenience, we present first the analysis for entropy-regularized FedNPG and then for vanilla FedNPG.

**Algorithm 5** Q-Sampler($\nu, \pi, \gamma, r$)

1: **Initialize:** $(s_0, a_0) \sim \nu$, time step $h, t = 0$, variable $X \sim$ Bernoulli($\gamma$)
2: **while** $X = 1$ **do**
3:     Sample $s_{h+1} \sim P(\cdot | s_h, a_h)$
4:     Sample $a_{h+1} \sim \pi(\cdot | s_{h+1})$
5:     $h \leftarrow h + 1$
6:     $X \sim$ Bernoulli($\gamma$)
7: **end while**
8: Set $xc(s_h, a_h) = r(s_h, a_h)$, $X \sim$ Bernoulli($\gamma$), $t = h$
9: **while** $X = 1$ **do**
10:     Sample $s_{t+1} \sim P(\cdot | s_t, a_t)$
11:     Sample $a_{t+1} \sim \pi(\cdot | s_{t+1})$
12:     $\widehat{Q}^\pi(s_h, a_h) \leftarrow \widehat{Q}^\pi(s_h, a_h) + r(s_{t+1}, a_{t+1})$
13:     $t \leftarrow t + 1$
14:     $X \sim$ Bernoulli($\gamma$)
15: **end while**
16: **Output:** $(s_h, a_h)$ and $\widehat{Q}^\pi(s_h, a_h)$

### D.1 Analysis of entropy-regularized FedNPG with exact policy evaluation

To facilitate analysis, we introduce several notation below. For all $t \geq 0$, we recall $\overline{\pi}^{(t)}$ as the normalized geometric mean of $\{\pi_n^{(t)}\}_{n \in [N]}$:

$$\overline{\pi}^{(t)} := \text{softmax}\left(\frac{1}{N} \sum_{n=1}^{N} \log \pi_n^{(t)}\right), \tag{36}$$

from which we can easily see that for each $(s, a) \in \mathcal{S} \times \mathcal{A}$, $\overline{\pi}^{(t)}(a|s) \propto \left(\prod_{n=1}^{N} \pi_n^{(t)}(a|s)\right)^{\frac{1}{N}}$. We denote the soft $Q$-functions of $\overline{\pi}^{(t)}$ by $\overline{\boldsymbol{Q}}_\tau^{(t)}$:

$$\overline{\boldsymbol{Q}}_\tau^{(t)} := \begin{pmatrix} Q_{\tau,1}^{\overline{\pi}^{(t)}} \\ \vdots \\ Q_{\tau,N}^{\overline{\pi}^{(t)}} \end{pmatrix}. \tag{37}$$

In addition, we define $\widehat{Q}_\tau^{(t)}, \overline{Q}_\tau^{(t)} \in \mathbb{R}^{|\mathcal{S}||\mathcal{A}|}$ and $\overline{V}_\tau^{(t)} \in \mathbb{R}^{|\mathcal{S}|}$ as follows

$$\widehat{Q}_\tau^{(t)} := \frac{1}{N} \sum_{n=1}^{N} Q_{\tau,n}^{\pi_n^{(t)}}, \tag{38a}$$

$$\overline{Q}_\tau^{(t)} := Q_\tau^{\overline{\pi}^{(t)}} = \frac{1}{N} \sum_{n=1}^{N} Q_{\tau,n}^{\overline{\pi}^{(t)}}. \tag{38b}$$

$$\overline{V}_\tau^{(t)} := V_\tau^{\overline{\pi}^{(t)}} = \frac{1}{N} \sum_{n=1}^{N} V_{\tau,n}^{\overline{\pi}^{(t)}}. \tag{38c}$$

For notational convenience, we also denote

$$\alpha := 1 - \frac{\eta\tau}{1-\gamma}. \tag{39}$$

Following [CCC$^+$22b], we introduce the following auxiliary sequence $\{\boldsymbol{\xi}^{(t)} = (\xi_1^{(t)}, \cdots, \xi_N^{(t)})^\top \in \mathbb{R}^{N \times |\mathcal{S}||\mathcal{A}|}\}_{t=0,1,\ldots}$, each recursively defined as

$$\forall (s, a) \in \mathcal{S} \times \mathcal{A}: \quad \boldsymbol{\xi}^{(0)}(s, a) := \frac{\|\exp(Q_\tau^\star(s, \cdot)/\tau)\|_1}{\left\|\exp\left(\frac{1}{N} \sum_{n=1}^{N} \log \pi_n^{(0)}(\cdot|s)\right)\right\|_1} \cdot \boldsymbol{\pi}^{(0)}(a|s), \tag{40a}$$

$$\log \boldsymbol{\xi}^{(t+1)}(s, a) = \boldsymbol{W}\left(\alpha \log \boldsymbol{\xi}^{(t)}(s, a) + (1-\alpha)\boldsymbol{T}^{(t)}(s, a)/\tau\right), \tag{40b}$$

where $\boldsymbol{T}^{(t)}(s, a)$ is updated via (16). Similarly, we introduce an averaged auxiliary sequence $\{\overline{\xi}^{(t)} \in \mathbb{R}^{|\mathcal{S}||\mathcal{A}|}\}$ given by

$$\forall (s, a) \in \mathcal{S} \times \mathcal{A} : \quad \overline{\xi}^{(0)}(s, a) := \|\exp\left(Q_\tau^\star(s, \cdot)/\tau\right)\|_1 \cdot \overline{\pi}^{(0)}(a|s) , \tag{41a}$$

$$\log \overline{\xi}^{(t+1)}(s, a) = \alpha \log \overline{\xi}^{(t)}(s, a) + (1 - \alpha)\widehat{Q}_\tau^{(t)}(s, a)/\tau. \tag{41b}$$

We introduces four error metrics defined as

$$\Omega_1^{(t)} := \|u^{(t)}\|_\infty , \tag{42a}$$

$$\Omega_2^{(t)} := \|v^{(t)}\|_\infty , \tag{42b}$$

$$\Omega_3^{(t)} := \|Q_\tau^\star - \tau \log \overline{\xi}^{(t)}\|_\infty , \tag{42c}$$

$$\Omega_4^{(t)} := \max \left\{ 0, -\min_{s,a} \left( \overline{Q}_\tau^{(t)}(s, a) - \tau \log \overline{\xi}^{(t)}(s, a) \right) \right\} , \tag{42d}$$

where $u^{(t)}, v^{(t)} \in \mathbb{R}^{|\mathcal{S}||\mathcal{A}|}$ are defined as

$$u^{(t)}(s, a) := \|\log \boldsymbol{\xi}^{(t)}(s, a) - \log \overline{\xi}^{(t)}(s, a)\mathbf{1}_N\|_2 , \tag{43}$$

$$v^{(t)}(s, a) := \|\boldsymbol{T}^{(t)}(s, a) - \widehat{Q}_\tau^{(t)}(s, a)\mathbf{1}_N\|_2 . \tag{44}$$

We collect the error metrics above in a vector $\boldsymbol{\Omega}^{(t)} \in \mathbb{R}^4$:

$$\boldsymbol{\Omega}^{(t)} := \left( \Omega_1^{(t)}, \Omega_2^{(t)}, \Omega_3^{(t)}, \Omega_4^{(t)} \right)^\top . \tag{45}$$

With the above preparation, we are ready to state the convergence guarantee of Algorithm 2 in Theorem D.1 below, which is the formal version of Theorem 3.6.

**Theorem D.1.** *For any $N \in \mathbb{N}_+, \tau > 0, \gamma \in (0, 1)$, there exists $\eta_0 > 0$ which depends only on $N, \gamma, \tau, \sigma, |\mathcal{A}|$, such that if $0 < \eta \leq \eta_0$ and $1 - \sigma > 0$, then the updates of Algorithm 2 satisfy*

$$\|\overline{Q}_\tau^{(t)} - Q_\tau^\star\|_\infty \leq 2\gamma\rho(\eta)^t \|\boldsymbol{\Omega}^{(0)}\|_2 , \tag{46}$$

$$\|\log \pi_\tau^\star - \log \overline{\pi}^{(t)}\|_\infty \leq \frac{2}{\tau}\rho(\eta)^t \|\boldsymbol{\Omega}^{(0)}\|_2 , \tag{47}$$

*where*

$$\rho(\eta) \leq \max \left\{ 1 - \frac{\tau\eta}{2}, \frac{3 + \sigma}{4} \right\} < 1 .$$

*Moreover, the consensus errors satisfy:*

$$\forall n \in [N] : \quad \|\log \pi_n^{(t)} - \log \overline{\pi}^{(t)}\|_\infty \leq 2\rho(\eta)^t \|\boldsymbol{\Omega}^{(0)}\|_2 . \tag{48}$$

The dependency of $\eta_0$ on $N, \gamma, \tau, \sigma, |\mathcal{A}|$ is made clear in Lemma D.3 that will be presented momentarily in this section. The rest of this section is dedicated to the proof of Theorem D.1. We first state a key lemma that tracks the error recursion of Algorithm 2.

**Lemma D.2.** *The following linear system holds for all $t \geq 0$:*

$$\boldsymbol{\Omega}^{(t+1)} \leq \underbrace{\begin{pmatrix} \sigma\alpha & \frac{\eta\sigma}{1-\gamma} & 0 & 0 \\ S\sigma & \left(1 + \frac{\eta M\sqrt{N}}{1-\gamma}\sigma\right)\sigma & \frac{(2+\gamma)\eta MN}{1-\gamma}\sigma & \frac{\gamma\eta MN}{1-\gamma}\sigma \\ (1-\alpha)M & 0 & (1-\alpha)\gamma + \alpha & (1-\alpha)\gamma \\ \frac{2\gamma+\eta\tau}{1-\gamma}M & 0 & 0 & \alpha \end{pmatrix}}_{=:\boldsymbol{A}(\eta)} \boldsymbol{\Omega}^{(t)} , \tag{49}$$

*where we let*

$$S := M\sqrt{N} \left( 2\alpha + (1 - \alpha) \cdot \sqrt{2N} + \frac{1 - \alpha}{\tau} \cdot \sqrt{N}M \right) , \tag{50}$$

*and*

$$M := \frac{1 + \gamma + 2\tau(1 - \gamma)\log|\mathcal{A}|}{(1 - \gamma)^2} \cdot \gamma .$$

*In addition, it holds for all $t \geq 0$ that*

$$\left\| \overline{Q}_\tau^{(t)} - Q_\tau^\star \right\|_\infty \leq \gamma \Omega_3^{(t)} + \gamma \Omega_4^{(t)} , \tag{51}$$

$$\left\| \log \overline{\pi}^{(t)} - \log \pi_\tau^\star \right\|_\infty \leq \frac{2}{\tau} \Omega_3^{(t)} . \tag{52}$$

*Proof.* See Appendix F.1. □

Let $\rho(\eta)$ denote the spectral norm of $\boldsymbol{A}(\eta)$. As $\boldsymbol{\Omega}^{(t)} \geq 0$, it is immediate from (49) that

$$\left\| \boldsymbol{\Omega}^{(t)} \right\|_2 \leq \rho(\eta)^t \left\| \boldsymbol{\Omega}^{(0)} \right\|_2 ,$$

and therefore we have

$$\left\| \overline{Q}_\tau^{(t)} - Q_\tau^\star \right\|_\infty \leq 2\gamma \left\| \boldsymbol{\Omega}^{(t)} \right\|_\infty \leq 2\gamma \rho(\eta)^t \left\| \boldsymbol{\Omega}^{(0)} \right\|_2 ,$$

and

$$\left\| \log \overline{\pi}^{(t)} - \log \pi_\tau^\star \right\|_\infty \leq \frac{2}{\tau} \left\| \boldsymbol{\Omega}^{(t)} \right\|_\infty \leq \frac{2}{\tau} \rho(\eta)^t \left\| \boldsymbol{\Omega}^{(0)} \right\|_2 .$$

It remains to bound the spectral radius $\rho(\eta)$, which is achieved by the following lemma.

**Lemma D.3** (Bounding the spectral norm of $\boldsymbol{A}(\eta)$). *Let*

$$\zeta := \frac{(1-\gamma)(1-\sigma)^2 \tau}{8 \left( \tau S_0 \sigma^2 + 10 M c \sigma^2 / (1-\gamma) + (1-\sigma)^2 \tau^2 / 16 \right)} , \tag{53}$$

*where $S_0 := M\sqrt{N} \left( 2 + \sqrt{2N} + \frac{M\sqrt{N}}{\tau} \right)$, $c := MN/(1-\gamma)$. For any $N \in \mathbb{N}_+, \tau > 0, \gamma \in (0,1)$, if*

$$0 < \eta \leq \eta_0 := \min \left\{ \frac{1-\gamma}{\tau}, \zeta \right\} , \tag{54}$$

*then we have*

$$\rho(\eta) \leq \max \left\{ \frac{3+\sigma}{4}, \frac{1 + (1-\alpha)\gamma + \alpha}{2} \right\} < 1 . \tag{55}$$

*Proof.* See Appendix F.2. □

### D.2 Analysis of entropy-regularized FedNPG with inexact policy evaluation

We define the collection of *inexact* Q-function estimates as

$$\boldsymbol{q}_\tau^{(t)} := \left( q_{\tau,1}^{\pi_1^{(t)}}, \cdots, q_{\tau,N}^{\pi_N^{(t)}} \right)^\top ,$$

and then the update rule ($U_T$) should be understood as

$$\boldsymbol{T}^{(t+1)}(s,a) = \boldsymbol{W} \left( \boldsymbol{T}^{(t)}(s,a) + \boldsymbol{q}_\tau^{(t+1)}(s,a) - \boldsymbol{q}_\tau^{(t)}(s,a) \right) \tag{56}$$

in the inexact setting. For notational simplicity, we define $e_n \in \mathbb{R}$ as

$$e_n := \max_{t \in [T]} \left\| Q_{\tau,n}^{\pi_n^{(t)}} - q_{\tau,n}^{\pi_n^{(t)}} \right\|_\infty , \quad n \in [N] , \tag{57}$$

and let $\boldsymbol{e} = (e_1, \cdots, e_n)^\top$. Define $\widehat{q}_\tau^{(t)}$, the approximation of $\widehat{Q}_\tau^{(t)}$ as

$$\widehat{q}_\tau^{(t)} := \frac{1}{N} \sum_{n=1}^N q_{\tau,n}^{\pi_n^{(t)}} . \tag{58}$$

With slight abuse of notation, we adapt the auxiliary sequence $\{\overline{\xi}^{(t)}\}_{t=0,\cdots}$ to the inexact updates as

$$\overline{\xi}^{(0)}(s,a) := \left\| \exp\left( Q_\tau^\star(s,\cdot)/\tau \right) \right\|_1 \cdot \overline{\pi}^{(0)}(a|s) , \tag{59a}$$

$$\overline{\xi}^{(t+1)}(s,a) := \left[ \overline{\xi}^{(t)}(s,a) \right]^\alpha \exp\left( (1-\alpha)\frac{\widehat{q}_\tau^{(t)}(s,a)}{\tau} \right) , \quad \forall (s,a) \in \mathcal{S} \times \mathcal{A}, \ t \geq 0 . \tag{59b}$$

In addition, we define

$$\Omega_1^{(t)} := \left\| u^{(t)} \right\|_\infty , \tag{60a}$$

$$\Omega_2^{(t)} := \left\| v^{(t)} \right\|_\infty , \tag{60b}$$

$$\Omega_3^{(t)} := \left\| Q_\tau^\star - \tau \log \overline{\xi}^{(t)} \right\|_\infty , \tag{60c}$$

$$\Omega_4^{(t)} := \max \left\{ 0, -\min_{s,a} \left( \overline{q}_\tau^{(t)}(s,a) - \tau \log \overline{\xi}^{(t)}(s,a) \right) \right\} , \tag{60d}$$

where

$$u^{(t)}(s,a) := \left\| \log \boldsymbol{\xi}^{(t)}(s,a) - \log \overline{\xi}^{(t)}(s,a) \mathbf{1}_N \right\|_2 , \tag{61}$$

$$v^{(t)}(s,a) := \left\| \boldsymbol{T}^{(t)}(s,a) - \widehat{q}_\tau^{(t)}(s,a) \mathbf{1}_N \right\|_2 . \tag{62}$$

We let $\boldsymbol{\Omega}^{(t)}$ be

$$\boldsymbol{\Omega}^{(t)} := \left( \Omega_1^{(t)}, \Omega_2^{(t)}, \Omega_3^{(t)}, \Omega_4^{(t)} \right)^\top . \tag{63}$$

With the above preparation, we are ready to state the inexact convergence guarantee of Algorithm 2 in Theorem D.4 below, which is the formal version of Theorem 3.8.

**Theorem D.4.** *Suppose that $q_{\tau,n}^{\pi_n^{(t)}}$ are used in replace of $Q_{\tau,n}^{\pi_n^{(t)}}$ in Algorithm 2. For any $N \in \mathbb{N}_+, \tau > 0, \gamma \in (0,1)$, there exists $\eta_0 > 0$ which depends only on $N, \gamma, \tau, \sigma, |\mathcal{A}|$, such that if $0 < \eta \leq \eta_0$ and $1 - \sigma > 0$, we have*

$$\left\| \overline{Q}_\tau^{(t)} - Q_\tau^\star \right\|_\infty \leq 2\gamma \left( \rho(\eta)^t \left\| \boldsymbol{\Omega}^{(0)} \right\|_2 + C_2 \max_{n \in [N], t \in [T]} \left\| Q_{\tau,n}^{\pi_n^{(t)}} - q_{\tau,n}^{\pi_n^{(t)}} \right\|_\infty \right) , \tag{64}$$

$$\left\| \log \pi_\tau^\star - \log \overline{\pi}^{(t)} \right\|_\infty \leq \frac{2}{\tau} \left( \rho(\eta)^t \left\| \boldsymbol{\Omega}^{(0)} \right\|_2 + C_2 \max_{n \in [N], t \in [T]} \left\| Q_{\tau,n}^{\pi_n^{(t)}} - q_{\tau,n}^{\pi_n^{(t)}} \right\|_\infty \right) . \tag{65}$$

*Moreover, the consensus errors satisfy:*

$$\forall n \in [N] : \quad \left\| \log \pi_n^{(t)} - \log \overline{\pi}^{(t)} \right\|_\infty \leq 2 \left( \rho(\eta)^t \left\| \boldsymbol{\Omega}^{(0)} \right\|_2 + C_2 \max_{n \in [N], t \in [T]} \left\| Q_{\tau,n}^{\pi_n^{(t)}} - q_{\tau,n}^{\pi_n^{(t)}} \right\|_\infty \right) , \tag{66}$$

*where $\rho(\eta) \leq \max\{1 - \frac{\tau\eta}{2}, \frac{3+\sigma}{4}\} < 1$ is the same as in Theorem D.1, and $C_2 := \frac{\sigma\sqrt{N}(2(1-\gamma) + M\sqrt{N}\eta) + 2\gamma^2 + \eta\tau}{(1-\gamma)(1-\rho(\eta))}$.*

From Theorem D.4, we can conclude that if

$$\max_{n \in [N], t \in [T]} \left\| Q_{\tau,n}^{\pi_n^{(t)}} - q_{\tau,n}^{\pi_n^{(t)}} \right\|_\infty \leq \frac{(1-\gamma)(1-\rho(\eta))\varepsilon}{2\gamma \left( \sigma\sqrt{N}(2(1-\gamma) + M\sqrt{N}\eta) + 2\gamma^2 + \eta\tau \right)} , \tag{67}$$

then inexact entropy-regularized FedNPG could still achieve $2\varepsilon$-accuracy (i.e. $\left\| \overline{Q}_\tau^{(t)} - Q_\tau^\star \right\|_\infty \leq 2\varepsilon$) within $\max \left\{ \frac{2}{\tau\eta}, \frac{4}{1-\sigma} \right\} \log \frac{2\gamma \left\| \boldsymbol{\Omega}^{(0)} \right\|_2}{\varepsilon}$ iterations.

*Remark* D.5. When $\eta = \eta_0$ (cf. (54) and (53)) and $\tau \leq 1$, the RHS of (67) is of the order

$$\mathcal{O} \left( \frac{(1-\gamma)\tau\eta_0\varepsilon}{\gamma(\gamma^2 + \sigma\sqrt{N}(1-\gamma))} \right) = \mathcal{O} \left( \frac{(1-\gamma)^8\tau^2(1-\sigma)^2\varepsilon}{\gamma(\gamma^2 + \sigma\sqrt{N}(1-\gamma))(\gamma^2 N\sigma^2 + (1-\sigma)^2\tau^2(1-\gamma)^6)} \right) ,$$

which can be translated into a crude sample complexity bound when using fresh samples to estimate the soft Q-functions in each iteration.

The rest of this section outlines the proof of Theorem D.4. We first state a key lemma that tracks the error recursion of Algorithm 2 with inexact policy evaluation, which is a modified version of Lemma D.2.

**Lemma D.6.** *The following linear system holds for all $t \geq 0$:*

$$\boldsymbol{\Omega}^{(t+1)} \leq \boldsymbol{A}(\eta)\boldsymbol{\Omega}^{(t)} + \underbrace{\begin{pmatrix} 0 \\ \sigma\sqrt{N}\left(2 + \frac{M\sqrt{N}\eta}{1-\gamma}\right) \\ \frac{\eta\tau}{1-\gamma} \\ \frac{2\gamma^2}{1-\gamma} \end{pmatrix}}_{=:\boldsymbol{b}(\eta)} \|\boldsymbol{e}\|_\infty, \tag{68}$$

*where $\boldsymbol{A}(\eta)$ is provided in Lemma D.2. In addition, it holds for all $t \geq 0$ that*

$$\left\|\overline{Q}_\tau^{(t)} - Q_\tau^\star\right\|_\infty \leq \gamma\Omega_3^{(t)} + \gamma\Omega_4^{(t)}, \tag{69}$$

$$\left\|\log\overline{\pi}^{(t)} - \log\pi_\tau^\star\right\|_\infty \leq \frac{2}{\tau}\Omega_3^{(t)}. \tag{70}$$

*Proof.* See Appendix F.3. $\qquad\square$

By (68), we have

$$\forall t \in N_+ : \quad \boldsymbol{\Omega}^{(t)} \leq \boldsymbol{A}(\eta)^t\boldsymbol{\Omega}^{(0)} + \sum_{s=1}^t \boldsymbol{A}(\eta)^{t-s}\boldsymbol{b}(\eta),$$

which gives

$$\left\|\boldsymbol{\Omega}^{(t)}\right\|_2 \leq \rho(\eta)^t\left\|\boldsymbol{\Omega}^{(0)}\right\|_2 + \sum_{s=1}^t \rho(\eta)^{t-s}\|\boldsymbol{b}(\eta)\|_2\|\boldsymbol{e}\|_\infty$$

$$\leq \rho(\eta)^t\left\|\boldsymbol{\Omega}^{(0)}\right\|_2 + \frac{\sigma\sqrt{N}(2(1-\gamma) + M\sqrt{N}\eta) + 2\gamma^2 + \eta\tau}{(1-\gamma)(1-\rho(\eta))}\|\boldsymbol{e}\|_\infty. \tag{71}$$

Here, (71) follows from $\|\boldsymbol{b}(\eta)\|_2 \leq \|\boldsymbol{b}(\eta)\|_1 = \frac{\sigma\sqrt{N}(2(1-\gamma)+M\sqrt{N}\eta)+2\gamma^2+\eta\tau}{1-\gamma}\|\boldsymbol{e}\|_\infty$ and $\sum_{s=1}^t \rho(\eta)^{t-s} \leq 1/(1-\rho(\eta))$. Recall that the bound on $\rho(\eta)$ has already been established in Lemma D.3. Therefore we complete the proof of Theorem D.4 by combining the above inequality with (69) and (70) in a similar fashion as before. We omit further details for conciseness.

### D.3 Analysis of FedNPG with exact policy evaluation

We state the formal version of Theorem 3.3 below.

**Theorem D.7.** *Suppose all $\pi_n^{(0)}$ in Algorithm 1 are initialized as uniform distribution. When*

$$0 < \eta \leq \eta_1 := \frac{(1-\sigma)^2(1-\gamma)^3}{8(1+\gamma)\gamma\sqrt{N}\sigma^2},$$

*we have*

$$\frac{1}{T}\sum_{t=0}^{T-1}\left(V^\star(\rho) - V^{\overline{\pi}^{(t)}}(\rho)\right) \leq \frac{V^\star(d_\rho^{\pi^\star})}{(1-\gamma)T} + \frac{\log|\mathcal{A}|}{\eta T} + \frac{8(1+\gamma)^2\gamma^2N\sigma^2}{(1-\gamma)^9(1-\sigma)^2}\eta^2 \tag{72}$$

*for any fixed state distribution $\rho$. Furthermore, we have*

$$\forall n \in [N] : \quad \left\|\log\pi_n^{(t)} - \log\overline{\pi}^{(t)}\right\|_\infty \leq \frac{32N\sigma}{3(1-\gamma)^4(1-\sigma)}\eta. \tag{73}$$

The rest of this section is dedicated to prove Theorem D.7. Similar to (37), we denote the $Q$-functions of $\overline{\pi}^{(t)}$ by $\overline{Q}^{(t)}$:

$$\overline{Q}^{(t)} := \begin{pmatrix} Q_1^{\overline{\pi}^{(t)}} \\ \vdots \\ Q_N^{\overline{\pi}^{(t)}} \end{pmatrix}. \tag{74}$$

In addition, similar to (38), we define $\widehat{Q}^{(t)}, \overline{Q}^{(t)} \in \mathbb{R}^{|\mathcal{S}||\mathcal{A}|}$ and $\overline{V}^{(t)} \in \mathbb{R}^{|\mathcal{S}|}$ as follows

$$\widehat{Q}^{(t)} := \frac{1}{N} \sum_{n=1}^{N} Q_n^{\pi_n^{(t)}} , \tag{75a}$$

$$\overline{Q}^{(t)} := Q^{\overline{\pi}^{(t)}} = \frac{1}{N} \sum_{n=1}^{N} Q_n^{\overline{\pi}^{(t)}} . \tag{75b}$$

$$\overline{V}^{(t)} := V^{\overline{\pi}^{(t)}} = \frac{1}{N} \sum_{n=1}^{N} V_n^{\overline{\pi}^{(t)}} . \tag{75c}$$

Following the same strategy in the analysis of entropy-regularized FedNPG, we introduce the auxiliary sequence $\{\boldsymbol{\xi}^{(t)} = (\xi_1^{(t)}, \cdots, \xi_N^{(t)})^\top \in \mathbb{R}^{N \times |\mathcal{S}||\mathcal{A}|}\}$ recursively:

$$\boldsymbol{\xi}^{(0)}(s, a) := \frac{1}{\left\| \exp\left( \frac{1}{N} \sum_{n=1}^{N} \log \pi_n^{(0)}(\cdot | s) \right) \right\|_1} \cdot \boldsymbol{\pi}^{(0)}(a|s) , \tag{76a}$$

$$\log \boldsymbol{\xi}^{(t+1)}(s, a) = \boldsymbol{W} \left( \log \boldsymbol{\xi}^{(t)}(s, a) + \frac{\eta}{1 - \gamma} \boldsymbol{T}^{(t)}(s, a) \right) , \tag{76b}$$

as well as the averaged auxiliary sequence $\{\overline{\xi}^{(t)} \in \mathbb{R}^{|\mathcal{S}||\mathcal{A}|}\}$:

$$\overline{\xi}^{(0)}(s, a) := \overline{\pi}^{(0)}(a|s) , \tag{77a}$$

$$\log \overline{\xi}^{(t+1)}(s, a) := \log \overline{\xi}^{(t)}(s, a) + \frac{\eta}{1 - \gamma} \widehat{Q}^{(t)}(s, a) , \quad \forall (s, a) \in \mathcal{S} \times \mathcal{A}, \ t \geq 0 . \tag{77b}$$

As usual, we collect the consensus errors in a vector $\boldsymbol{\Omega}^{(t)} = (\|u^{(t)}\|_\infty, \|v^{(t)}\|_\infty)^\top$, where $u^{(t)}, v^{(t)} \in \mathbb{R}^{|\mathcal{S}||\mathcal{A}|}$ are defined as:

$$u^{(t)}(s, a) := \left\| \log \boldsymbol{\xi}^{(t)}(s, a) - \log \overline{\xi}^{(t)}(s, a) \mathbf{1}_N \right\|_2 , \tag{78}$$

$$v^{(t)}(s, a) := \left\| \boldsymbol{T}^{(t)}(s, a) - \widehat{Q}^{(t)}(s, a) \mathbf{1}_N \right\|_2 . \tag{79}$$

**Step 1: establishing the error recursion.** The next key lemma establishes the error recursion of Algorithm 1.

**Lemma D.8.** *The updates of FedNPG satisfy*

$$\boldsymbol{\Omega}^{(t+1)} \leq \underbrace{\begin{pmatrix} \sigma & \frac{\eta}{1-\gamma}\sigma \\ J\sigma & \sigma\left(1 + \frac{(1+\gamma)\gamma\sqrt{N}\eta}{(1-\gamma)^3}\sigma\right) \end{pmatrix}}_{=:\boldsymbol{B}(\eta)} \boldsymbol{\Omega}^{(t)} + \underbrace{\begin{pmatrix} 0 \\ \frac{(1+\gamma)\gamma N\sigma}{(1-\gamma)^4}\eta \end{pmatrix}}_{=:\boldsymbol{d}(\eta)} \tag{80}$$

*for all $t \geq 0$, where*

$$J := \frac{2(1 + \gamma)\gamma}{(1 - \gamma)^2} \sqrt{N} . \tag{81}$$

*In addition, we have*

$$\phi^{(t+1)}(\eta) \leq \phi^{(t)}(\eta) + \frac{2(1+\gamma)\gamma}{(1-\gamma)^4}\eta\|u^{(t)}\|_\infty - \eta\left(V^\star(\rho) - \overline{V}^{(t)}(\rho)\right) , \tag{82}$$

*where*

$$\phi^{(t)}(\eta) := \mathbb{E}_{s \sim d_\rho^{\pi^\star}}\left[ \mathsf{KL}\left(\pi^\star(\cdot|s) \,\|\, \overline{\pi}^{(t)}(\cdot|s)\right) \right] - \frac{\eta}{1-\gamma}\overline{V}^{(t)}(d_\rho^{\pi^\star}) , \quad \forall t \geq 0 . \tag{83}$$

*Moreover, when $\eta \leq \eta_1$, we have*

$$\forall n \in [N]: \quad \left\| \log \pi_n^{(t)} - \log \overline{\pi}^{(t)} \right\|_\infty \leq 2\left(\frac{3}{8}\sigma + \frac{5}{8}\right)^t \left\| \boldsymbol{\Omega}^{(0)} \right\|_2 + \frac{32N\sigma}{3(1-\gamma)^4(1-\sigma)}\eta . \tag{84}$$

*Proof.* See Appendix F.4. $\qquad\qquad\square$

Note that when all $\pi_n^{(0)}$ in Algorithm 1 are initialized as uniform distribution, $\boldsymbol{\Omega}^{(0)} = \boldsymbol{0}$ and (84) indicates (73) in Theorem D.7.

**Step 2: bounding the value functions.** Let $\boldsymbol{p} \in \mathbb{R}^2$ be defined as:

$$\boldsymbol{p}(\eta) = \begin{pmatrix} p_1(\eta) \\ p_2(\eta) \end{pmatrix} := \frac{2(1+\gamma)\gamma}{(1-\gamma)^4} \begin{pmatrix} \frac{\sigma(1-\gamma)\left(1-\sigma-(1+\gamma)\gamma\sqrt{N}\sigma\eta/(1-\gamma)^3\right)\eta}{(1-\gamma)\left(1-\sigma-(1+\gamma)\gamma\sqrt{N}\sigma^2\eta/(1-\gamma)^3\right)(1-\sigma)-J\sigma^2\eta} \\ \frac{\sigma\eta^2}{(1-\gamma)\left(1-\sigma-(1+\gamma)\gamma\sqrt{N}\sigma^2\eta/(1-\gamma)^3\right)(1-\sigma)-J\sigma^2\eta} \end{pmatrix} ; \quad (85)$$

the rationale for this choice will be made clear momentarily. We define the following Lyapunov function

$$\Phi^{(t)}(\eta) = \phi^{(t)}(\eta) + \boldsymbol{p}(\eta)^\top \boldsymbol{\Omega}^{(t)}, \quad \forall t \geq 0, \quad (86)$$

which satisfies

$$
\begin{aligned}
\Phi^{(t+1)}(\eta) &= \phi^{(t+1)}(\eta) + \boldsymbol{p}(\eta)^\top \boldsymbol{\Omega}^{(t+1)} \\
&\leq \phi^{(t)}(\eta) + \frac{2(1+\gamma)\gamma}{(1-\gamma)^4}\eta\|u^{(t)}\|_\infty - \eta\left(V^\star(\rho) - \overline{V}^{(t)}(\rho)\right) + \boldsymbol{p}(\eta)^\top\left(\boldsymbol{B}(\eta)\boldsymbol{\Omega}^{(t)} + \boldsymbol{d}(\eta)\right) \\
&= \Phi^{(t)}(\eta) + \left[\boldsymbol{p}(\eta)^\top\left(\boldsymbol{B}(\eta) - \boldsymbol{I}\right) + \left(\frac{2(1+\gamma)\gamma}{(1-\gamma)^4}\eta, 0\right)\right]\boldsymbol{\Omega}^{(t)} - \eta\left(V^\star(\rho) - \overline{V}^{(t)}(\rho)\right) \\
&\quad + p_2(\eta)\frac{(1+\gamma)\gamma N\sigma}{(1-\gamma)^4}\eta.
\end{aligned}
\quad (87)
$$

Here, the second inequality follows from (82). One can verify that the second term vanishes due to the choice of $\boldsymbol{p}(\eta)$:

$$\boldsymbol{p}(\eta)^\top\left(\boldsymbol{B}(\eta) - \boldsymbol{I}\right) + \left(\frac{2(1+\gamma)\gamma}{(1-\gamma)^4}\eta, 0\right) = (0, 0). \quad (88)$$

Therefore, we conclude that

$$V^\star(\rho) - \overline{V}^{(t)}(\rho) \leq \frac{\Phi^{(t)}(\eta) - \Phi^{(t+1)}(\eta)}{\eta} + p_2(\eta)\frac{(1+\gamma)\gamma N\sigma}{(1-\gamma)^4}.$$

Averaging over $t = 0, \cdots, T-1$,

$$
\begin{aligned}
&\frac{1}{T}\sum_{t=0}^{T-1}\left(V^\star(\rho) - \overline{V}^{(t)}(\rho)\right) \\
&\leq \frac{\Phi^{(0)}(\eta) - \Phi^{(T)}(\eta)}{\eta T} + \frac{2(1+\gamma)^2\gamma^2}{(1-\gamma)^8} \cdot \frac{N\sigma^2\eta^2}{(1-\gamma)(1-\sigma-(1+\gamma)\gamma\sqrt{N}\sigma^2\eta/(1-\gamma)^3)(1-\sigma) - \sigma^2 J\eta}.
\end{aligned}
\quad (89)
$$

**Step 3: simplifying the expression.** We first upper bound the first term in the RHS of (89). Assuming uniform initialization for all $\pi_n^{(0)}$ in Algorithm 1, we have $\|u^{(0)}\|_\infty = \|v^{(0)}\|_\infty = 0$, and

$$\mathbb{E}_{s\sim d_\rho^{\pi^\star}}\left[\mathsf{KL}\left(\pi^\star(\cdot|s) \,\|\, \overline{\pi}^{(0)}(\cdot|s)\right)\right] \leq \log|\mathcal{A}|.$$

Therefore, putting together relations (86) and (221) we have

$$\frac{\Phi^{(0)}(\eta) - \Phi^{(T)}(\eta)}{\eta T} \leq \frac{\log|\mathcal{A}|}{T\eta} + \frac{1}{T}\left(\boldsymbol{p}(\eta)^\top\boldsymbol{\Omega}^{(0)}/\eta + \frac{V^\star(d_\rho^{\pi^\star})}{1-\gamma}\right) = \frac{\log|\mathcal{A}|}{T\eta} + \frac{V^\star(d_\rho^{\pi^\star})}{T(1-\gamma)}, \quad (90)$$

To continue, we upper bound the second term in the RHS of (89). Note that

$$\eta \leq \eta_1 \leq \frac{(1-\sigma)(1-\gamma)^3}{2(1+\gamma)\gamma\sqrt{N}\sigma^2},$$

which gives

$$\frac{(1+\gamma)\gamma\sqrt{N}\sigma^2}{(1-\gamma)^3}\eta \leq \frac{1-\sigma}{2}. \quad (91)$$

Thus we have

$$
\begin{aligned}
&(1-\gamma)(1-\sigma-(1+\gamma)\gamma\sqrt{N}\sigma^2\eta/(1-\gamma)^3)(1-\sigma)-J\sigma^2\eta\\
&\geq (1-\gamma)(1-\sigma)^2/2 - J\sigma^2\eta_1\\
&\geq (1-\gamma)(1-\sigma)^2/4\,,
\end{aligned}
\tag{92}
$$

where the first inequality follows from (91) and the second inequality follows from the definition of $\eta_1$ and $J$. By (92), we deduce

$$
\frac{2(1+\gamma)^2\gamma^2}{(1-\gamma)^8}\cdot\frac{N\sigma^2\eta^2}{(1-\gamma)(1-\sigma-(1+\gamma)\gamma\sqrt{N}\sigma^2\eta/(1-\gamma)^3)(1-\sigma)-J\sigma^2\eta}\leq\frac{8(1+\gamma)^2\gamma^2 N\sigma^2}{(1-\gamma)^9(1-\sigma)^2}\eta^2\,,
\tag{93}
$$

and our advertised bound (72) thus follows from plugging (90) and (93) into (89).

### D.4  Analysis of FedNPG with inexact policy evaluation

We state the formal version of Theorem 3.5 below.

**Theorem D.9.** *Suppose that $q_n^{\pi_n^{(t)}}$ are used in replace of $Q_n^{\pi_n^{(t)}}$ in Algorithm 1. Suppose all $\pi_n^{(0)}$ in Algorithm 1 set to uniform distribution. Let*

$$
0 < \eta \leq \eta_1 := \frac{(1-\sigma)^2(1-\gamma)^3}{8(1+\gamma)\gamma\sqrt{N}\sigma^2}\,,
$$

*we have*

$$
\begin{aligned}
&\frac{1}{T}\sum_{t=0}^{T-1}\left(V^\star(\rho)-V^{\overline{\pi}^{(t)}}(\rho)\right)\\
&\leq \frac{V^\star(d_\rho^{\pi^\star})}{(1-\gamma)T}+\frac{\log|\mathcal{A}|}{\eta T}+\frac{8(1+\gamma)^2\gamma^2 N\sigma^2}{(1-\gamma)^9(1-\sigma)^2}\eta^2\\
&\quad+\left[\frac{8(1+\gamma)\gamma}{(1-\gamma)^5(1-\sigma)^2}\sqrt{N}\sigma\eta\left(\frac{(1+\gamma)\gamma\eta\sqrt{N}}{(1-\gamma)^3}+2\right)+\frac{2}{(1-\gamma)^2}\right]\max_{n\in[N],t\in[T]}\left\|Q_n^{\pi_n^{(t)}}-q_n^{\pi_n^{(t)}}\right\|_\infty
\end{aligned}
$$

*for any fixed state distribution $\rho$.*

*Furthermore, we have*

$$
\forall n\in[N]:\quad\left\|\log\pi_n^{(t)}-\log\overline{\pi}^{(t)}\right\|_\infty\leq\frac{32}{3(1-\sigma)}\left(\frac{N\sigma}{(1-\gamma)^4}\eta+\sqrt{N}\sigma\left(\frac{\eta\sqrt{N}}{(1-\gamma)^3}+1\right)\max_{n\in[N],t\in[T]}\left\|Q_n^{\pi_n^{(t)}}-q_n^{\pi_n^{(t)}}\right\|_\infty\right).
\tag{94}
$$

We next outline the proof of Theorem D.9. With slight abuse of notation, we again define $e_n\in\mathbb{R}$ as

$$
e_n := \max_{t\in[T]}\left\|Q_n^{\pi_n^{(t)}}-q_n^{\pi_n^{(t)}}\right\|_\infty,\quad n\in[N]\,,
\tag{95}
$$

and let $\boldsymbol{e}=(e_1,\cdots,e_n)^\top$. We define the collection of *inexact* Q-function estimates as

$$
\boldsymbol{q}^{(t)} := \left(q_1^{\pi_1^{(t)}},\cdots,q_N^{\pi_N^{(t)}}\right)^\top,
$$

and then the update rule (16) should be understood as

$$
\boldsymbol{T}^{(t+1)}(s,a)=\boldsymbol{W}\left(\boldsymbol{T}^{(t)}(s,a)+\boldsymbol{q}^{(t+1)}(s,a)-\boldsymbol{q}^{(t)}(s,a)\right)
\tag{96}
$$

in the inexact setting. Define $\widehat{q}^{(t)}$, the approximation of $\widehat{Q}^{(t)}$ as

$$
\widehat{q}^{(t)} := \frac{1}{N}\sum_{n=1}^{N}q_n^{\pi_n^{(t)}},
\tag{97}
$$

we adapt the averaged auxiliary sequence $\{\overline{\xi}^{(t)} \in \mathbb{R}^{|\mathcal{S}||\mathcal{A}|}\}$ to the inexact updates as follows:

$$\overline{\xi}^{(0)}(s,a) := \overline{\pi}^{(0)}(a|s)\,, \tag{98a}$$

$$\overline{\xi}^{(t+1)}(s,a) := \overline{\xi}^{(t)}(s,a)\exp\left(\frac{\eta}{1-\gamma}\widehat{q}^{(t)}(s,a)\right)\,, \quad \forall(s,a) \in \mathcal{S}\times\mathcal{A},\ t \geq 0\,. \tag{98b}$$

As usual, we define the consensus error vector as $\mathbf{\Omega}^{(t)} = (\|u^{(t)}\|_\infty, \|v^{(t)}\|_\infty)^\top$, where $u^{(t)}, v^{(t)} \in \mathbb{R}^{|\mathcal{S}||\mathcal{A}|}$ are given by

$$u^{(t)}(s,a) := \left\|\log\boldsymbol{\xi}^{(t)}(s,a) - \log\overline{\xi}^{(t)}(s,a)\mathbf{1}_N\right\|_2\,, \tag{99}$$

$$v^{(t)}(s,a) := \left\|\boldsymbol{T}^{(t)}(s,a) - \widehat{q}^{(t)}(s,a)\mathbf{1}_N\right\|_2\,. \tag{100}$$

The following lemma characterizes the dynamics of the error vector $\mathbf{\Omega}^{(t)}$, perturbed by additional approximation error.

**Lemma D.10.** *The updates of inexact FedNPG satisfy*

$$\mathbf{\Omega}^{(t+1)} \leq \boldsymbol{B}(\eta)\mathbf{\Omega}^{(t)} + \boldsymbol{d}(\eta) + \underbrace{\left(\begin{array}{c} 0 \\ \sqrt{N}\sigma\left(\frac{(1+\gamma)\gamma\eta\sqrt{N}}{(1-\gamma)^3} + 2\right)\end{array}\right)\|\boldsymbol{e}\|_\infty}_{=:\boldsymbol{c}(\eta)}\,. \tag{101}$$

*In addition, we have*

$$\phi^{(t+1)}(\eta) \leq \phi^{(t)}(\eta) + \frac{2(1+\gamma)\gamma}{(1-\gamma)^4}\eta\left\|u^{(t)}\right\|_\infty + \frac{2\eta}{(1-\gamma)^2}\|\boldsymbol{e}\|_\infty - \eta\left(V^\star(\rho) - \overline{V}^{(t)}(\rho)\right)\,, \tag{102}$$

*where $\phi^{(t)}(\eta)$ is defined in (83).*

*Moreover, when $\eta \leq \eta_1$, we have*

$$\forall n \in [N]: \quad \left\|\log\pi_n^{(t)} - \log\overline{\pi}^{(t)}\right\|_\infty \leq 2\left(\frac{3}{8}\sigma + \frac{5}{8}\right)^t\left\|\mathbf{\Omega}^{(0)}\right\|_2 + \frac{32}{3(1-\sigma)}\left(\frac{N\sigma}{(1-\gamma)^4}\eta + \sqrt{N}\sigma\left(\frac{\eta\sqrt{N}}{(1-\gamma)^3} + 1\right)\|\boldsymbol{e}\|_\infty\right)\,. \tag{103}$$

*Proof.* See Appendix F.5. $\qquad\square$

Similar to (87), we can recursively bound $\Phi^{(t)}(\eta)$ (defined in (86)) as

$$\begin{aligned}
\Phi^{(t+1)}(\eta) &= \phi^{(t+1)}(\eta) + \boldsymbol{p}(\eta)^\top\mathbf{\Omega}^{(t+1)} \\
&\overset{(102)}{\leq} \phi^{(t)}(\eta) + \frac{2(1+\gamma)\gamma}{(1-\gamma)^4}\eta\left\|u^{(t)}\right\|_\infty + \frac{2\eta}{(1-\gamma)^2}\|\boldsymbol{e}\|_\infty - \eta\left(V^\star(\rho) - \overline{V}^{(t)}(\rho)\right) \\
&\quad + \boldsymbol{p}(\eta)^\top\left(\boldsymbol{B}(\eta)\mathbf{\Omega}^{(t)} + \boldsymbol{d}(\eta) + \boldsymbol{c}(\eta)\right) \\
&= \Phi^{(t)}(\eta) + \underbrace{\left[\boldsymbol{p}(\eta)^\top(\boldsymbol{B}(\eta) - \boldsymbol{I}) + \left(\frac{2(1+\gamma)\gamma}{(1-\gamma)^4}\eta, 0\right)\right]}_{=(0,0)\text{ via }(88)}\mathbf{\Omega}^{(t)} - \eta\left(V^\star(\rho) - \overline{V}^{(t)}(\rho)\right) \\
&\quad + p_2(\eta)\frac{(1+\gamma)\gamma N\sigma}{(1-\gamma)^4}\eta + \left[p_2(\eta)\sqrt{N}\sigma\left(\frac{(1+\gamma)\gamma\eta\sqrt{N}}{(1-\gamma)^3} + 2\right) + \frac{2\eta}{(1-\gamma)^2}\right]\|\boldsymbol{e}\|_\infty\,.
\end{aligned} \tag{104}$$

From the above expression we know that

$$V^\star(\rho) - \overline{V}^{(t)}(\rho) \leq \frac{\Phi^{(t)}(\eta) - \Phi^{(t+1)}(\eta)}{\eta} + p_2(\eta)\frac{(1+\gamma)\gamma N\sigma}{(1-\gamma)^4} + \left[p_2(\eta)\sqrt{N}\sigma\left(\frac{(1+\gamma)\gamma\sqrt{N}}{(1-\gamma)^3} + \frac{2}{\eta}\right) + \frac{2}{(1-\gamma)^2}\right]\|\boldsymbol{e}\|_\infty\,,$$

which gives

$$\frac{1}{T}\sum_{t=0}^{T-1}\left(V^\star(\rho)-\overline{V}^{(t)}(\rho)\right) \le \frac{\Phi^{(0)}(\eta)-\Phi^{(T)}(\eta)}{\eta T} + p_2(\eta)\frac{(1+\gamma)\gamma N\sigma}{(1-\gamma)^4}$$
$$+ \left[p_2(\eta)\sqrt{N}\sigma\left(\frac{(1+\gamma)\gamma\sqrt{N}}{(1-\gamma)^3}+\frac{2}{\eta}\right)+\frac{2}{(1-\gamma)^2}\right]\|\boldsymbol{e}\|_\infty \tag{105}$$

via telescoping. Combining the above expression with (90), (92) and (93), we have

$$\frac{1}{T}\sum_{t=0}^{T-1}\left(V^\star(\rho)-\overline{V}^{(t)}(\rho)\right) \le \frac{\log|\mathcal{A}|}{T\eta}+\frac{V^\star(d_\rho^{\pi^\star})}{T(1-\gamma)}+\frac{8(1+\gamma)^2\gamma^2 N\sigma}{(1-\gamma)^9(1-\sigma)^2}\eta^2$$
$$+ \left[\frac{8(1+\gamma)\gamma}{(1-\gamma)^5(1-\sigma)^2}\sqrt{N}\sigma\eta\left(\frac{(1+\gamma)\gamma\eta\sqrt{N}}{(1-\gamma)^3}+2\right)+\frac{2}{(1-\gamma)^2}\right]\|\boldsymbol{e}\|_\infty\,, \tag{106}$$

which establishes (94).

# E    Convergence analysis of FedNAC

Let $\pi^\star$ be an optimal policy and does not need to belong to the log-linear policy class. Fix a state distribution $\rho \in \Delta(\mathcal{S})$ and a state-action distribution $\nu$. To simplify the notation, we denote $d_\rho^{\pi^\star}$ as $d_\star$, $d^{f_{\bar{\boldsymbol{\xi}}^{(t)}}}$ as $d^{(t)}$, $\tilde{d}_n^{(t)}$ as $\tilde{d}_\nu^{f_{\xi_n^{(t)}}}$, and define $d_n^{(t)}$ and $\bar{d}_n^{(t)}$ analogously. We also let $Q_n^{(t)}$ denote $Q_n^{\xi_n^{(t)}}$. Define

$$\vartheta_\rho := \frac{1}{1-\gamma}\left\|\frac{d_\star}{\rho}\right\|_\infty \ge \frac{1}{1-\gamma} \tag{107}$$

and assume $\vartheta_\rho < \infty$.

We also introduce a weighted KL divergence given by

$$D_\star^{(t)} := \mathbb{E}_{s\sim d_\star}\left[\mathsf{KL}\big(\pi^\star(\cdot|s)\,\|\,\pi^{(t)}(\cdot|s)\big)\right]\,, \tag{108}$$

where $\mathsf{KL}(\cdot\,\|\,\cdot) : \mathbb{R}^{|\mathcal{A}|}\times\mathbb{R}^{|\mathcal{A}|}\to\mathbb{R}$ is the Kullback-Leibler (KL) divergence:

$$\forall f,g \in \mathbb{R}^{|\mathcal{A}|}: \quad \mathsf{KL}\big(f\,\|\,g\big) := \sum_{a\in\mathcal{A}}f(a)\log\left(\frac{f(a)}{g(a)}\right)\,. \tag{109}$$

Given a state distribution $\rho$ and an optimal policy $\pi^\star$, we define a state-action measure $\tilde{d}^\star$ as

$$\tilde{d}^\star(s,a) := d_\star(s)\cdot\mathrm{Unif}_\mathcal{A}(a) = \frac{d_\star(s)}{|\mathcal{A}|}. \tag{110}$$

The following theorem guarantees that for any fixed policy $\pi$ and state-action distribution $\nu \in \Delta(\mathcal{S}\times\mathcal{A})$, the Q-Sampler algorithm (cf. Algorithm 5) samples $(s,a)$ from $\tilde{d}_\nu^\pi$ and gives an unbiased estimate $\widehat{Q}^\pi(s,a)$ of $Q^\pi(s,a)$, whose proof can be found in [YDG+22, Lemma 4].

**Lemma E.1** (Lemma 4 in [YDG+22]). *Consider the output $(s_h,a_h)$ and $\widehat{Q}^\pi(s_h,a_h)$ of Algorithm 5. It follows that*

$$\mathbb{E}[h+1] = \frac{1}{1-\gamma}\,,$$
$$P(s_h=s,a_h=a) = \tilde{d}_\nu^\pi(s,a)\,,$$
$$\mathbb{E}\left[\widehat{Q}^\pi(s_h,a_h)|s_h,a_h\right] = Q^\pi(s_h,a_h)\,.$$

To present the convergence results of FedNAC, we further introduce the following notation, where $t \in \mathbb{N}$ represents the iteration step in FedNAC:

$$\hat{\boldsymbol{w}}^{(t)} := \frac{1}{N} \sum_{n=1}^{N} \boldsymbol{w}_n^{(t)}, \tag{111a}$$

$$\bar{\boldsymbol{\xi}}^{(t)} := \frac{1}{N} \sum_{n=1}^{N} \boldsymbol{\xi}_n^{(t)}, \tag{111b}$$

$$\bar{f}^{(t)} := f_{\bar{\xi}^{(t)}}, \tag{111c}$$

$$f_n^{(t)} := f_{\xi_n^{(t)}}, \tag{111d}$$

$$\boldsymbol{w}_{\star,n}^{(t)} \in \arg\min_{\boldsymbol{w}} \ell\left(\boldsymbol{w}, Q_n^{(t)}, \tilde{d}_n^{(t)}\right), \tag{111e}$$

$$\hat{\boldsymbol{w}}_\star^{(t)} := \frac{1}{N} \sum_{n=1}^{N} \boldsymbol{w}_{\star,n}^{(t)}. \tag{111f}$$

For convenience of narration, we introduce the following bounded statistical error assumption.

**Assumption E.2** (Bounded statistical error). For all $n \in [N]$, there exists $\varepsilon_{\text{stat}}^n > 0$ such that for all $t \in \mathbb{N}$ in Algorithm 3, we have

$$\mathbb{E}\left[\ell\left(\boldsymbol{w}_n^{(t)}, Q_n^{(t)}, \tilde{d}_n^{(t)}\right) - \ell\left(\boldsymbol{w}_{\star,n}^{(t)}, Q_n^{(t)}, \tilde{d}_n^{(t)}\right)\right] \leq \varepsilon_{\text{stat}}^n. \tag{112}$$

When solving the regression problem with sampling based approaches, we can expect $\varepsilon_{\text{stat}}^n = \mathcal{O}(1/K)$, where $K$ is the iteration number of Algorithm 4.

**Theorem E.3** (Convergence rate of Critic (Algorithm 4)). *For Algorithm 4, let $\boldsymbol{w}_0 = \boldsymbol{0}$ and $\beta = \frac{1}{2C_\phi}$. Then under Assumption 4.1, we have*

$$\mathbb{E}\left[\ell\left(\boldsymbol{w}_{out}, Q_\xi, \tilde{d}_\xi\right)\right] - \ell\left(\boldsymbol{w}^\star, Q_\xi, \tilde{d}_\xi\right) \leq \frac{4}{K}\left(\frac{\sqrt{2p}}{1-\gamma}\left(\frac{C_\phi^2}{\mu(1-\gamma)} + 1\right) + \frac{C_\phi^2}{\mu(1-\gamma)^2}\right)^2, \tag{113}$$

*where $\boldsymbol{w}^\star \in \arg\min_{\boldsymbol{w}} \ell\left(\boldsymbol{w}, Q^\xi, \tilde{d}_\xi\right)$.*

The proof of Theorem E.3 is postponed to Appendix G.5.

The following lemma provide a (very pessimistic) upper bound of $C_\nu$ in Assumption 4.3.

**Lemma E.4** (Upper bound of $C_\nu$). *If $\nu(s,a) > 0$ for all state-action pairs $(s,a) \in \mathcal{S} \times \mathcal{A}$, then we have*

$$C_\nu \leq \frac{1}{(1-\gamma)^2 \nu_{\min}^2}.$$

*Proof.* We only need to note that

$$\sqrt{\mathbb{E}_{(s.a) \sim \tilde{d}^{(t)}}\left[\left(\frac{h^{(t)}(s,a)}{\tilde{d}_n^{(t)}(s,a)}\right)^2\right]} \leq \max_{(s,a) \in \mathcal{S} \times \mathcal{A}} \frac{h^{(t)}(s,a)}{\tilde{d}_n^{(t)}(s,a)} \leq \frac{1}{(1-\gamma)\nu_{\min}},$$

where the last inequality follows from (??). $\qquad\square$

We give some key lemmas which will be used in our proof of Theorem 4.4.

**Lemma E.5** (consensus properties). *For all $t \in \mathbb{N}$, we have*

$$\bar{\boldsymbol{\xi}}^{(t+1)} = \bar{\boldsymbol{\xi}}^{(t)} + \alpha \hat{\boldsymbol{w}}^{(t)}, \tag{114}$$

$$\frac{1}{N} \boldsymbol{1}^\top \boldsymbol{h}^{(t)} = \frac{1}{N} \sum_{n=1}^{N} \boldsymbol{h}_n^{(t)} = \hat{\boldsymbol{w}}^{(t)}. \tag{115}$$

*Proof.* (115) could be obtained directly by using mathematical induction and update rule (33) (note that $\frac{1}{N}\mathbf{1}^\top \boldsymbol{h}^{(-1)} = \hat{\boldsymbol{w}}^{(-1)} = \mathbf{0}$, see line 2 of Algorithm 3), and (114) could be obtained by averaging both sides of (34) and using (115). $\qquad\square$

**Lemma E.6** (Young's inequalities). *Let $\{\boldsymbol{x}_1, \cdots, \boldsymbol{x}_m\}$ be a set of $m$ vectors in $\mathbb{R}^l$. Then for any $\zeta > 0$, we have*

$$\|\boldsymbol{x}_i + \boldsymbol{x}_j\|_2^2 \leq (1+\zeta)\|\boldsymbol{x}_i\|_2^2 + (1+1/\zeta)\|\boldsymbol{x}_j\|_2^2, \tag{116}$$

$$\left\|\sum_{i=1}^m \boldsymbol{x}_i\right\|_2^2 \leq m \sum_{i=1}^m \|\boldsymbol{x}_i\|_2^2. \tag{117}$$

**Lemma E.7** (Lipschitzness of $Q$-function with function approximation). *Assume that $r(s,a) \in [0,1], \forall (s,a) \in \mathcal{S} \times \mathcal{A}$. For any $\boldsymbol{\xi}, \boldsymbol{\xi}' \in \mathbb{R}^p$, we have*

$$\forall (s,a) \in \mathcal{S} \times \mathcal{A}: \quad |Q^{f_{\boldsymbol{\xi}'}}(s,a) - Q^{f_{\boldsymbol{\xi}}}(s,a)| \leq \underbrace{\frac{2C_\phi\gamma(1+\gamma)}{(1-\gamma)^2}}_{:=L_Q}\|\boldsymbol{\xi}' - \boldsymbol{\xi}\|_2. \tag{118}$$

*Proof.* See Appendix G.6. $\qquad\square$

For each iteration step $t$ in Algorithm 3, we let $\bar{\boldsymbol{\xi}}^{(t)} := \frac{1}{N}\sum_{n=1}^N \boldsymbol{\xi}_n^{(t)} = \frac{1}{N}\boldsymbol{\xi}^{(t)\top}\mathbf{1}_N$. We define

$$\Omega_1^{(t)} := \mathbb{E}\left\|\boldsymbol{\xi}^{(t)} - \mathbf{1}_N \bar{\boldsymbol{\xi}}^{(t)\top}\right\|_F^2, \tag{119}$$

$$\Omega_2^{(t)} := \mathbb{E}\left\|\boldsymbol{h}^{(t)} - \mathbf{1}_N \hat{\boldsymbol{w}}^{(t)\top}\right\|_F^2, \tag{120}$$

We let

$$\bar{\varepsilon}_{\text{stat}} := \frac{1}{N}\sum_{n=1}^N \varepsilon_{\text{stat}}^n, \tag{121}$$

$$\bar{\varepsilon}_{\text{approx}} := \frac{1}{N}\sum_{n=1}^N \varepsilon_{\text{approx}}^n, \tag{122}$$

and define $\delta^{(t)} := V^\star - \bar{V}^{(t)}(\rho)$, where $\bar{V}^{(t)}$ is shorthand for $V^{\bar{f}^{(t)}}$. We give the following performance improvement lemma.

**Lemma E.8** (Performance improvement of FedNAC). *Fix a state distribution $\rho$, then we have*

$$\vartheta_\rho \delta^{(t+1)} + \frac{D_\star^{(t+1)}}{(1-\gamma)\alpha} \leq \vartheta_\rho \delta^{(t)} + \frac{D_\star^{(t)}}{(1-\gamma)\alpha} - \delta^{(t)}$$
$$+ \frac{2\sqrt{C_\nu}(\vartheta_\rho + 1)}{1-\gamma}\left(\sqrt{\bar{\varepsilon}_{stat}} + \sqrt{2\left(\bar{\varepsilon}_{approx} + \frac{L_Q^2}{N}\left\|\boldsymbol{\xi}^{(t)} - \mathbf{1}_N\bar{\boldsymbol{\xi}}^{(t)\top}\right\|_F^2\right)}\right). \tag{123}$$

*Proof.* See Appendix G.7. $\qquad\square$

**Lemma E.9** (linear system). *For any $t \in \mathbb{N}$, we let $\boldsymbol{\Omega}^{(t)} = (\Omega_1^{(t)}, \Omega_2^{(t)})^\top$. Then for any $\zeta > 0$, we have*

$$\boldsymbol{\Omega}^{(t+1)} \leq \boldsymbol{C}\boldsymbol{\Omega}^{(t)} + \boldsymbol{s}, \tag{124}$$

*where*

$$\boldsymbol{C} = (c_{ij}) = \begin{pmatrix} (1+\zeta)\sigma^2 & \alpha^2(1+1/\zeta)\sigma^2 \\ (1+1/\zeta)\frac{96\sigma^2 L_Q^2}{(1-\gamma)\mu} & \sigma^2\left(1+\zeta+(1+1/\zeta)\frac{24L_Q^2\alpha^2}{(1-\gamma)\mu}\right) \end{pmatrix}, \tag{125}$$

*and*

$$\boldsymbol{s} = \begin{pmatrix} s_1 \\ s_2 \end{pmatrix} = \begin{pmatrix} 0 \\ (1+1/\zeta)\frac{6\sigma^2}{(1-\gamma)\mu}\left(N(\bar{\varepsilon}_{stat} + C_\nu\bar{\varepsilon}_{approx}) + 4L_Q^2\left(\frac{\alpha^2 N\bar{\varepsilon}_{stat}}{(1-\gamma)\mu} + \frac{\alpha^2 NC_\phi^2}{\mu^2(1-\gamma)^2}\right)\right) \end{pmatrix}. \tag{126}$$

*Proof.* See Appendix G.8. □

Now we are ready to give the formal version of Theorem 4.4 and its proof.

**Theorem E.10** (Convergence rate of FedNAC (formal)). *Let $\boldsymbol{\xi}_1^{(0)} = \cdots = \boldsymbol{\xi}_N^{(0)}$ in FedNAC (Algorithm 3), let the $\boldsymbol{w}^{(0)} = \boldsymbol{0}$ and the critic stepsize $\beta = \frac{1}{2C_\phi}$ in Algorithm 4. Then under Assumptions 3.1, 4.1, 4.2 and 4.3, when the actor stepsize satisfies*

$$\alpha \le \alpha_1 := \frac{(1-\sigma^2)^3 \sqrt{(1-\gamma)\mu}}{768\sqrt{6}\sigma L_Q}, \tag{127}$$

*where $L_Q$ is defined in Lemma E.7, we have*

$$
\begin{aligned}
&V^\star(\rho) - \frac{1}{T}\sum_{t=0}^{T-1}\mathbb{E}\left[\bar{V}^{(t)}(\rho)\right] \\
&\le \frac{D_\star^{(0)} + \alpha\vartheta_\rho}{T(1-\gamma)\alpha} + \frac{1}{T} \cdot \frac{512\sqrt{6}C_\phi\sqrt{C_\nu}(\vartheta_\rho+1)\sigma\alpha}{(1-\sigma^2)^{3/2}(1-\gamma)^3\sqrt{N}}\sqrt{\Omega_2^{(0)}} \\
&\quad + \left[\frac{2\sqrt{C_\nu}(\vartheta_\rho+1)}{1-\gamma} + \sqrt{1 + \frac{64C_\phi^2\alpha^2}{(1-\gamma)^5\mu}} \cdot \frac{3072\sqrt{3}C_\phi\sqrt{C_\nu}(\vartheta_\rho+1)\sigma^2\alpha}{(1-\sigma^2)^3(1-\gamma)^{7/2}\sqrt{\mu}}\right] \\
&\qquad \cdot \frac{2}{(1-\gamma)^2\sqrt{K}}\left((\sqrt{2p}+1)C_\phi^2 + \sqrt{2p}\mu(1-\gamma)\right) \\
&\quad + \left[\frac{2\sqrt{2C_\nu}(\vartheta_\rho+1)}{1-\gamma} + \frac{3072\sqrt{3}C_\phi C_\nu(\vartheta_\rho+1)\sigma^2\alpha}{(1-\sigma^2)^3(1-\gamma)^{7/2}\sqrt{\mu}}\right]\sqrt{\bar{\varepsilon}_{approx}} + \frac{6144\sqrt{2}\sigma^2 C_\nu(\vartheta_\rho+1)C_\phi^3\alpha^2}{(1-\gamma)^{13/2}\mu^{3/2}(1-\sigma^2)^3}.
\end{aligned}
\tag{128}
$$

*Moreover, the consensus errors could be upper bounded by*

$$\mathbb{E}\left\|\boldsymbol{\xi}^{(t)} - \mathbf{1}_N\bar{\boldsymbol{\xi}}^{(t)\top}\right\|_F^2 \le \left(\frac{49}{64}\sigma^2 + \frac{15}{64}\right)^t \mathbb{E}\left\|\boldsymbol{h}^{(0)} - \mathbf{1}_N\hat{\boldsymbol{w}}^{(0)\top}\right\|_F^2 + \frac{64\delta(\alpha,K)}{15(1-\sigma^2)}, \tag{129}$$

*where*

$$\delta(\alpha,K) := \frac{18\sigma^2 N}{(1-\sigma^2)(1-\gamma)\mu}\left(\bar{\varepsilon}_{stat} + C_\nu\bar{\varepsilon}_{approx}\right) + \frac{72\sigma^2 L_Q^2 N}{(1-\gamma)^3\mu^3(1-\sigma^2)}\left((1-\gamma)\mu\bar{\varepsilon}_{stat} + C_\phi^2\right)\alpha^2, \tag{130}$$

*and*

$$\bar{\varepsilon}_{stat} \le \frac{4}{(1-\gamma)^4 K}\left((\sqrt{2p}+1)C_\phi^2 + \sqrt{2p}\mu(1-\gamma)\right)^2.$$

*Remark* E.11 (Sample and communication complexity). When $\sigma > 0$ and

$$\alpha = \frac{\sqrt{\mu}(D_\star^{(0)})^{1/3}}{6144^{1/3}2^{1/6}C_\nu^{1/3}(1+\vartheta_\rho)^{1/3}C_\phi} \cdot \frac{(1-\gamma)^{11/6}(1-\sigma^2)}{T^{1/3}\sigma^{2/3}},$$

it follows from Theorem E.10 that

$$
\begin{aligned}
&V^\star(\rho) - \frac{1}{T}\sum_{t=0}^{T-1}\mathbb{E}\left[\bar{V}^{(t)}(\rho)\right] \\
&\le \frac{3^{1/3}\cdot 2^{29/6}(D_\star^{(0)})^{2/3}C_\nu^{1/3}(1+\vartheta_\rho)^{1/3}C_\phi\sigma^{2/3}}{T^{2/3}(1-\gamma)^{17/6}(1-\sigma^2)\sqrt{\mu}} + \frac{\vartheta_\rho}{(1-\gamma)T} + \frac{2^{17/3}3^{1/6}C_\nu^{1/6}(1+\vartheta_\rho)^{2/3}\sigma^{1/3}\sqrt{\mu}(D_\star^{(0)})^{1/3}}{T^{4/3}(1-\sigma^2)^{1/2}(1-\gamma)^{7/6}\sqrt{N}} \\
&\quad + \left[\frac{2\sqrt{C_\nu}(\vartheta_\rho+1)}{1-\gamma} + \sqrt{1 + \frac{(D_\star^{(0)})^{2/3}(1-\sigma^2)^2}{3^{3/2}\cdot 4C_\nu^{2/3}(1-\gamma)^{4/3}(1+\vartheta_\rho)^{1/3}T^{2/3}\sigma^{4/3}} \cdot \frac{2^{37/6}\cdot 3^{7/6}C_\nu^{1/6}(\vartheta_\rho+1)^{2/3}\sigma^{4/3}(D_\star^{(0)})^{1/3}}{(1-\sigma^2)^2(1-\gamma)^{5/3}T^{1/3}}}\right] \\
&\qquad \cdot \frac{2}{(1-\gamma)^2\sqrt{K}}\left((\sqrt{2p}+1)C_\phi^2 + \sqrt{2p}\mu(1-\gamma)\right) \\
&\quad + \left[\frac{2\sqrt{2C_\nu}(\vartheta_\rho+1)}{1-\gamma} + \frac{2^{37/6}\cdot 3^{7/6}C_\nu^{1/6}(\vartheta_\rho+1)^{2/3}\sigma^{4/3}(D_\star^{(0)})^{1/3}}{(1-\sigma^2)^2(1-\gamma)^{5/3}T^{1/3}}\right]\sqrt{\bar{\varepsilon}_{approx}}.
\end{aligned}
\tag{131}
$$

Consequently, we need

$$T \gtrsim \left\{ \frac{\sigma}{\varepsilon^{3/2}(1-\gamma)^{17/4}(1-\sigma^2)^{3/2}}, \frac{1}{\varepsilon(1-\gamma)}, \frac{\sigma^{1/4}}{\varepsilon^{3/4}(1-\sigma^2)^{3/8}(1-\gamma)^{7/8}N^{3/8}}, \frac{\sigma^4}{(1-\gamma)^2(1-\gamma^2)^6} \right\}$$

and

$$K = \mathcal{O}\left( \frac{1}{(1-\gamma)^6\varepsilon^2} \right)$$

such that $V^\star(\rho) - \frac{1}{T}\sum_{t=0}^{T-1} \mathbb{E}\left[ \bar{V}^{(t)}(\rho) \right] \lesssim \varepsilon + \frac{\bar{\varepsilon}_{\text{approx}}}{1-\gamma}$. In Algorithm 5, each trajectory has the expected length $1/(1-\gamma)$. Consider only the term where $\varepsilon$ dominates, FedNAC requires $\mathcal{O}\left( \frac{1}{(1-\gamma)^{45/4}\varepsilon^{7/2}(1-\sigma^2)^{3/2}} \right)$ samples for each agent and $\mathcal{O}\left( \frac{1}{\varepsilon^{3/2}(1-\gamma)^{17/4}(1-\sigma^2)^{3/2}} \right)$ rounds of communication.

On the other end, when $\sigma = 0$, (128) becomes:

$$V^\star(\rho) - \frac{1}{T}\sum_{t=0}^{T-1} \mathbb{E}\left[ \bar{V}^{(t)}(\rho) \right] \leq \frac{D_\star^{(0)} + \alpha\vartheta_\rho}{T(1-\gamma)\alpha} + \frac{4\sqrt{C_\nu}(\vartheta_\rho + 1)}{(1-\gamma)^3\sqrt{K}} \left( (\sqrt{2p}+1)C_\phi^2 + \sqrt{2p}\mu(1-\gamma) \right)$$

$$+ \frac{2\sqrt{2C_\nu}(\vartheta_\rho + 1)}{1-\gamma}\sqrt{\bar{\varepsilon}_{\text{approx}}}, \tag{132}$$

Consequently, for any fixed $\alpha > 0$, when $\sigma = 0$ or close to 0, with $T = \mathcal{O}\left( \frac{1}{(1-\gamma)\varepsilon} \right)$ and $K = \mathcal{O}\left( \frac{1}{(1-\gamma)^6\varepsilon^2} \right)$, FedNAC requires $KT/(1-\gamma) = \mathcal{O}\left( \frac{1}{(1-\gamma)^8\varepsilon^3} \right)$ samples for each agent and $T = \mathcal{O}\left( \frac{1}{(1-\gamma)\varepsilon} \right)$ rounds of communication such that $V^\star(\rho) - \frac{1}{T}\sum_{t=0}^{T-1} \mathbb{E}\left[ \bar{V}^{(t)}(\rho) \right] \lesssim \varepsilon + \frac{\bar{\varepsilon}_{\text{approx}}}{1-\gamma}$.

### E.1 Proof of Theorem E.10

We suppose Assumptions 3.1, E.2, 4.1, 4.2 and 4.3 holds. By Lemma E.9 and nonnegativity of each entry of $C$, $s$ and $\Omega^{(t)}$ where $t \in \mathbb{N}$, it's easy to see that

$$\sqrt{\Omega^{(t+1)}} \leq \sqrt{C}\sqrt{\Omega^{(t)}} + \sqrt{s}, \tag{133}$$

where $\sqrt{\cdot}$ is exerted element-wise.

In addition, taking expectation on both sides of (123) and using the act that

$$\mathbb{E}\left[ \sqrt{2\left( \bar{\varepsilon}_{\text{approx}} + \frac{L_Q^2}{N}\left\| \boldsymbol{\xi}^{(t)} - \mathbf{1}_N\bar{\xi}^{(t)\top} \right\|_F^2 \right)} \right] \leq \sqrt{2\bar{\varepsilon}_{\text{approx}}} + \sqrt{\frac{2L_Q^2}{N}\Omega_1^{(t)}},$$

we have

$$\vartheta_\rho\mathbb{E}[\delta^{(t+1)}] + \frac{\mathbb{E}[D_\star^{(t+1)}]}{(1-\gamma)\alpha} \leq \vartheta_\rho\mathbb{E}[\delta^{(t)}] + \frac{\mathbb{E}[D_\star^{(t)}]}{(1-\gamma)\alpha} - \mathbb{E}[\delta^{(t)}]$$

$$+ \frac{2\sqrt{C_\nu}(\vartheta_\rho + 1)}{1-\gamma} \left( \sqrt{\bar{\varepsilon}_{\text{stat}}} + \sqrt{2\bar{\varepsilon}_{\text{approx}}} + \sqrt{\frac{2L_Q^2}{N}\Omega_1^{(t)}} \right). \tag{134}$$

We define the Lyapunov function $\Phi^{(t)}$ as follows:

$$\Phi^{(t)} := \vartheta_\rho\mathbb{E}[\delta^{(t)}] + \frac{\mathbb{E}[D_\star^{(t)}]}{(1-\gamma)\alpha} + \boldsymbol{q}^\top\sqrt{\Omega^{(t)}}, \tag{135}$$

where

$$\boldsymbol{q} = \begin{pmatrix} q_1 \\ q_2 \end{pmatrix} = \begin{pmatrix} \frac{2L_Q\sqrt{2C_\nu}(\vartheta_\rho+1)}{(1-\gamma)\sqrt{N}} \cdot \frac{1}{1-\sqrt{1+\zeta}\sigma-\sqrt{(1+1/\zeta)c_{21}}\sigma\alpha/(1-\sqrt{c_{22}})} \\ \frac{2L_Q\sqrt{2C_\nu}(\vartheta_\rho+1)}{(1-\gamma)\sqrt{N}} \cdot \frac{\sqrt{1+1/\zeta}\sigma\alpha}{(1-\sqrt{1+\zeta}\sigma)(1-\sqrt{c_{22}})-\sqrt{(1+1/\zeta)c_{21}}\sigma\alpha} \end{pmatrix}. \tag{136}$$

It's straightforward to verify that when $\zeta = \frac{1-\sigma^2}{2}$, we have the entries in $C$ (cf. (125)) satisfies

$$c_{11} < \frac{1+\sigma^2}{2}, \tag{137}$$

$$c_{12} \leq \frac{3\sigma^2\alpha^2}{1-\sigma^2}. \tag{138}$$

Moreover, from $\alpha \leq \frac{\sqrt{(1-\gamma)\mu}(1-\sigma^2)}{12\sqrt{2}\sigma L_Q}$ we deduce

$$c_{22} \leq \frac{3+\sigma^2}{4}, \tag{139}$$

which gives

$$1 - \sqrt{c_{22}} \geq 1 - \sqrt{\frac{3+\sigma^2}{4}} \geq \frac{1-\sigma^2}{8}, \tag{140}$$

Also note that $\alpha \leq \frac{(1-\sigma^2)^3\sqrt{(1-\gamma)\mu}}{768\sqrt{6}\sigma^2 L_Q}$ yields

$$\sqrt{(1+1/\zeta)c_{21}}\sigma\alpha \leq \frac{(1-\sqrt{1+\zeta}\sigma)(1-\sqrt{c_{22}})}{2}.$$

which together with (140) and the fact $1 - \sqrt{1+\zeta}\sigma \geq \frac{1-\sigma^2}{4}$ indicates $q_1, q_2 > 0$ and that

$$q_1 \leq \frac{16\sqrt{2}L_Q\sqrt{C_\nu}(\vartheta_\rho + 1)}{(1-\sigma^2)(1-\gamma)\sqrt{N}}, \tag{141}$$

$$q_2 \leq \frac{128\sqrt{6}L_Q\sqrt{C_\nu}(\vartheta_\rho + 1)\sigma\alpha}{(1-\sigma^2)^{5/2}(1-\gamma)\sqrt{N}}. \tag{142}$$

Thus by (133) and (134) we have

$$\Phi^{(t+1)} = \vartheta_\rho\mathbb{E}[\delta^{(t+1)}] + \frac{\mathbb{E}[D_\star^{(t+1)}]}{(1-\gamma)\alpha} + q^\top\sqrt{\Omega^{(t+1)}}$$

$$\leq \vartheta_\rho\mathbb{E}[\delta^{(t)}] + \frac{\mathbb{E}[D_\star^{(t)}]}{(1-\gamma)\alpha} - \mathbb{E}[\delta^{(t)}] + q^\top\left(\sqrt{C}\sqrt{\Omega^{(t)}} + \sqrt{s}\right)$$

$$+ \frac{2\sqrt{C_\nu}(\vartheta_\rho + 1)}{1-\gamma}\left(\sqrt{\bar{\varepsilon}_{\text{stat}}} + \sqrt{2\bar{\varepsilon}_{\text{approx}}} + \sqrt{\frac{2L_Q^2}{N}\Omega_1^{(t)}}\right)$$

$$= \Phi^{(t)} + \left(\underbrace{q^\top(\sqrt{C} - I) + \left(\frac{2L_Q\sqrt{2C_\nu}(\vartheta_\rho + 1)}{(1-\gamma)\sqrt{N}}, 0\right)}_{=(0,0)}\right)\sqrt{\Omega^{(t)}}$$

$$+ \frac{2\sqrt{C_\nu}(\vartheta_\rho + 1)}{1-\gamma}\left(\sqrt{\bar{\varepsilon}_{\text{stat}}} + \sqrt{2\bar{\varepsilon}_{\text{approx}}}\right) + q_2\sqrt{s_2} - \mathbb{E}[\delta^{(t)}], \tag{143}$$

which gives

$$\mathbb{E}[\delta^{(t)}] \leq \Phi^{(t)} - \Phi^{(t+1)} + \frac{2\sqrt{C_\nu}(\vartheta_\rho + 1)}{1-\gamma}\left(\sqrt{\bar{\varepsilon}_{\text{stat}}} + \sqrt{2\bar{\varepsilon}_{\text{approx}}}\right) + q_2\sqrt{s_2}. \tag{144}$$

Summing the above inequality over $t = 0, 1, \cdots, T-1$ and divide both sides by $T$, we have

$$\frac{1}{T}\sum_{t=0}^{T-1}\mathbb{E}[\delta^{(t)}] \leq \frac{\Phi^{(0)} - \Phi^{(t)}}{T} + \frac{2\sqrt{C_\nu}(\vartheta_\rho + 1)}{1-\gamma}\left(\sqrt{\bar{\varepsilon}_{\text{stat}}} + \sqrt{2\bar{\varepsilon}_{\text{approx}}}\right) + q_2\sqrt{s_2}. \tag{145}$$

Since

$$s_2 \leq \frac{18\sigma^2 N}{(1-\sigma^2)(1-\gamma)\mu}\left(\bar{\varepsilon}_{\text{stat}} + C_\nu\bar{\varepsilon}_{\text{approx}}\right) + \frac{72\sigma^2 L_Q^2 N}{(1-\gamma)^3\mu^3(1-\sigma^2)}\left((1-\gamma)\mu\bar{\varepsilon}_{\text{stat}} + C_\phi^2\right)\alpha^2, \tag{146}$$

and

$$\Phi^{(0)} - \Phi^{(t)} \le \Phi^{(0)} \le \frac{\vartheta_\rho}{1-\gamma} + \frac{\mathbb{E}[D_\star^{(0)}]}{(1-\gamma)\alpha} + \frac{16\sqrt{2}L_Q\sqrt{C_\nu}(\vartheta_\rho+1)}{(1-\sigma^2)(1-\gamma)\sqrt{N}}\left(\sqrt{\Omega_1^{(0)}} + \frac{8\sqrt{3}\sigma\alpha}{\sqrt{1-\sigma^2}}\sqrt{\Omega_2^{(0)}}\right),$$

(147)

we have (recall that $L_Q = \frac{2C_\phi\gamma(1+\gamma)}{(1-\gamma)^2} \le \frac{4C_\phi}{(1-\gamma)^2}$)

$$V^\star(\rho) - \frac{1}{T}\sum_{t=0}^{T-1}\mathbb{E}\left[\bar{V}^{(t)}(\rho)\right]$$

$$\le \frac{D_\star^{(0)} + \alpha\vartheta_\rho}{T(1-\gamma)\alpha} + \frac{1}{T}\cdot\frac{64\sqrt{2}C_\phi\sqrt{C_\nu}(\vartheta_\rho+1)}{(1-\sigma^2)(1-\gamma)^3\sqrt{N}}\left(\sqrt{\Omega_1^{(0)}} + \frac{8\sqrt{3}\sigma\alpha}{\sqrt{1-\sigma^2}}\sqrt{\Omega_2^{(0)}}\right)$$

$$+ \left[\frac{2\sqrt{C_\nu}(\vartheta_\rho+1)}{1-\gamma} + \sqrt{\frac{18\sigma^2 N}{(1-\sigma^2)(1-\gamma)\mu} + \frac{1152\sigma^2 C_\phi^2 N\alpha^2}{(1-\gamma)^6\mu^2(1-\sigma^2)}}\cdot\frac{512\sqrt{6}C_\phi\sqrt{C_\nu}(\vartheta_\rho+1)\sigma\alpha}{(1-\sigma^2)^{5/2}(1-\gamma)^3\sqrt{N}}\right]\sqrt{\bar{\varepsilon}_{\text{stat}}}$$

$$+ \left[\frac{2\sqrt{2C_\nu}(\vartheta_\rho+1)}{1-\gamma} + \sqrt{\frac{18\sigma^2 NC_\nu}{(1-\sigma^2)(1-\gamma)\mu}}\cdot\frac{512\sqrt{6}C_\phi\sqrt{C_\nu}(\vartheta_\rho+1)\sigma\alpha}{(1-\sigma^2)^{5/2}(1-\gamma)^3\sqrt{N}}\right]\sqrt{\bar{\varepsilon}_{\text{approx}}}$$

$$+ \frac{6144\sqrt{2}\sigma^2\sqrt{C_\nu}(\vartheta_\rho+1)C_\phi^3\alpha^2}{(1-\gamma)^{13/2}\mu^{3/2}(1-\sigma^2)^3}.$$

(148)

By Theorem E.3 we know that $\sqrt{\bar{\varepsilon}_{\text{stat}}}$ could be upper bounded as follows:

$$\sqrt{\bar{\varepsilon}_{stat}} \le \frac{2}{(1-\gamma)^2\sqrt{K}}\left((\sqrt{2p}+1)C_\phi^2 + \sqrt{2p}\mu(1-\gamma)\right).$$

(149)

(128) follows from plugging (149) into (148) and noting that when $\boldsymbol{\xi}_1^{(0)} = \cdots = \boldsymbol{\xi}_N^{(0)}$, $\Omega_1^{(0)} = 0$.

**Bounding the consensus errors.** Similar to Step 4 in Appendix F.4, to bound the consensus error $\left\|\log f_n^{(t)} - \log \bar{f}^{(t)}\right\|_\infty$ for all $n \in [N]$, we first upper bound the eigenvalue of $\rho(\boldsymbol{C})$—the spectral norm of $\boldsymbol{C}$.

The characteristic polynomial of $\boldsymbol{C}$ is

$$f(\lambda) = (\lambda - c_{11})(\lambda - c_{22}) - c_{12}c_{21}$$
$$= \lambda^2 - (c_{11}+c_{22})\lambda + c_{11}c_{22} - c_{12}c_{21},$$

which gives

$$\rho(\boldsymbol{C}) \le \frac{c_{11}+c_{22}+\sqrt{(c_{11}+c_{22})^2 - 4(c_{11}c_{12} - c_{12}c_{21})}}{2}$$

$$= \frac{c_{11}+c_{22}+\sqrt{(c_{22}-c_{11})^2 + 4c_{12}c_{21}}}{2}$$

$$\le \frac{c_{11}+c_{22}+c_{22}-c_{11}+2\sqrt{c_{12}c_{21}}}{2}$$

$$= c_{22} + \sqrt{c_{12}c_{21}}$$

$$\le \frac{3+\sigma^2}{4} + \frac{\sqrt{3}\sigma\alpha}{\sqrt{1-\sigma^2}}\cdot\frac{12\sqrt{2}L_Q\sigma}{\sqrt{1-\sigma^2}(1-\gamma)\mu}$$

$$\le \frac{3+\sigma^2}{4} + \frac{\sigma(1-\sigma^2)^2}{64}$$

$$\le \frac{49+15\sigma^2}{64} < 1,$$

(150)

where the third inequality uses (138), (139), and the fourth inequality uses (127).

Therefore, similar to (230), when $\alpha \le \alpha_1$, we have

$$\left\|\boldsymbol{\Omega}^{(t)}\right\|_2 \le \left(\frac{49}{64}\sigma + \frac{15}{64}\right)^t\left\|\boldsymbol{\Omega}^{(0)}\right\|_2 + \frac{64s_2}{15(1-\sigma^2)}.$$

(151)

Combining the above inequality with (146), and (149), we obtain (129).

## E.2 Proof of Theorem E.3

The proof of Theorem E.3 could be found in Appendix C.5 in [YDG$^+$22]. We present it for completeness. To prove Theorem E.3, we need the following Theorem G.2.

**Theorem E.12** (Theorem 1 in [BM13]). *Consider the following assumptions:*

    *(i) The observations $(\boldsymbol{a}_k, \boldsymbol{b}_k) \in \mathbb{R}^p \times \mathbb{R}^p$ are independent and identically distributed.*

    *(ii) $\mathbb{E}\left[\|\boldsymbol{a}_k\|^2\right]$[7] and $\mathbb{E}\left[\|\boldsymbol{b}_k\|^2\right]$ are finite. The covariance $\mathbb{E}\left[\boldsymbol{a}_k \boldsymbol{a}_k^\top\right]$ is invertible.*

    *(iii) The global minimum of $g(w) = \frac{1}{2}\mathbb{E}\left[\langle \boldsymbol{w}, \boldsymbol{a}_k \rangle^2 - 2\langle \boldsymbol{w}, \boldsymbol{b}_k \rangle\right]$ is attained at a certain $\boldsymbol{w}^\star \in \mathbb{R}^p$. Let $\Delta_k = \boldsymbol{b}_k - \langle \boldsymbol{w}^\star, \boldsymbol{a}_k \rangle \boldsymbol{a}_k$ denote the residual. We have $\mathbb{E}[\Delta_k] = 0$.*

    *(iv) $\exists R > 0$ and $\sigma > 0$ such that $\mathbb{E}\left[\Delta_k \Delta_k^\top\right] \leq \sigma^2 \mathbb{E}\left[\boldsymbol{a}_k \boldsymbol{a}_k^\top\right]$ and $\mathbb{E}\left[\|\boldsymbol{a}_k\|^2 \boldsymbol{a}_k \boldsymbol{a}_k^\top\right] \leq R^2 \mathbb{E}\left[\boldsymbol{a}_k \boldsymbol{a}_k^\top\right]$.*

*Consider the stochastic gradient recursion*

$$w_{k+1} = w_k - \eta\left(\langle w_k, a_k \rangle a_k - b_k\right)$$

*started from $w_0 \in \mathbb{R}^p$. Let $w_{out} = \frac{1}{K}\sum_{k=1}^{K} w_k$. When $\eta = \frac{1}{4R^2}$, we have*

$$\mathbb{E}\left[g(w_{out}) - g(w^\star)\right] \leq \frac{2}{K}(\sigma\sqrt{p} + R\|w_0 - w^\star\|)^2. \tag{152}$$

In the proof of Theorem E.3 we'll show that for Algorithm 4, the assumptions in Theorem G.2 are all satisfied and thus we can use the result (267).

*Proof of Theorem E.3.* We let $a_k$ and $b_k$ in Theorem G.2 be $\phi(s,a)$ and $\widehat{Q}_\xi \phi(s,a)$ in Algorithm 4, respectively. And we let $\|\cdot\| = \|\cdot\|_2$ in Theorem G.2. Since the observations $\left(\phi(s,a), \widehat{Q}_\xi(s,a)\phi(s,a)\right) \in \mathbb{R}^p \times \mathbb{R}^p$ are i.i.d., (i) is satisfied.

As we assume $\|\phi(s,a)\|_2 \leq C_\phi$, $\mathbb{E}\left[\|\phi(s,a)\|_2^2\right]$ is finite. From Assumption 4.1 we know that $\mathbb{E}\left[\phi(s,a)\phi(s,a)^\top\right]$ is invertible.

Let $H$ be the length of trajectory for estimating $\widehat{Q}_\xi(s,a)$. Then $\left(\widehat{Q}_\xi(s,a)\right)^2$ is bounded by

$$\mathbb{E}\left[\left(\widehat{Q}_\xi(s,a)\right)^2\right] = \mathbb{E}_{(s,a)\sim \tilde{d}_\nu^{\pi_\xi}}\left[\sum_{\tau=0}^{\infty} Pr(H=\tau)\mathbb{E}\left[\left(\sum_{t=0}^{\tau} r(s_t, a_t)\right)^2 \middle| H=\tau, s_0=s, a_0=a\right]\right]$$

$$= \mathbb{E}_{(s,a)\sim \tilde{d}_\nu^{\pi_\xi}}\left[(1-\gamma)\sum_{\tau=0}^{\infty} \gamma^\tau \mathbb{E}\left[\left(\sum_{t=0}^{\tau} r(s_t, a_t)\right)^2 \middle| H=\tau, s_0=s, a_0=a\right]\right]$$

$$\leq \mathbb{E}_{(s,a)\sim \tilde{d}_\nu^{\pi_\xi}}\left[(1-\gamma)\sum_{\tau=0}^{\infty} \gamma^\tau (\tau+1)^2\right] \leq \frac{2}{(1-\gamma)^2}, \tag{153}$$

from which we deduce $\mathbb{E}\left[\left\|\widehat{Q}_\xi(s,a)\phi(s,a)\right\|_2^2\right] \leq C_\phi^2 \mathbb{E}\left[\widehat{Q}_\xi(s,a)^2\right]$ is bounded. Thus (ii) holds.

Furthermore, we introduce the residual

$$\Delta := \left(\widehat{Q}_\xi(s,a) - \phi(s,a)^\top w^\star\right)\phi(s,a), \tag{154}$$

then from Lemma 7 in [YDG$^+$22] we know that $\mathbb{E}[\Delta] = \frac{1}{2}\nabla_w \ell(w, \widehat{Q}_\xi, d_\nu^{\pi_\xi}) = 0$, which gives (iii).

---

[7]Here $\|\cdot\|$ could be any norm in $\mathbb{R}^p$.

To verify (iv), we let $R = C_\phi$ in Theorem G.2, then $\mathbb{E}\left[\|\phi(s,a)\|_2^2 \phi(s,a)\phi(s,a)^\top\right] \leq C_\phi^2 \mathbb{E}\left[\phi(s,a)\phi(s,a)^\top\right]$. Also note that

$$
\begin{aligned}
w^\star &= \left(\mathbb{E}_{(s,a)\sim \tilde{d}_\nu^{\pi\xi}}\left[\phi(s,a)\phi(s,a)^\top\right]\right)^\dagger \mathbb{E}_{(s,a)\sim \tilde{d}_\nu^{\pi\xi}}\left[\widehat{Q}_\xi(s,a)\phi(s,a)\right] \\
&\leq \frac{1}{1-\gamma}\left(\mathbb{E}_{(s,a)\sim\nu}\left[\phi(s,a)\phi(s,a)^\top\right]\right)^\dagger \mathbb{E}_{(s,a)\sim \tilde{d}_\nu^{\pi\xi}}\left[\widehat{Q}_\xi(s,a)\phi(s,a)\right],
\end{aligned}
\tag{155}
$$

from which we deduce

$$
\|w^\star\|_2 \leq \frac{B}{\mu(1-\gamma)^2}.
\tag{156}
$$

$$
\mathbb{E}\left[\left(\widehat{Q}_\xi(s,a) - \phi(s,a)^\top w^\star\right)^2 | s,a\right] = \mathbb{E}\left[\left(\widehat{Q}_\xi(s,a)\right)^2 | s,a\right] - 2Q_\xi(s,a)\phi(s,a)^\top w^\star + (\phi(s,a)^\top w^\star)^2
\tag{157}
$$

$$
\leq \frac{2}{(1-\gamma)^2} + \frac{2C_\phi^2}{\mu(1-\gamma)^3} + \frac{C_\phi^4}{\mu^2(1-\gamma)^4}
$$

$$
\leq \frac{2}{(1-\gamma)^2}\left(\frac{C_\phi^2}{\mu(1-\gamma)} + 1\right)^2.
\tag{158}
$$

The above expression implies

$$
\begin{aligned}
\mathbb{E}\left[\Delta\Delta^\top\right] &= \mathbb{E}_{(s,a)\sim \tilde{d}_\nu^{\pi\xi}}\left[\left(\widehat{Q}_\xi(s,a) - \phi(s,a)^\top w^\star\right)^2 \phi(s,a)\phi(s,a)^\top | s,a\right] \\
&= \mathbb{E}_{(s,a)\sim \tilde{d}_\nu^{\pi\xi}}\left[\mathbb{E}\left[\left(\widehat{Q}_\xi(s,a) - \phi(s,a)^\top w^\star\right)^2 | s,a\right]\phi(s,a)\phi(s,a)^\top\right] \\
&\leq \left(\underbrace{\frac{\sqrt{2}}{1-\gamma}\left(\frac{C_\phi^2}{\mu(1-\gamma)} + 1\right)}_{\sigma}\right)\mathbb{E}[\phi(s,a)\phi(s,a)^\top].
\end{aligned}
\tag{159}
$$

Therefore, (iv) is verified.

Thus by (267), with stepsize $\beta = \frac{1}{2C_\phi^2}$, initialization $w_0 = 0$ and $K$ steps of critic updates, we have

$$
\begin{aligned}
\mathbb{E}\left[\ell\left(w_{\text{out}}, \widehat{Q}_\xi, \tilde{d}_\xi\right)\right] - \ell\left(w^\star, \widehat{Q}_\xi, \tilde{d}_\xi\right) &\leq \frac{4}{K}\left(\sigma\sqrt{p} + C_\phi\|w^\star\|_2\right)^2 \\
&\leq \frac{4}{K}\left(\frac{\sqrt{2p}}{1-\gamma}\left(\frac{C_\phi^2}{\mu(1-\gamma)} + 1\right) + \frac{C_\phi^2}{\mu(1-\gamma)^2}\right)^2,
\end{aligned}
$$

which gives (113). $\qquad\square$

## F  Proof of key lemmas

### F.1  Proof of Lemma D.2

Before proceeding, we summarize several useful properties of the auxiliary sequences (cf. (40) and (41)), whose proof is postponed to Appendix G.1.

**Lemma F.1** (Properties of auxiliary sequences $\{\overline{\xi}^{(t)}\}$ and $\{\boldsymbol{\xi}^{(t)}\}$). $\{\overline{\xi}^{(t)}\}$ and $\{\boldsymbol{\xi}^{(t)}\}$ *have the following properties:*

1. $\boldsymbol{\xi}^{(t)}$ *can be viewed as an unnormalized version of* $\boldsymbol{\pi}^{(t)}$, *i.e.,*

$$
\pi_n^{(t)}(\cdot|s) = \frac{\xi_n^{(t)}(s,\cdot)}{\left\|\xi_n^{(t)}(s,\cdot)\right\|_1}, \quad \forall n \in [N], s \in \mathcal{S}.
\tag{160}
$$

2. *For any $t \geq 0$, $\log \overline{\xi}^{(t)}$ keeps track of the average of $\log \boldsymbol{\xi}^{(t)}$, i.e.,*

$$\frac{1}{N}\mathbf{1}_N^\top \log \boldsymbol{\xi}^{(t)} = \log \overline{\xi}^{(t)}. \tag{161}$$

*It follows that*

$$\forall s \in \mathcal{S}, \, t \geq 0: \quad \overline{\pi}^{(t)}(\cdot|s) = \frac{\overline{\xi}^{(t)}(s,\cdot)}{\left\|\overline{\xi}^{(t)}(s,\cdot)\right\|_1}. \tag{162}$$

**Lemma F.2** ([CCC$^+$22b, Appendix. A.2]). *For any vector $\theta = [\theta_a]_{a\in\mathcal{A}} \in \mathbb{R}^{|\mathcal{A}|}$, we denote by $\pi_\theta \in \mathbb{R}^{|\mathcal{A}|}$ the softmax transform of $\theta$ such that*

$$\pi_\theta(a) = \frac{\exp(\theta_a)}{\sum_{a'\in\mathcal{A}}\exp(\theta_{a'})}, \quad a \in \mathcal{A}. \tag{163}$$

*For any $\theta_1, \theta_2 \in \mathbb{R}^{|\mathcal{A}|}$, we have*

$$\left|\log(\|\exp(\theta_1)\|_1) - \log(\|\exp(\theta_2)\|_1)\right| \leq \|\theta_1 - \theta_2\|_\infty, \tag{164}$$

$$\|\log \pi_{\theta_1} - \log \pi_{\theta_2}\|_\infty \leq 2\|\theta_1 - \theta_2\|_\infty. \tag{165}$$

**Step 1: bound** $u^{(t+1)}(s,a) = \left\|\log \boldsymbol{\xi}^{(t+1)}(s,a) - \log \overline{\xi}^{(t+1)}(s,a)\mathbf{1}_N\right\|_2$. By (40b) and (41b) we have

$$
\begin{aligned}
u^{(t+1)}(s,a) &= \left\|\log \boldsymbol{\xi}^{(t+1)}(s,a) - \log \overline{\xi}^{(t+1)}(s,a)\mathbf{1}_N\right\|_2\\
&= \left\|\alpha\left(\boldsymbol{W}\log\boldsymbol{\xi}^{(t)}(s,a) - \log\overline{\xi}^{(t)}(s,a)\mathbf{1}_N\right) + (1-\alpha)\left(\boldsymbol{W}\boldsymbol{T}^{(t)}(s,a) - \widehat{Q}_\tau^{(t)}(s,a)\mathbf{1}_N\right)/\tau\right\|_2\\
&\leq \sigma\alpha\left\|\log\boldsymbol{\xi}^{(t)}(s,a) - \log\overline{\xi}^{(t)}(s,a)\mathbf{1}_N\right\|_2 + \frac{1-\alpha}{\tau}\sigma\left\|\boldsymbol{T}^{(t)}(s,a) - \widehat{Q}_\tau^{(t)}(s,a)\mathbf{1}_N\right\|_2\\
&\leq \sigma\alpha\left\|u^{(t)}\right\|_\infty + \frac{1-\alpha}{\tau}\sigma\left\|v^{(t)}\right\|_\infty, \tag{166}
\end{aligned}
$$

where the penultimate step results from the averaging property of $\boldsymbol{W}$ (property (11)). Taking maximum over $(s,a) \in \mathcal{S}\times\mathcal{A}$ establishes the bound on $\Omega_1^{(t+1)}$ in (49).

**Step 2: bound** $v^{(t+1)}(s,a) = \left\|\boldsymbol{T}^{(t+1)}(s,a) - \widehat{Q}_\tau^{(t+1)}(s,a)\mathbf{1}_N\right\|_2$. By $(U_T)$ we have

$$
\begin{aligned}
&\left\|\boldsymbol{T}^{(t+1)}(s,a) - \widehat{Q}_\tau^{(t+1)}(s,a)\mathbf{1}_N\right\|_2\\
&= \left\|\boldsymbol{W}\left(\boldsymbol{T}^{(t)}(s,a) + \boldsymbol{Q}_\tau^{(t+1)}(s,a) - \boldsymbol{Q}_\tau^{(t)}(s,a)\right) - \widehat{Q}_\tau^{(t+1)}(s,a)\mathbf{1}_N\right\|_2\\
&= \left\|\left(\boldsymbol{W}\boldsymbol{T}^{(t)}(s,a) - \widehat{Q}_\tau^{(t)}(s,a)\mathbf{1}_N\right) + \boldsymbol{W}\left(\boldsymbol{Q}_\tau^{(t+1)}(s,a) - \boldsymbol{Q}_\tau^{(t)}(s,a)\right) + \left(\widehat{Q}_\tau^{(t)}(s,a) - \widehat{Q}_\tau^{(t+1)}(s,a)\right)\mathbf{1}_N\right\|_2\\
&\leq \sigma\left\|\boldsymbol{T}^{(t)}(s,a) - \widehat{Q}_\tau^{(t)}(s,a)\mathbf{1}_N\right\|_2 + \sigma\left\|\left(\boldsymbol{Q}_\tau^{(t+1)}(s,a) - \boldsymbol{Q}_\tau^{(t)}(s,a)\right) + \left(\widehat{Q}_\tau^{(t)}(s,a) - \widehat{Q}_\tau^{(t+1)}(s,a)\right)\mathbf{1}_N\right\|_2\\
&\leq \sigma\left\|\boldsymbol{T}^{(t)}(s,a) - \widehat{Q}_\tau^{(t)}(s,a)\mathbf{1}_N\right\|_2 + \sigma\left\|\boldsymbol{Q}_\tau^{(t+1)}(s,a) - \boldsymbol{Q}_\tau^{(t)}(s,a)\right\|_2, \tag{167}
\end{aligned}
$$

where the penultimate step uses property (11), and the last step is due to

$$
\begin{aligned}
&\left\|\left(\boldsymbol{Q}_\tau^{(t+1)}(s,a) - \boldsymbol{Q}_\tau^{(t)}(s,a)\right) + \left(\widehat{Q}_\tau^{(t)}(s,a) - \widehat{Q}_\tau^{(t+1)}(s,a)\right)\mathbf{1}_N\right\|_2^2\\
&= \left\|\boldsymbol{Q}_\tau^{(t+1)}(s,a) - \boldsymbol{Q}_\tau^{(t)}(s,a)\right\|_2^2 + N\left(\widehat{Q}_\tau^{(t)}(s,a) - \widehat{Q}_\tau^{(t+1)}(s,a)\right)^2\\
&\quad - 2\sum_{n=1}^N \left(Q_{\tau,n}^{\pi_n^{(t+1)}}(s,a) - Q_{\tau,n}^{\pi_n^{(t)}}(s,a)\right)\left(\widehat{Q}_\tau^{(t+1)}(s,a) - \widehat{Q}_\tau^{(t)}(s,a)\right)\\
&= \left\|\boldsymbol{Q}_\tau^{(t+1)}(s,a) - \boldsymbol{Q}_\tau^{(t)}(s,a)\right\|_2^2 - N\left(\widehat{Q}_\tau^{(t)}(s,a) - \widehat{Q}_\tau^{(t+1)}(s,a)\right)^2\\
&\leq \left\|\boldsymbol{Q}_\tau^{(t+1)}(s,a) - \boldsymbol{Q}_\tau^{(t)}(s,a)\right\|_2^2.
\end{aligned}
$$

**Step 3: bound** $\left\|Q_\tau^\star - \tau\log\overline{\xi}^{(t+1)}\right\|_\infty$. We decompose the term of interest as

$$
\begin{aligned}
Q_\tau^\star - \tau\log\overline{\xi}^{(t+1)} &= Q_\tau^\star - \tau\alpha\log\overline{\xi}^{(t)} - (1-\alpha)\widehat{Q}_\tau^{(t)}\\
&= \alpha(Q_\tau^\star - \tau\log\overline{\xi}^{(t)}) + (1-\alpha)(Q_\tau^\star - \overline{Q}_\tau^{(t)}) + (1-\alpha)(\overline{Q}_\tau^{(t)} - \widehat{Q}_\tau^{(t)}),
\end{aligned}
$$

which gives

$$\big\|Q_\tau^\star - \tau\log\overline{\xi}^{(t+1)}\big\|_\infty \le \alpha\big\|Q_\tau^\star - \tau\log\overline{\xi}^{(t)}\big\|_\infty + (1-\alpha)\big\|Q_\tau^\star - \overline{Q}_\tau^{(t)}\big\|_\infty + (1-\alpha)\big\|\overline{Q}_\tau^{(t)} - \widehat{Q}_\tau^{(t)}\big\|_\infty. \tag{168}$$

Note that we can upper bound $\big\|\overline{Q}_\tau^{(t)} - \widehat{Q}_\tau^{(t)}\big\|_\infty$ by

$$\begin{aligned}
\big\|\overline{Q}_\tau^{(t)} - \widehat{Q}_\tau^{(t)}\big\|_\infty &= \left\|\frac{1}{N}\sum_{n=1}^N Q_{\tau,n}^{\pi_n^{(t)}} - \frac{1}{N}\sum_{n=1}^N Q_{\tau,n}^{\overline{\pi}^{(t)}}\right\|_\infty \\
&\le \frac{1}{N}\sum_{n=1}^N \big\|Q_{\tau,n}^{\pi_n^{(t)}} - Q_{\tau,n}^{\overline{\pi}^{(t)}}\big\|_\infty \\
&\le \frac{M}{N}\sum_{n=1}^N \big\|\log\xi_n^{(t)} - \log\overline{\xi}^{(t)}\big\|_\infty \le M\big\|u^{(t)}\big\|_\infty. \tag{169}
\end{aligned}$$

The last step is due to $\big|\log\xi_n^{(t)}(s,a) - \log\overline{\xi}^{(t)}(s,a)\big| \le u^{(t)}(s,a)$, while the penultimate step results from writing

$$\begin{aligned}
\overline{\pi}^{(t)}(\cdot|s) &= \operatorname{softmax}\left(\log\overline{\xi}^{(t)}(s,\cdot)\right), \\
\pi_n^{(t)}(\cdot|s) &= \operatorname{softmax}\left(\log\xi_n^{(t)}(s,\cdot)\right),
\end{aligned}$$

and applying the following lemma.

**Lemma F.3** (Lipschitz constant of soft Q-function). *Assume that $r(s,a) \in [0,1], \forall(s,a) \in \mathcal{S}\times\mathcal{A}$ and $\tau \ge 0$. For any $\theta, \theta' \in \mathbb{R}^{|\mathcal{S}||\mathcal{A}|}$, we have*

$$\big\|Q_\tau^{\pi_{\theta'}} - Q_\tau^{\pi_\theta}\big\|_\infty \le \underbrace{\frac{1+\gamma+2\tau(1-\gamma)\log|\mathcal{A}|}{(1-\gamma)^2}\cdot\gamma}_{=:M}\big\|\theta' - \theta\big\|_\infty. \tag{170}$$

Plugging (169) into (168) gives

$$\big\|Q_\tau^\star - \tau\log\overline{\xi}^{(t+1)}\big\|_\infty \le \alpha\big\|Q_\tau^\star - \tau\log\overline{\xi}^{(t)}\big\|_\infty + (1-\alpha)\big\|Q_\tau^\star - \overline{Q}_\tau^{(t)}\big\|_\infty + (1-\alpha)M\big\|u^{(t)}\big\|_\infty. \tag{171}$$

**Step 4: bound $\big\|\boldsymbol{Q}_\tau^{(t+1)}(s,a) - \boldsymbol{Q}_\tau^{(t)}(s,a)\big\|_2$.**

Let $w^{(t)} : \mathcal{S}\times\mathcal{A} \to \mathbb{R}$ be defined as

$$\forall(s,a)\in\mathcal{S}\times\mathcal{A}: \quad w^{(t)}(s,a) := \big\|\log\boldsymbol{\xi}^{(t+1)}(s,a) - \log\boldsymbol{\xi}^{(t)}(s,a) - (1-\alpha)V_\tau^\star(s)\mathbf{1}_N/\tau\big\|_2. \tag{172}$$

Again, we treat $w^{(t)}$ as vectors in $\mathbb{R}^{|\mathcal{S}||\mathcal{A}|}$ whenever it is clear from context. For any $(s,a)\in\mathcal{S}\times\mathcal{A}$ and $n\in[N]$, by Lemma F.3 it follows that

$$\begin{aligned}
\big|Q_{\tau,n}^{\pi_n^{(t+1)}}(s,a) - Q_{\tau,n}^{\pi_n^{(t)}}(s,a)\big| &\le M\max_{s\in\mathcal{S}}\big\|\log\xi_n^{(t+1)}(s,\cdot) - \log\xi_n^{(t)}(s,\cdot) - (1-\alpha)V_\tau^\star(s)\mathbf{1}_{|\mathcal{A}|}/\tau\big\|_\infty \\
&\le M\max_{s\in\mathcal{S}}\max_{a\in\mathcal{A}}w^{(t)}(s,a) \le M\big\|w^{(t)}\big\|_\infty, \tag{173}
\end{aligned}$$

and consequently

$$\big\|\boldsymbol{Q}_\tau^{(t+1)}(s,a) - \boldsymbol{Q}_\tau^{(t)}(s,a)\big\|_2 \le M\sqrt{N}\big\|w^{(t)}\big\|_\infty. \tag{174}$$

It boils down to control $\big\|w^{(t)}\big\|_\infty$. To do so, we first note that for each $(s,a)\in\mathcal{S}\times\mathcal{A}$, we have

$$\begin{aligned}
&w^{(t)}(s,a) \\
&= \big\|\boldsymbol{W}\left(\alpha\log\boldsymbol{\xi}^{(t)}(s,a) + (1-\alpha)\boldsymbol{T}^{(t)}(s,a)/\tau\right) - \log\boldsymbol{\xi}^{(t)}(s,a) - (1-\alpha)V_\tau^\star(s)\mathbf{1}_N/\tau\big\|_2 \\
&\overset{(a)}{=} \big\|\alpha(\boldsymbol{W}-\boldsymbol{I}_N)\left(\log\boldsymbol{\xi}^{(t)}(s,a) - \log\overline{\xi}^{(t)}(s,a)\mathbf{1}_N\right) + (1-\alpha)\left(\boldsymbol{W}\boldsymbol{T}^{(t)}(s,a)/\tau - \log\boldsymbol{\xi}^{(t)}(s,a) - V_\tau^\star(s)\mathbf{1}_N/\tau\right)\big\|_2 \\
&\overset{(b)}{\le} 2\alpha\big\|\log\boldsymbol{\xi}^{(t)}(s,a) - \log\overline{\xi}^{(t)}(s,a)\mathbf{1}_N\big\|_2 + \frac{1-\alpha}{\tau}\big\|\boldsymbol{W}\boldsymbol{T}^{(t)}(s,a) - \tau\log\boldsymbol{\xi}^{(t)}(s,a) - V_\tau^\star(s)\mathbf{1}_N\big\|_2
\end{aligned}$$
$$\tag{175}$$

where (a) is due to the doubly stochasticity property of $\boldsymbol{W}$ and (b) is from the fact $\|\boldsymbol{W} - \boldsymbol{I}_N\|_2 \leq 2$. We further bound the second term as follows:

$$
\begin{aligned}
&\left\|\boldsymbol{W}\boldsymbol{T}^{(t)}(s,a) - \tau\log\boldsymbol{\xi}^{(t)}(s,a) - V_\tau^\star(s)\boldsymbol{1}_N\right\|_2 \\
&= \left\|\boldsymbol{W}\boldsymbol{T}^{(t)}(s,a) - \tau\log\boldsymbol{\xi}^{(t)}(s,a) - \left(Q_\tau^\star(s,a) - \tau\log\pi_\tau^\star(a|s)\right)\boldsymbol{1}_N\right\|_2 \\
&\leq \left\|\boldsymbol{W}\boldsymbol{T}^{(t)}(s,a) - Q_\tau^\star(s,a)\boldsymbol{1}_N\right\|_2 + \tau\left\|\log\boldsymbol{\xi}^{(t)}(s,a) - \log\pi_\tau^\star(a|s)\boldsymbol{1}_N\right\|_2 \\
&\leq \left\|\boldsymbol{W}\boldsymbol{T}^{(t)}(s,a) - \widehat{Q}_\tau(s,a)\boldsymbol{1}_N\right\|_2 + \left\|\widehat{Q}_\tau(s,a)\boldsymbol{1}_N - Q_\tau^\star(s,a)\boldsymbol{1}_N\right\|_2 \\
&\quad + \tau\left\|\log\boldsymbol{\xi}^{(t)}(s,a) - \log\overline{\pi}^{(t)}(a|s)\boldsymbol{1}_N\right\|_2 + \tau\left\|\log\overline{\pi}^{(t)}(a|s)\boldsymbol{1}_N - \log\pi_\tau^\star(a|s)\boldsymbol{1}_N\right\|_2 \\
&= \sigma\left\|\boldsymbol{T}^{(t)}(s,a) - \widehat{Q}_\tau^{(t)}(s,a)\boldsymbol{1}_N\right\|_2 + \sqrt{N}\left|\widehat{Q}_\tau^{(t)}(s,a) - Q_\tau^\star(s,a)\right| \\
&\quad + \tau\left\|\log\boldsymbol{\xi}^{(t)}(s,a) - \log\overline{\pi}^{(t)}(a|s)\boldsymbol{1}_N\right\|_2 + \tau\sqrt{N}\left|\log\overline{\pi}^{(t)}(a|s) - \log\pi_\tau^\star(a|s)\right|. \quad (176)
\end{aligned}
$$

Here, the first step results from the following relation established in [NNXS17]:

$$\forall(s,a)\in\mathcal{S}\times\mathcal{A}: \quad V_\tau^\star(s) = -\tau\log\pi_\tau^\star(a|s) + Q_\tau^\star(s,a), \quad (177)$$

which also leads to

$$\left\|\log\overline{\pi}^{(t)} - \log\pi_\tau^\star\right\|_\infty \leq \frac{2}{\tau}\left\|Q_\tau^\star - \tau\log\overline{\xi}^{(t)}\right\|_\infty \quad (178)$$

by Lemma F.2. For the remaining terms in (176), we have

$$\left|\widehat{Q}_\tau^{(t)}(s,a) - Q_\tau^\star(s,a)\right| \leq \left\|\widehat{Q}_\tau^{(t)} - \overline{Q}_\tau^{(t)}\right\|_\infty + \left\|\overline{Q}_\tau^{(t)} - Q_\tau^\star\right\|_\infty, \quad (179)$$

and

$$
\begin{aligned}
\left\|\log\boldsymbol{\xi}^{(t)}(s,a) - \log\overline{\pi}^{(t)}(a|s)\boldsymbol{1}_N\right\|_2 &= \sqrt{\sum_{n=1}^N \left(\log\xi_n^{(t)}(s,a) - \log\overline{\pi}^{(t)}(a|s)\right)^2} \\
&\leq \sqrt{\sum_{n=1}^N 2\left\|\log\xi_n^{(t)} - \log\overline{\xi}^{(t)}\right\|_\infty^2} \\
&\leq \sqrt{\sum_{n=1}^N 2\left\|u^{(t)}\right\|_\infty^2} = \sqrt{2N}\left\|u^{(t)}\right\|_\infty, \quad (180)
\end{aligned}
$$

where the first inequality again results from Lemma F.2. Plugging (178), (179), (180) into (176) and using the definition of $u^{(t)}, v^{(t)}$, we arrive at

$$
\begin{aligned}
w^{(t)}(s,a) &\leq \left(2\alpha + (1-\alpha)\cdot\sqrt{2N}\right)\left\|u^{(t)}\right\|_\infty + \frac{1-\alpha}{\tau}\left\|v^{(t)}\right\|_\infty + \frac{1-\alpha}{\tau}\cdot\sqrt{N}\left(\left\|\widehat{Q}_\tau^{(t)} - \overline{Q}_\tau^{(t)}\right\|_\infty + \left\|\overline{Q}_\tau^{(t)} - Q_\tau^\star\right\|_\infty\right) \\
&\quad + \frac{1-\alpha}{\tau}\cdot 2\sqrt{N}\left\|Q_\tau^\star - \tau\log\overline{\xi}^{(t)}\right\|_\infty.
\end{aligned}
$$

Using previous display, we can write (174) as

$$
\begin{aligned}
&\left\|\boldsymbol{Q}_\tau^{(t+1)}(s,a) - \boldsymbol{Q}_\tau^{(t)}(s,a)\right\|_2 \\
&\leq M\sqrt{N}\bigg\{\left(2\alpha + (1-\alpha)\cdot\sqrt{2N}\right)\left\|u^{(t)}\right\|_\infty + \frac{1-\alpha}{\tau}\sigma\left\|v^{(t)}\right\|_\infty \\
&\quad + \frac{1-\alpha}{\tau}\cdot\sqrt{N}\left(M\left\|u^{(t)}\right\|_\infty + \left\|\overline{Q}_\tau^{(t)} - Q_\tau^\star\right\|_\infty\right) + \frac{1-\alpha}{\tau}\cdot 2\sqrt{N}\left\|Q_\tau^\star - \tau\log\overline{\xi}^{(t)}\right\|_\infty\bigg\}. \\
&\quad\quad (181)
\end{aligned}
$$

Combining (167) with the above expression (181), we get

$$
\begin{aligned}
\left\|v^{(t+1)}\right\|_\infty &\leq \sigma\left(1 + \frac{\eta M\sqrt{N}}{1-\gamma}\sigma\right)\left\|v^{(t)}\right\|_\infty + \sigma M\sqrt{N}\bigg\{\left(2\alpha + (1-\alpha)\cdot\sqrt{2N} + \frac{1-\alpha}{\tau}\cdot\sqrt{N}M\right)\left\|u^{(t)}\right\|_\infty \\
&\quad + \frac{1-\alpha}{\tau}\cdot\sqrt{N}\left\|\overline{Q}_\tau^{(t)} - Q_\tau^\star\right\|_\infty + \frac{1-\alpha}{\tau}\cdot 2\sqrt{N}\left\|Q_\tau^\star - \tau\log\overline{\xi}^{(t)}\right\|_\infty\bigg\}. \quad (182)
\end{aligned}
$$

**Step 5: bound** $\left\|\overline{Q}_\tau^{(t+1)} - Q_\tau^\star\right\|_\infty$. For any state-action pair $(s,a) \in \mathcal{S} \times \mathcal{A}$, we observe that

$$
Q_\tau^\star(s,a) - \overline{Q}_\tau^{(t+1)}(s,a)
$$

$$
= r(s,a) + \gamma \mathop{\mathbb{E}}_{s' \sim P(\cdot|s,a)} [V_\tau^\star(s')] - \left( r(s,a) + \gamma \mathop{\mathbb{E}}_{s' \sim P(\cdot|s,a)} \left[ V_\tau^{\overline{\pi}^{(t+1)}}(s') \right] \right)
$$

$$
= \gamma \mathop{\mathbb{E}}_{s' \sim P(\cdot|s,a)} \left[ \tau \log \left( \left\| \exp\left( \frac{Q_\tau^\star(s',\cdot)}{\tau} \right) \right\|_1 \right) \right] - \gamma \mathop{\mathbb{E}}_{\substack{s' \sim P(\cdot|s,a), \\ a' \sim \overline{\pi}^{(t+1)}(\cdot|s')}} \left[ \overline{Q}_\tau^{(t+1)}(s',a') - \tau \log \overline{\pi}^{(t+1)}(a'|s') \right] ,
$$

$$\tag{183}$$

where the first step invokes the definition of $Q_\tau$ (cf. (6a)), and the second step is due to the following expression of $V_\tau^\star$ established in [NNXS17]:

$$
V_\tau^\star(s) = \tau \log \left( \left\| \exp\left( \frac{Q_\tau^\star(s,\cdot)}{\tau} \right) \right\|_1 \right) . \tag{184}
$$

To continue, note that by (162) and (41b) we have

$$
\log \overline{\pi}^{(t+1)}(a|s) = \log \overline{\xi}^{(t+1)}(s,a) - \log \left( \left\| \overline{\xi}^{(t+1)}(s,\cdot) \right\|_1 \right)
$$

$$
= \alpha \log \overline{\xi}^{(t)}(s,a) + (1-\alpha) \frac{\widehat{Q}_\tau^{(t)}(s,a)}{\tau} - \log \left( \left\| \overline{\xi}^{(t+1)}(s,\cdot) \right\|_1 \right) . \tag{185}
$$

Plugging (185) into (183) and (181) establishes the bounds on

$$
Q_\tau^\star(s,a) - \overline{Q}_\tau^{(t+1)}(s,a) = \gamma \mathop{\mathbb{E}}_{s' \sim P(\cdot|s,a)} \left[ \tau \log \left( \left\| \exp\left( \frac{Q_\tau^\star(s',\cdot)}{\tau} \right) \right\|_1 \right) - \tau \log \left( \left\| \overline{\xi}^{(t+1)}(s',\cdot) \right\|_1 \right) \right]
$$

$$
- \gamma \mathop{\mathbb{E}}_{\substack{s' \sim P(\cdot|s,a), \\ a' \sim \overline{\pi}^{(t+1)}(\cdot|s')}} \left[ \overline{Q}_\tau^{(t+1)}(s',a') - \tau \underbrace{\left( \alpha \log \overline{\xi}^{(t)}(s',a') + (1-\alpha) \frac{\widehat{Q}_\tau^{(t)}(s',a')}{\tau} \right)}_{= \log \overline{\xi}^{(t+1)}(s',a')} \right]
$$

$$\tag{186}$$

for any $(s,a) \in \mathcal{S} \times \mathcal{A}$. In view of property (164), the first term on the right-hand side of (186) can be bounded by

$$
\tau \log \left( \left\| \exp\left( \frac{Q_\tau^\star(s',\cdot)}{\tau} \right) \right\|_1 \right) - \tau \log \left( \left\| \overline{\xi}^{(t+1)}(s',\cdot) \right\|_1 \right) \leq \left\| Q_\tau^\star - \tau \log \overline{\xi}^{(t+1)} \right\|_\infty .
$$

Plugging the above expression into (186), we have

$$
0 \leq Q_\tau^\star(s,a) - \overline{Q}_\tau^{(t+1)}(s,a) \leq \gamma \left\| Q_\tau^\star - \tau \log \overline{\xi}^{(t+1)} \right\|_\infty - \gamma \min_{s,a} \left( \overline{Q}_\tau^{(t+1)}(s,a) - \tau \log \overline{\xi}^{(t+1)}(s,a) \right) ,
$$

which gives

$$
\left\| Q_\tau^\star - \overline{Q}_\tau^{(t+1)} \right\|_\infty \leq \gamma \left\| Q_\tau^\star - \tau \log \overline{\xi}^{(t+1)} \right\|_\infty + \gamma \max \left\{ 0, - \min_{s,a} \left( \overline{Q}_\tau^{(t+1)}(s,a) - \tau \log \overline{\xi}^{(t+1)}(s,a) \right) \right\} .
$$

$$\tag{187}$$

Plugging the above inequality into (171) and (182) establishes the bounds on $\Omega_3^{(t+1)}$ and $\Omega_2^{(t+1)}$ in (49), respectively. **Step 6: bound** $-\min_{s,a} \left( \overline{Q}_\tau^{(t+1)}(s,a) - \tau \log \overline{\xi}^{(t+1)}(s,a) \right)$. We need the following lemma which is adapted from Lemma 1 in [CCC$^+$22b]:

**Lemma F.4** (Performance improvement of FedNPG with entropy regularization). *Suppose* $0 < \eta \leq (1-\gamma)/\tau$. *For any state-action pair* $(s_0, a_0) \in \mathcal{S} \times \mathcal{A}$, *one has*

$$
\overline{V}_\tau^{(t+1)}(s_0) - \overline{V}_\tau^{(t)}(s_0) \geq \frac{1}{\eta} \mathop{\mathbb{E}}_{s \sim d_{s_0}^{\overline{\pi}^{(t+1)}}} \left[ \alpha \mathsf{KL} \left( \overline{\pi}^{(t+1)}(\cdot|s_0) \,\|\, \overline{\pi}^{(t)}(\cdot|s_0) \right) + \mathsf{KL} \left( \overline{\pi}^{(t)}(\cdot|s_0) \,\|\, \overline{\pi}^{(t+1)}(\cdot|s_0) \right) \right]
$$

$$
- \frac{2}{1-\gamma} \left\| \widehat{Q}_\tau^{(t)} - \overline{Q}_\tau^{(t)} \right\|_\infty , \tag{188}
$$

$$
\overline{Q}_\tau^{(t+1)}(s_0, a_0) - \overline{Q}_\tau^{(t)}(s_0, a_0) \geq - \frac{2\gamma}{1-\gamma} \left\| \widehat{Q}_\tau^{(t)} - \overline{Q}_\tau^{(t)} \right\|_\infty . \tag{189}
$$

*Proof.* See Appendix G.3. □

Using (189), we have

$$\overline{Q}_\tau^{(t+1)}(s,a) - \tau\left(\alpha\log\overline{\xi}^{(t)}(s,a) + (1-\alpha)\frac{\widehat{Q}_\tau^{(t)}(s,a)}{\tau}\right)$$

$$\geq \overline{Q}_\tau^{(t)}(s,a) - \tau\left(\alpha\log\overline{\xi}^{(t)}(s,a) + (1-\alpha)\frac{\widehat{Q}_\tau^{(t)}(s,a)}{\tau}\right) - \frac{2\gamma}{1-\gamma}\|\widehat{Q}_\tau^{(t)} - \overline{Q}_\tau^{(t)}\|_\infty$$

$$\geq \alpha\left(\overline{Q}_\tau^{(t)}(s,a) - \tau\log\overline{\xi}^{(t)}(s,a)\right) - \frac{2\gamma+\eta\tau}{1-\gamma}\|\widehat{Q}_\tau^{(t)} - \overline{Q}_\tau^{(t)}\|_\infty, \tag{190}$$

which gives

$$-\min_{s,a}\left(\overline{Q}_\tau^{(t+1)}(s,a) - \tau\log\overline{\xi}^{(t+1)}(s,a)\right)$$

$$\leq -\alpha\min_{s,a}\left(\overline{Q}_\tau^{(t)}(s,a) - \tau\log\overline{\xi}^{(t)}(s,a)\right) + \frac{2\gamma+\eta\tau}{1-\gamma}M\|u^{(t)}\|_\infty$$

$$\leq \alpha\max\left\{0, \min_{s,a}\left(\overline{Q}_\tau^{(t)}(s,a) - \tau\log\overline{\xi}^{(t)}(s,a)\right)\right\} + \frac{2\gamma+\eta\tau}{1-\gamma}M\|u^{(t)}\|_\infty. \tag{191}$$

This establishes the bounds on $\Omega_4^{(t+1)}$ in (49).

### F.2  Proof of Lemma D.3

Let $f(\lambda)$ denote the characteristic function. In view of some direct calculations, we obtain

$$f(\lambda) = (\lambda - \alpha)\Bigg\{\underbrace{(\lambda - \sigma\alpha)(\lambda - \sigma(1+\sigma b\eta))(\lambda - (1-\alpha)\gamma - \alpha)}_{=:f_0(\lambda)}$$

$$-\frac{\eta\sigma^2}{1-\gamma}\underbrace{[S(\lambda - (1-\alpha)\gamma - \alpha) + \gamma cdM\eta + (1-\alpha)(2+\gamma)Mc\eta]}_{=:f_1(\lambda)}\Bigg\} \tag{192}$$

$$-\frac{\tau\eta^3\gamma}{(1-\gamma)^2}\cdot 2cdM\sigma^2,$$

where, for the notation simplicity, we let

$$b := \frac{M\sqrt{N}}{1-\gamma}, \tag{193a}$$

$$c := \frac{MN}{1-\gamma} = \sqrt{N}b, \tag{193b}$$

$$d := \frac{2\gamma+\eta\tau}{1-\gamma}. \tag{193c}$$

Note that among all these new notation we introduce, $S, d$ are dependent of $\eta$. To decouple the dependence, we give their upper bounds as follows

$$d_0 := \frac{1+\gamma}{1-\gamma} \geq d, \tag{194}$$

$$S_0 := M\sqrt{N}\left(2 + \sqrt{2N} + \frac{M\sqrt{N}}{\tau}\right) \geq S, \tag{195}$$

where (194) follows from $\eta \leq (1-\gamma)/\tau$, and (195) uses the fact that $\alpha \leq 1$ and $1-\alpha \leq 1$.

Let

$$\lambda^\star := \max\left\{\frac{3+\sigma}{4}, \frac{1+(1-\alpha)\gamma+\alpha}{2}\right\}. \tag{196}$$

Since $\boldsymbol{A}(\rho)$ is a nonnegative matrix, by Perron-Frobenius Theorem (see [HJ12], Theorem 8.3.1), $\rho(\eta)$ is an eigenvalue of $\boldsymbol{A}(\rho)$. So to verify (55), it suffices to show that $f(\lambda) > 0$ for any $\lambda \in [\lambda^\star, \infty)$. To do so, in the following we first show that $f(\lambda^\star) > 0$, and then we prove that $f$ is non-decreasing on $[\lambda^\star, \infty)$.

- *Showing $f(\lambda^\star) > 0$.* We first lower bound $f_0(\lambda^\star)$. Since $\lambda^\star \geq \frac{3+\sigma}{4}$, we have

$$\lambda^\star - \sigma(1 + \sigma b\eta) \geq \frac{1 - \sigma}{4}, \tag{197}$$

and from $\lambda^\star \geq \frac{1+(1-\alpha)\gamma+\alpha}{2}$ we deduce

$$\lambda^\star - (1 - \alpha)\gamma - \alpha \geq \frac{(1 - \gamma)(1 - \alpha)}{2} \tag{198}$$

and

$$\lambda^\star > \frac{1 + \alpha}{2}, \tag{199}$$

which gives

$$\lambda^\star - \sigma\alpha \geq \frac{1 + \alpha}{2} - \sigma\alpha. \tag{200}$$

Combining (200), (197), (198), we have that

$$f_0(\lambda^\star) \geq \frac{1 - \sigma}{8}\left(\frac{1 + \alpha}{2} - \sigma\alpha\right)\eta\tau. \tag{201}$$

To continue, we upper bound $f_1(\lambda^\star)$ as follows.

$$f_1(\lambda^\star) \leq S\tau\eta + \gamma cdM\eta + \frac{2 + \gamma}{1 - \gamma}cM\tau\eta^2$$
$$= \eta\left(\tau\left(S + \frac{2 + \gamma}{1 - \gamma}Mc\eta\right) + \gamma cdM\right). \tag{202}$$

Plugging (201),(202) into (192) and using (199), we have

$$f(\lambda^\star) > \frac{1 - \alpha}{2}\left(f_0(\lambda^\star) - \frac{\eta\sigma^2}{1 - \gamma}f_1(\lambda^\star)\right) - \frac{\tau\eta^3\gamma}{(1 - \gamma)^2} \cdot 2cdM\sigma^2$$
$$\geq \frac{\tau\eta^2}{2(1 - \gamma)}\left[\frac{1 - \sigma}{8}\tau\left(1 - \sigma + (1 - \alpha)(\sigma - \frac{1}{2})\right) - \frac{\eta\sigma^2}{1 - \gamma}\left(\tau\left(S + \frac{2 + \gamma}{1 - \gamma}Mc\eta\right) + 5\gamma cdM\right)\right]$$
$$= \frac{\tau\eta^2}{2(1 - \gamma)}\left[\frac{(1 - \sigma)^2}{8}\tau - \frac{\eta}{1 - \gamma}\left(S\tau\sigma^2 + \frac{2 + \gamma}{1 - \gamma}Mc\sigma^2\tau\eta + \tau^2\left(\frac{1}{2} - \sigma^2\right) \cdot \frac{1 - \sigma}{8} + 5\gamma cdM\sigma^2\right)\right]$$
$$\geq \frac{\tau\eta^2}{2(1 - \gamma)}\left[\frac{(1 - \sigma)^2}{8}\tau - \frac{\eta}{1 - \gamma}\left(S_0\tau\sigma^2 + \frac{(1 - \sigma)^2}{16}\tau^2 + (2 + \gamma + 5\gamma d_0)cM\sigma^2\right)\right] \geq 0,$$

where the penultimate inequality uses $\frac{1}{2} - \sigma \leq \frac{1-\sigma}{2}$, and the last inequality follows from the definition of $\zeta$ (cf. (53)).

- *Proving $f$ is non-decreasing on $[\lambda^\star, \infty)$.* Note that

$$\eta \leq \zeta \leq \frac{(1 - \gamma)(1 - \sigma)^2}{8S_0\sigma^2},$$

thus we have

$$\forall \lambda \geq \lambda^\star: \quad f_0'(\lambda) - \frac{\eta\sigma^2}{1 - \gamma}f_1'(\lambda) \geq (\lambda - \sigma\alpha)(\lambda - \sigma(1 + \sigma b\eta)) - \frac{\eta}{1 - \gamma}S\sigma^2 \geq 0,$$

which indicates that $f_0 - f_1$ is non-decreasing on $[\lambda^\star, \infty)$. Therefore, $f$ is non-decreasing on $[\lambda^\star, \infty)$.

## F.3 Proof of Lemma D.6

Note that bounding $u^{(t+1)}(s, a)$ is identical to the proof in Appendix F.1 and shall be omitted. The rest of the proof also follows closely that of Lemma D.2, and we only highlight the differences due to approximation error for simplicity.

**Step 2: bound** $v^{(t+1)}(s,a) = \big\|\boldsymbol{T}^{(t+1)}(s,a) - \widehat{q}_\tau^{(t+1)}(s,a)\mathbf{1}_N\big\|_2$**. Let** $\boldsymbol{q}_\tau^{(t)} := \left(q_{\tau,1}^{\pi_1^{(t)}}, \cdots, q_{\tau,N}^{\pi_N^{(t)}}\right)^\top$**.**
Similar to (167) we have

$$
\begin{aligned}
&\big\|\boldsymbol{T}^{(t+1)}(s,a) - \widehat{q}_\tau^{(t+1)}(s,a)\mathbf{1}_N\big\|_2 \\
&\leq \sigma\big\|\boldsymbol{T}^{(t)}(s,a) - \widehat{q}_\tau^{(t)}(s,a)\mathbf{1}_N\big\|_2 + \sigma\big\|\boldsymbol{q}_\tau^{(t+1)}(s,a) - \boldsymbol{q}_\tau^{(t)}(s,a)\big\|_2 \\
&\leq \sigma\big\|\boldsymbol{T}^{(t)}(s,a) - \widehat{q}_\tau^{(t)}(s,a)\mathbf{1}_N\big\|_2 + \sigma\big\|\boldsymbol{Q}_\tau^{(t+1)}(s,a) - \boldsymbol{Q}_\tau^{(t)}(s,a)\big\|_2 + 2\sigma\,\|e\|_2 .
\end{aligned}
\tag{203}
$$

**Step 3: bound** $\big\|Q_\tau^\star - \tau\log\overline{\xi}^{(t+1)}\big\|_\infty$**.** In the context of inexact updates, (168) writes

$$
\big\|Q_\tau^\star - \tau\log\overline{\xi}^{(t+1)}\big\|_\infty \leq \alpha\big\|Q_\tau^\star - \tau\log\overline{\xi}^{(t)}\big\|_\infty + (1-\alpha)\big\|Q_\tau^\star - \overline{Q}_\tau^{(t)}\big\|_\infty + (1-\alpha)\big\|\overline{Q}_\tau^{(t)} - \widehat{q}_\tau^{(t)}\big\|_\infty .
$$

For the last term, following a similar argument in (169) leads to

$$
\begin{aligned}
\big\|\overline{Q}_\tau^{(t)} - \widehat{q}_\tau^{(t)}\big\|_\infty &= \left\|\frac{1}{N}\sum_{n=1}^N Q_{\tau,n}^{\pi_n^{(t)}} - \frac{1}{N}\sum_{n=1}^N Q_{\tau,n}^{\overline{\pi}^{(t)}}\right\|_\infty + \left\|\frac{1}{N}\sum_{n=1}^N \left(Q_{\tau,n}^{\pi_n^{(t)}} - q_{\tau,n}^{\pi_n^{(t)}}\right)\right\|_\infty \\
&\leq M\cdot\frac{1}{N}\sum_{n=1}^N \big\|\log\xi_n^{(t)} - \log\overline{\xi}^{(t)}\big\|_\infty + \frac{1}{N}\sum_{n=1}^N e_n \\
&\leq M\big\|u^{(t)}\big\|_\infty + \|e\|_\infty .
\end{aligned}
$$

Combining the above two inequalities, we obtain

$$
\big\|Q_\tau^\star - \tau\log\overline{\xi}^{(t+1)}\big\|_\infty \leq \alpha\big\|Q_\tau^\star - \tau\log\overline{\xi}^{(t)}\big\|_\infty + (1-\alpha)\big\|Q_\tau^\star - \overline{Q}_\tau^{(t)}\big\|_\infty + (1-\alpha)\left(M\big\|u^{(t)}\big\|_\infty + \|e\|_\infty\right) .
\tag{204}
$$

**Step 4: bound** $\big\|\boldsymbol{Q}_\tau^{(t+1)}(s,a) - \boldsymbol{Q}_\tau^{(t)}(s,a)\big\|_2$**.** We remark that the bound established in (174) still holds in the inexact setting, with the same definition for $w^{(t)}$:

$$
\big\|\boldsymbol{Q}_\tau^{(t+1)}(s,a) - \boldsymbol{Q}_\tau^{(t)}(s,a)\big\|_2 \leq M\sqrt{N}\,\big\|w^{(t)}\big\|_\infty .
\tag{205}
$$

To deal with the approximation error, we rewrite (176) as

$$
\begin{aligned}
&\big\|\boldsymbol{W}\boldsymbol{T}^{(t)}(s,a) - \tau\log\boldsymbol{\xi}^{(t)}(s,a) - V_\tau^\star(s)\mathbf{1}_N\big\|_2 \\
&= \big\|\boldsymbol{W}\boldsymbol{T}^{(t)}(s,a) - \tau\log\boldsymbol{\xi}^{(t)}(s,a) - \left(Q_\tau^\star(s,a) - \tau\log\pi_\tau^\star(a|s)\right)\mathbf{1}_N\big\|_2 \\
&\leq \big\|\boldsymbol{W}\boldsymbol{T}^{(t)}(s,a) - Q_\tau^\star(s,a)\mathbf{1}_N\big\|_2 + \tau\big\|\log\boldsymbol{\xi}^{(t)}(s,a) - \log\pi_\tau^\star(a|s)\mathbf{1}_N\big\|_2 \\
&\leq \big\|\boldsymbol{W}\boldsymbol{T}^{(t)}(s,a) - \widehat{q}_\tau(s,a)\mathbf{1}_N\big\|_2 + \big\|\widehat{q}_\tau(s,a)\mathbf{1}_N - Q_\tau^\star(s,a)\mathbf{1}_N\big\|_2 \\
&\quad + \tau\big\|\log\boldsymbol{\xi}^{(t)}(s,a) - \log\overline{\pi}^{(t)}(a|s)\mathbf{1}_N\big\|_2 + \tau\big\|\log\overline{\pi}^{(t)}(a|s)\mathbf{1}_N - \log\pi_\tau^\star(a|s)\mathbf{1}_N\big\|_2 \\
&\leq \sigma\big\|\boldsymbol{T}^{(t)}(s,a) - \widehat{q}_\tau^{(t)}(s,a)\mathbf{1}_N\big\|_2 + \sqrt{N}\big|\widehat{q}_\tau^{(t)}(s,a) - Q_\tau^\star(s,a)\big| \\
&\quad + \tau\big\|\log\boldsymbol{\xi}^{(t)}(s,a) - \log\overline{\pi}^{(t)}(a|s)\mathbf{1}\big\|_2 + \tau\sqrt{N}\big|\log\overline{\pi}^{(t)}(a|s) - \log\pi_\tau^\star(a|s)\big| ,
\end{aligned}
\tag{206}
$$

where the second term can be upper-bounded by

$$
\begin{aligned}
\big|\widehat{q}_\tau^{(t)}(s,a) - Q_\tau^\star(s,a)\big| &\leq \big\|\widehat{Q}_\tau^{(t)} - \overline{Q}_\tau^{(t)}\big\|_\infty + \big\|\overline{Q}_\tau^{(t)} - Q_\tau^\star\big\|_\infty + \big\|\widehat{q}_\tau^{(t)}(s,a) - \widehat{Q}_\tau^{(t)}(s,a)\big\|_\infty \\
&\leq \big\|\widehat{Q}_\tau^{(t)} - \overline{Q}_\tau^{(t)}\big\|_\infty + \big\|\overline{Q}_\tau^{(t)} - Q_\tau^\star\big\|_\infty + \|e\|_\infty .
\end{aligned}
\tag{207}
$$

Combining (207), (206) and the established bounds in (175), (178), (180) leads to

$$
\begin{aligned}
w^{(t)}(s,a) &\leq \left(2\alpha + (1-\alpha)\cdot\sqrt{2N}\right)\big\|u^{(t)}\big\|_\infty + \frac{1-\alpha}{\tau}\big\|v^{(t)}\big\|_\infty \\
&\quad + \frac{1-\alpha}{\tau}\cdot\sqrt{N}\left(\big\|\widehat{Q}_\tau^{(t)} - \overline{Q}_\tau^{(t)}\big\|_\infty + \big\|\overline{Q}_\tau^{(t)} - Q_\tau^\star\big\|_\infty + \|e\|_\infty\right) + \frac{1-\alpha}{\tau}\cdot2\sqrt{N}\big\|Q_\tau^\star - \tau\log\overline{\xi}^{(t)}\big\|_\infty .
\end{aligned}
$$

Combining the above inequality with (205) and (203) gives

$$\left\|v^{(t+1)}\right\|_\infty \leq \sigma\left(1 + \frac{\eta M\sqrt{N}}{1-\gamma}\sigma\right)\left\|v^{(t)}\right\|_\infty + \sigma M\sqrt{N}\Bigg\{\left(2\alpha + (1-\alpha)\cdot\sqrt{2N} + \frac{1-\alpha}{\tau}\cdot\sqrt{N}M\right)\left\|u^{(t)}\right\|_\infty$$
$$+ \frac{1-\alpha}{\tau}\cdot\sqrt{N}\left(\left\|\overline{Q}_\tau^{(t)} - Q_\tau^\star\right\|_\infty + \left\|e\right\|_\infty\right) + \frac{1-\alpha}{\tau}\cdot 2\sqrt{N}\left\|Q_\tau^\star - \tau\log\overline{\xi}^{(t)}\right\|_\infty\Bigg\} + 2\sigma\sqrt{N}\left\|e\right\|_\infty .$$
$$(208)$$

**Step 5: bound** $\left\|\overline{Q}_\tau^{(t+1)} - Q_\tau^\star\right\|_\infty$. It is straightforward to verify that (187) applies to the inexact updates as well:

$$\left\|Q_\tau^\star - \overline{Q}_\tau^{(t+1)}\right\|_\infty \leq \gamma\left\|Q_\tau^\star - \tau\log\overline{\xi}^{(t+1)}\right\|_\infty + \gamma\left(-\min_{s,a}\left(\overline{Q}_\tau^{(t+1)}(s,a) - \tau\log\overline{\xi}^{(t+1)}(s,a)\right)\right) .$$

Plugging the above inequality into (204) and (208) establishes the bounds on $\Omega_3^{(t+1)}$ and $\Omega_2^{(t+1)}$ in (68), respectively. **Step 6: bound** $-\min_{s,a}\left(\overline{Q}_\tau^{(t+1)}(s,a) - \tau\log\overline{\xi}^{(t+1)}(s,a)\right)$. We obtain the following lemma by interpreting the approximation error $e$ as part of the consensus error $\left\|\widehat{Q}_\tau^{(t)} - \overline{Q}_\tau^{(t)}\right\|_\infty$ in Lemma F.4.

**Lemma F.5** (inexact version of Lemma F.4). *Suppose $0 < \eta \leq (1-\gamma)/\tau$. For any state-action pair $(s_0, a_0) \in \mathcal{S} \times \mathcal{A}$, one has*

$$\overline{V}_\tau^{(t+1)}(s_0) - \overline{V}_\tau^{(t)}(s_0) \geq \frac{1}{\eta}\mathop{\mathbb{E}}_{s\sim d_{s_0}^{\overline{\pi}^{(t+1)}}}\left[\alpha\mathsf{KL}\left(\overline{\pi}^{(t+1)}(\cdot|s_0)\,\|\,\overline{\pi}^{(t)}(\cdot|s_0)\right) + \mathsf{KL}\left(\overline{\pi}^{(t)}(\cdot|s_0)\,\|\,\overline{\pi}^{(t+1)}(\cdot|s_0)\right)\right]$$
$$- \frac{2}{1-\gamma}\left(\left\|\widehat{Q}_\tau^{(t)} - \overline{Q}_\tau^{(t)}\right\|_\infty + \left\|e\right\|_\infty\right) , \qquad (209)$$

$$\overline{Q}_\tau^{(t+1)}(s_0, a_0) - \overline{Q}_\tau^{(t)}(s_0, a_0) \geq -\frac{2\gamma}{1-\gamma}\left(\left\|\widehat{Q}_\tau^{(t)} - \overline{Q}_\tau^{(t)}\right\|_\infty + \left\|e\right\|_\infty\right) . \qquad (210)$$

Using (210), we have

$$\overline{Q}_\tau^{(t+1)}(s,a) - \tau\left(\alpha\log\overline{\xi}^{(t)}(s,a) + (1-\alpha)\frac{\widehat{Q}_\tau^{(t)}(s,a)}{\tau}\right)$$
$$\geq \overline{Q}_\tau^{(t)}(s,a) - \tau\left(\alpha\log\overline{\xi}^{(t)}(s,a) + (1-\alpha)\frac{\widehat{Q}_\tau^{(t)}(s,a)}{\tau}\right) - \frac{2\gamma}{1-\gamma}\left(\left\|\widehat{Q}_\tau^{(t)} - \overline{Q}_\tau^{(t)}\right\|_\infty + \left\|e\right\|_\infty\right)$$
$$\geq \alpha\left(\overline{Q}_\tau^{(t)}(s,a) - \tau\log\overline{\xi}^{(t)}(s,a)\right) - \frac{2\gamma + \eta\tau}{1-\gamma}\left\|\widehat{Q}_\tau^{(t)} - \overline{Q}_\tau^{(t)}\right\|_\infty - \frac{2\gamma}{1-\gamma}\left\|e\right\|_\infty , \qquad (211)$$

which gives

$$-\min_{s,a}\left(\overline{Q}_\tau^{(t+1)}(s,a) - \tau\log\overline{\xi}^{(t+1)}(s,a)\right)$$
$$\leq -\alpha\min_{s,a}\left(\overline{Q}_\tau^{(t)}(s,a) - \tau\log\overline{\xi}^{(t)}(s,a)\right) + \frac{2\gamma + \eta\tau}{1-\gamma}M\left\|u^{(t)}\right\|_\infty + \frac{2\gamma}{1-\gamma}\left\|e\right\|_\infty . \qquad (212)$$

### F.4 Proof of Lemma D.8

**Step 1: bound** $u^{(t+1)}(s,a) = \left\|\log\boldsymbol{\xi}^{(t+1)}(s,a) - \log\overline{\xi}^{(t+1)}(s,a)\mathbf{1}_N\right\|_2$. Following the same strategy in establishing (166), we have

$$\left\|\log\boldsymbol{\xi}^{(t+1)}(s,a) - \log\overline{\xi}^{(t+1)}(s,a)\mathbf{1}_N\right\|_2$$
$$= \left\|\left(\boldsymbol{W}\log\boldsymbol{\xi}^{(t)}(s,a) - \log\overline{\xi}^{(t)}(s,a)\mathbf{1}_N\right) + \frac{\eta}{1-\gamma}\left(\boldsymbol{W}\boldsymbol{T}^{(t)}(s,a) - \widehat{Q}^{(t)}(s,a)\mathbf{1}_N\right)\right\|_2$$
$$\leq \sigma\left\|\log\boldsymbol{\xi}^{(t)}(s,a) - \log\overline{\xi}^{(t)}(s,a)\mathbf{1}_N\right\|_2 + \frac{\eta}{1-\gamma}\sigma\left\|\boldsymbol{T}^{(t)}(s,a) - \widehat{Q}^{(t)}(s,a)\mathbf{1}_N\right\|_2 , \qquad (213)$$

or equivalently

$$\left\|u^{(t+1)}\right\|_\infty \le \sigma\left\|u^{(t)}\right\|_\infty + \frac{\eta}{1-\gamma}\sigma\left\|v^{(t)}\right\|_\infty. \tag{214}$$

**Step 2: bound** $v^{(t+1)}(s,a) = \left\|\boldsymbol{T}^{(t+1)}(s,a) - \widehat{Q}^{(t+1)}(s,a)\mathbf{1}_N\right\|_2$. In the same vein of establishing (167), we have

$$\begin{aligned}
&\left\|\boldsymbol{T}^{(t+1)}(s,a) - \widehat{Q}^{(t+1)}(s,a)\mathbf{1}_N\right\|_2 \\
&\le \sigma\left\|\boldsymbol{T}^{(t)}(s,a) - \widehat{Q}^{(t)}(s,a)\mathbf{1}_N\right\|_2 + \sigma\left\|\boldsymbol{Q}^{(t+1)}(s,a) - \boldsymbol{Q}^{(t)}(s,a)\right\|_2,
\end{aligned} \tag{215}$$

The term $\left\|\boldsymbol{Q}^{(t+1)}(s,a) - \boldsymbol{Q}^{(t)}(s,a)\right\|_2$ can be bounded in a similar way in (174):

$$\left\|\boldsymbol{Q}^{(t+1)}(s,a) - \boldsymbol{Q}^{(t)}(s,a)\right\|_2 \le \frac{(1+\gamma)\gamma}{(1-\gamma)^2}\sqrt{N}\left\|w_0^{(t)}\right\|_\infty, \tag{216}$$

where the coefficient $\frac{(1+\gamma)\gamma}{(1-\gamma)^2}$ comes from $M$ in Lemma F.3 when $\tau = 0$, and $w_0^{(t)} \in \mathbb{R}^{|\mathcal{S}||\mathcal{A}|}$ is defined as

$$\forall (s,a) \in \mathcal{S} \times \mathcal{A}: \quad w_0^{(t)}(s,a) := \left\|\log\boldsymbol{\xi}^{(t+1)}(s,a) - \log\boldsymbol{\xi}^{(t)}(s,a) - \frac{\eta}{1-\gamma}V^\star(s)\mathbf{1}_N\right\|_2. \tag{217}$$

It remains to bound $\left\|w_0^{(t)}\right\|_\infty$. Towards this end, we rewrite (175) as

$$\begin{aligned}
&w_0^{(t)}(s,a) \\
&= \left\|\boldsymbol{W}\left(\log\boldsymbol{\xi}^{(t)}(s,a) + \frac{\eta}{1-\gamma}\boldsymbol{T}^{(t)}(s,a)\right) - \log\boldsymbol{\xi}^{(t)}(s,a) - \frac{\eta}{1-\gamma}V^\star(s)\mathbf{1}_N\right\|_2 \\
&= \left\|(\boldsymbol{W} - \boldsymbol{I})\left(\log\boldsymbol{\xi}^{(t)}(s,a) - \log\overline{\xi}^{(t)}(s,a)\mathbf{1}_N\right) + \frac{\eta}{1-\gamma}\left(\boldsymbol{W}\boldsymbol{T}^{(t)}(s,a) - V^\star(s)\mathbf{1}_N\right)\right\|_2 \\
&\le 2\left\|\log\boldsymbol{\xi}^{(t)}(s,a) - \log\overline{\xi}^{(t)}(s,a)\mathbf{1}_N\right\|_2 + \frac{\eta}{1-\gamma}\left\|\boldsymbol{W}\boldsymbol{T}^{(t)}(s,a) - V^\star(s)\mathbf{1}_N\right\|_2 \\
&\le 2\left\|\log\boldsymbol{\xi}^{(t)}(s,a) - \log\overline{\xi}^{(t)}(s,a)\mathbf{1}_N\right\|_2 + \frac{\eta}{1-\gamma}\left\|\boldsymbol{W}\boldsymbol{T}^{(t)}(s,a) - \widehat{Q}^{(t)}(s,a)\mathbf{1}_N\right\|_2 \\
&\quad + \frac{\eta}{1-\gamma}\cdot\sqrt{N}\left|\widehat{Q}^{(t)}(s,a) - V^\star(s)\right|.
\end{aligned} \tag{218}$$

Note that it holds for all $(s,a) \in \mathcal{S} \times \mathcal{A}$:

$$\left|\widehat{Q}^{(t)}(s,a) - V^\star(s)\right| \le \frac{1}{1-\gamma}$$

since $\widehat{Q}^{(t)}(s,a)$ and $V^\star(s)$ are both in $[0, 1/(1-\gamma)]$. This along with (218) gives

$$w_0^{(t)}(s,a) \le 2\left\|u^{(t)}\right\|_\infty + \frac{\eta}{1-\gamma}\left\|v^{(t)}\right\|_\infty + \frac{\eta\sqrt{N}}{(1-\gamma)^2}.$$

Combining the above inequality with (216) and (215), we arrive at

$$\left\|v^{(t+1)}\right\|_\infty \le \sigma\left(1 + \frac{(1+\gamma)\gamma\sqrt{N}\eta}{(1-\gamma)^3}\sigma\right)\left\|v^{(t)}\right\|_\infty + \frac{(1+\gamma)\gamma}{(1-\gamma)^2}\sqrt{N}\sigma\left\{2\left\|u^{(t)}\right\|_\infty + \frac{\eta}{(1-\gamma)^2}\cdot\sqrt{N}\right\}. \tag{219}$$

**Step 3: establish the descent equation.** The following lemma characterizes the improvement in $\phi^{(t)}(\eta)$ for every iteration of Algorithm 1, with the proof postponed to Appendix G.4.

**Lemma F.6** (Performance improvement of exact FedNPG). *For all starting state distribution $\rho \in \Delta(\mathcal{S})$, we have the iterates of FedNPG satisfy*

$$\phi^{(t+1)}(\eta) \le \phi^{(t)}(\eta) + \frac{2\eta}{(1-\gamma)^2}\left\|\widehat{Q}^{(t)} - \overline{Q}^{(t)}\right\|_\infty - \eta\left(V^\star(\rho) - \overline{V}^{(t)}(\rho)\right), \tag{220}$$

*where*

$$\phi^{(t)}(\eta) := \mathbb{E}_{s \sim d_\rho^{\pi^\star}}\left[\mathsf{KL}\left(\pi^\star(\cdot|s) \,\|\, \overline{\pi}^{(t)}(\cdot|s)\right)\right] - \frac{\eta}{1-\gamma}\overline{V}^{(t)}(d_\rho^{\pi^\star}), \quad \forall t \ge 0. \tag{221}$$

It remains to control the term $\left\|\overline{Q}^{(t)} - \widehat{Q}^{(t)}\right\|_\infty$. Similar to (169), for all $t \geq 0$, we have

$$
\left\|\overline{Q}^{(t)} - \widehat{Q}^{(t)}\right\|_\infty = \left\|\frac{1}{N}\sum_{n=1}^{N} Q_n^{\pi_n^{(t)}} - \frac{1}{N}\sum_{n=1}^{N} Q_n^{\overline{\pi}^{(t)}}\right\|_\infty
$$

$$
\overset{(a)}{\leq} \frac{(1+\gamma)\gamma}{(1-\gamma)^2} \cdot \frac{1}{N}\sum_{n=1}^{N}\left\|\log \xi_n^{(t)} - \log \overline{\xi}^{(t)}\right\|_\infty
$$

$$
\overset{(b)}{\leq} \frac{(1+\gamma)\gamma}{(1-\gamma)^2}\left\|u^{(t)}\right\|_\infty, \tag{222}
$$

where (a) invokes Lemma F.3 with $\tau = 0$ and (b) stems from the definition of $u^{(t)}$. This along with (220) gives

$$
\phi^{(t+1)}(\eta) \leq \phi^{(t)}(\eta) + \frac{2(1+\gamma)\gamma}{(1-\gamma)^4}\eta\left\|u^{(t)}\right\|_\infty - \eta\left(V^\star(\rho) - \overline{V}^{(t)}(\rho)\right).
$$

**Step 4: bound the consensus error.** To bound the consensus error $\left\|\log \pi_n^{(t)} - \log \overline{\pi}^{(t)}\right\|_\infty$ for all $n \in [N]$, we first upper bound the spectral norm of $\boldsymbol{B}(\eta)$ which we denote as $\rho(\boldsymbol{B}(\eta))$. Since $\boldsymbol{B}(\eta)$ is a nonnegative matrix, by Perron-Frobenius Theorem, $\rho(\boldsymbol{B}(\eta))$ is an eigenvalue of $\boldsymbol{B}(\eta)$. So we only need to upper bound the eigenvalue of $\rho(\boldsymbol{B}(\eta))$.

The characteristic polynomial of $\boldsymbol{B}(\eta)$ is

$$
\begin{aligned}
f(\lambda) =& (\lambda - \sigma)\left(\lambda - \sigma\left(1 + \frac{(1+\gamma)\gamma\sqrt{N}\eta}{(1-\gamma)^3}\sigma\right)\right) - \frac{\eta J}{1-\gamma}\sigma^2 \\
=& \lambda^2 - \left(2 + \frac{(1+\gamma)\gamma\sqrt{N}\eta}{(1-\gamma)^3}\sigma\right)\sigma\lambda + \left(1 + \frac{(1+\gamma)\gamma\sqrt{N}\eta}{(1-\gamma)^3}\sigma - \frac{\eta J}{1-\gamma}\right)\sigma^2.
\end{aligned}
$$

which gives

$$
\begin{aligned}
\rho(\boldsymbol{B}(\eta)) \leq& \frac{\sigma}{2}\left[\left(2 + \frac{(1+\gamma)\gamma\sqrt{N}\eta}{(1-\gamma)^3}\sigma\right) + \sqrt{\left(2 + \frac{(1+\gamma)\gamma\sqrt{N}\eta}{(1-\gamma)^3}\sigma\right)^2 - 4\left(1 + \frac{(1+\gamma)\gamma\sqrt{N}\eta}{(1-\gamma)^3}\sigma\right) + 4\frac{\eta J}{1-\gamma}}\right] \\
\leq& \frac{\sigma}{2}\left[\left(2 + \frac{(1+\gamma)\gamma\sqrt{N}\eta}{(1-\gamma)^3}\sigma\right) + \sqrt{\left(\frac{(1+\gamma)\gamma\sqrt{N}\eta}{(1-\gamma)^3}\sigma\right)^2 + 4\frac{\eta J}{1-\gamma}}\right] \\
\leq& \sigma\left[1 + \frac{(1+\gamma)\gamma\sqrt{N}\eta}{(1-\gamma)^3}\sigma + \sqrt{\frac{\eta J}{1-\gamma}}\right]. \tag{223}
\end{aligned}
$$

Note that when $\eta \leq \eta_1$, we have (recall that $J = \frac{2(1+\gamma)\gamma}{(1-\gamma)^2}\sqrt{N}$):

$$
\frac{(1+\gamma)\gamma\sqrt{N}\eta}{(1-\gamma)^3}\sigma \leq \frac{(1-\sigma)^2}{8},
$$

and

$$
\frac{\eta J}{1-\gamma} \leq \frac{(1-\sigma)^2}{4\sigma}.
$$

Plugging the above two expressions into (223) yields

$$
\begin{aligned}
\rho(\boldsymbol{B}(\eta)) \leq& \sigma\left(1 + (1-\sigma)^2/8 + (1-\sigma)/(2\sqrt{\sigma})\right) \\
\leq& \sigma\left(1 + (1-\sigma)/(8\sigma) + (1-\sigma)/(2\sigma)\right) = \frac{3}{8}\sigma + \frac{5}{8} < 1.
\end{aligned}
$$

Therefore, when $\eta \leq \eta_1$, we have

$$\left\|\mathbf{\Omega}^{(t)}\right\|_2 \leq \rho(\boldsymbol{B}(\eta)) \left\|\mathbf{\Omega}^{(t-1)}\right\|_2 + d_2(\eta)$$
$$\leq \cdots \leq \rho^t(\boldsymbol{B}(\eta)) \left\|\mathbf{\Omega}^{(0)}\right\|_2 + \sum_{i=0}^{t-1} \rho^i(\boldsymbol{B}(\eta)) \frac{(1+\gamma)\gamma N\sigma}{(1-\gamma)^4} \eta$$
$$\leq \rho^t(\boldsymbol{B}(\eta)) \left\|\mathbf{\Omega}^{(0)}\right\|_2 + \frac{2N\sigma}{(1-\gamma)^4(1-\rho(\boldsymbol{B}(\eta)))} \eta$$
$$\leq \left(\frac{3}{8}\sigma + \frac{5}{8}\right)^t \left\|\mathbf{\Omega}^{(0)}\right\|_2 + \frac{16N\sigma}{3(1-\gamma)^4(1-\sigma)} \eta . \tag{224}$$

Combining the above inequality with the following fact:

$$\forall n \in [N]: \quad \left\|\log \pi_n^{(t)} - \log \bar{\pi}^{(t)}\right\|_\infty \leq 2\left\|\log \xi_n^{(t)} - \log \bar{\xi}^{(t)}\right\|_\infty \leq \Omega_1^{(t)} \leq \left\|\mathbf{\Omega}^{(t)}\right\|_2$$

where the first inequality uses (165), we obtain (84).

### F.5 Proof of Lemma D.10

The bound on $u^{(t+1)}(s,a)$ is already established in Step 1 in Appendix F.1 and shall be omitted. As usual we only highlight the key differences with the proof of Lemma D.8 due to approximation error.

**Step 1: bound** $v^{(t+1)}(s,a) = \left\|\boldsymbol{T}^{(t+1)}(s,a) - \widehat{q}^{(t+1)}(s,a)\mathbf{1}_N\right\|_2$. Let $\boldsymbol{q}^{(t)} := \left(q_1^{\pi_1^{(t)}}, \cdots, q_N^{\pi_N^{(t)}}\right)^\top$. From (96), we have

$$\left\|\boldsymbol{T}^{(t+1)}(s,a) - \widehat{q}^{(t+1)}(s,a)\mathbf{1}_N\right\|_2$$
$$= \left\|\boldsymbol{W}\left(\boldsymbol{T}^{(t)}(s,a) + \boldsymbol{q}^{(t+1)}(s,a) - \boldsymbol{q}^{(t)}(s,a)\right) - \widehat{q}^{(t+1)}(s,a)\mathbf{1}_N\right\|_2$$
$$= \left\|\left(\boldsymbol{W}\boldsymbol{T}^{(t)}(s,a) - \widehat{q}^{(t)}(s,a)\mathbf{1}_N\right) + \boldsymbol{W}\left(\boldsymbol{q}^{(t+1)}(s,a) - \boldsymbol{q}^{(t)}(s,a)\right) + \left(\widehat{q}^{(t)}(s,a) - \widehat{q}^{(t+1)}(s,a)\right)\mathbf{1}_N\right\|_2$$
$$\leq \sigma\left\|\boldsymbol{T}^{(t)}(s,a) - \widehat{q}^{(t)}(s,a)\mathbf{1}_N\right\|_2 + \sigma\left\|\left(\boldsymbol{q}^{(t+1)}(s,a) - \boldsymbol{q}^{(t)}(s,a)\right) + \left(\widehat{q}^{(t)}(s,a) - \widehat{q}^{(t+1)}(s,a)\right)\mathbf{1}_N\right\|_2$$
$$\leq \sigma\left\|\boldsymbol{T}^{(t)}(s,a) - \widehat{q}^{(t)}(s,a)\mathbf{1}_N\right\|_2 + \sigma\left\|\boldsymbol{q}^{(t+1)}(s,a) - \boldsymbol{q}^{(t)}(s,a)\right\|_2$$
$$\leq \sigma\left\|\boldsymbol{T}^{(t)}(s,a) - \widehat{q}^{(t)}(s,a)\mathbf{1}_N\right\|_2 + \sigma\left\|\boldsymbol{Q}^{(t+1)}(s,a) - \boldsymbol{Q}^{(t)}(s,a)\right\|_2 + 2\sigma\sqrt{N}\left\|\boldsymbol{e}\right\|_\infty . \tag{225}$$

Note that (216) still holds for inexact FedNPG:

$$\left\|\boldsymbol{Q}^{(t+1)}(s,a) - \boldsymbol{Q}^{(t)}(s,a)\right\|_2 \leq \frac{(1+\gamma)\gamma}{(1-\gamma)^2}\sqrt{N}\left\|w_0^{(t)}\right\|_\infty, \tag{226}$$

where $w_0^{(t)}$ is defined in (217). We rewrite (218), the bound on $w_0^{(t)}(s,a)$, as

$$w_0^{(t)}(s,a) \leq 2\left\|\log \boldsymbol{\xi}^{(t)}(s,a) - \log \bar{\xi}^{(t)}(s,a)\mathbf{1}_N\right\|_2$$
$$+ \frac{\eta}{1-\gamma}\left\|\boldsymbol{T}^{(t)}(s,a) - \widehat{q}^{(t)}(s,a)\mathbf{1}_N\right\|_2 + \frac{\eta\sigma}{1-\gamma} \cdot \sqrt{N}\left|\widehat{q}^{(t)}(s,a) - V^\star(s)\right| . \tag{227}$$

With the following bound

$$\forall(s,a) \in \mathcal{S} \times \mathcal{A}: \quad \left|\widehat{q}^{(t)}(s,a) - V^\star(s)\right| \leq \left\|\widehat{q}^{(t)} - \overline{Q}^{(t)}\right\|_\infty + \frac{1}{1-\gamma}$$

in mind, we write (218) as

$$w_0^{(t)}(s,a) \leq 2\left\|u^{(t)}\right\|_\infty + \frac{\eta\sigma}{1-\gamma}\left\|v^{(t)}\right\|_\infty + \frac{\eta}{1-\gamma} \cdot \sqrt{N}\left(\left\|\widehat{q}^{(t)} - \overline{q}^{(t)}\right\|_\infty + \frac{1}{1-\gamma}\right) .$$

Putting all pieces together, we obtain

$$\left\|v^{(t+1)}\right\|_\infty \leq \sigma\left(1 + \frac{(1+\gamma)\gamma\sqrt{N}\eta}{(1-\gamma)^3}\sigma\right)\left\|v^{(t)}\right\|_\infty$$
$$+ \frac{(1+\gamma)\gamma}{(1-\gamma)^2}\sqrt{N}\sigma\left\{2\left\|u^{(t)}\right\|_\infty + \frac{\eta\sqrt{N}}{(1-\gamma)^2} + \frac{\eta\sqrt{N}}{1-\gamma}\left\|\boldsymbol{e}\right\|_\infty\right\} \tag{228}$$
$$+ 2\sigma\sqrt{N}\left\|\boldsymbol{e}\right\|_\infty .$$

**Step 2: establish the descent equation.** Note that Lemma F.6 directly applies by replacing $\widehat{Q}^{(t)}$ with $\widehat{q}^{(t)}$:

$$\phi^{(t+1)}(\eta) \le \phi^{(t)}(\eta) + \frac{2\eta}{(1-\gamma)^2} \left\| \widehat{q}^{(t)} - \overline{Q}^{(t)} \right\|_\infty - \eta \left( V^\star(\rho) - \overline{V}^{(t)}(\rho) \right).$$

To bound the middle term, for all $t \ge 0$, we have

$$
\begin{aligned}
\left\| \overline{Q}^{(t)} - \widehat{q}^{(t)} \right\|_\infty &= \left\| \frac{1}{N} \sum_{n=1}^N Q_n^{\pi_n^{(t)}} - \frac{1}{N} \sum_{n=1}^N Q_n^{\overline{\pi}^{(t)}} \right\|_\infty + \frac{1}{N} \left\| \sum_{n=0}^N \left( q_n^{\pi_n^{(t)}} - Q_n^{\pi_n^{(t)}} \right) \right\|_\infty \\
&\le \frac{(1+\gamma)\gamma}{(1-\gamma)^2} \cdot \frac{1}{N} \sum_{n=1}^N \left\| \log \xi_n^{(t)} - \log \overline{\xi}^{(t)} \right\|_\infty + \frac{1}{N} \sum_{n=1}^N e_n \\
&\le \frac{(1+\gamma)\gamma}{(1-\gamma)^2} \left\| u^{(t)} \right\|_\infty + \| e \|_\infty.
\end{aligned}
\tag{229}
$$

Hence, (102) is established by combining the above two inequalities.

**Step 4: bound the consensus error.** Similar as (224), here we have

$$
\begin{aligned}
\left\| \mathbf{\Omega}^{(t)} \right\|_2 &\le \rho(\boldsymbol{B}(\eta)) \left\| \mathbf{\Omega}^{(t-1)} \right\|_2 + (d_2(\eta) + c_2(\eta)) \\
&\le \cdots \le \rho^t(\boldsymbol{B}(\eta)) \left\| \mathbf{\Omega}^{(0)} \right\|_2 + \sum_{i=0}^{t-1} \rho^i(\boldsymbol{B}(\eta)) \left( \frac{(1+\gamma)\gamma N\sigma}{(1-\gamma)^4} \eta + \sqrt{N}\sigma \left( \frac{(1+\gamma)\gamma\eta\sqrt{N}}{(1-\gamma)^3} + 2 \right) \| e \|_\infty \right) \\
&\le \rho^t(\boldsymbol{B}(\eta)) \left\| \mathbf{\Omega}^{(0)} \right\|_2 + \frac{2}{1 - \rho(\boldsymbol{B}(\eta))} \left( \frac{N\sigma}{(1-\gamma)^4} \eta + \sqrt{N}\sigma \left( \frac{\eta\sqrt{N}}{(1-\gamma)^3} + 1 \right) \| e \|_\infty \right) \\
&\le \left( \frac{3}{8}\sigma + \frac{5}{8} \right)^t \left\| \mathbf{\Omega}^{(0)} \right\|_2 + \frac{16}{3(1-\sigma)} \left( \frac{N\sigma}{(1-\gamma)^4} \eta + \sqrt{N}\sigma \left( \frac{\eta\sqrt{N}}{(1-\gamma)^3} + 1 \right) \| e \|_\infty \right),
\end{aligned}
\tag{230}
$$

which indicates 103.

# G  Proof of auxiliary lemmas

## G.1  Proof of Lemma F.1

The first claim is easily verified as $\log \xi_n^{(t)}(s, \cdot)$ always deviate from $\log \pi_n^{(t)}(\cdot|s)$ by a global constant shift, as long as it holds for $t = 0$:

$$
\begin{aligned}
\log \xi_n^{(t+1)}(s, \cdot) &= \sum_{n'=1}^N [W]_{n,n'} \left( \alpha \log \xi_{n'}^{(t)}(s, \cdot) + (1-\alpha) T_n^{(t)}(s, \cdot)/\tau \right) \\
&= \alpha \sum_{n'=1}^N [W]_{n,n'} \left( \alpha \left( \log \pi_{n'}^{(t)}(s, \cdot) + c_{n'}^{(t)}(s) \mathbf{1}_{|\mathcal{A}|} \right) + (1-\alpha) T_n^{(t)}(s, \cdot)/\tau \right) \\
&= \alpha \sum_{n'=1}^N [W]_{n,n'} \left( \alpha \log \pi_{n'}^{(t)}(s, \cdot) + (1-\alpha) T_n^{(t)}(s, \cdot)/\tau \right) - \log z_n^{(t)}(s) \mathbf{1}_{|\mathcal{A}|} + c_n^{(t+1)}(s) \mathbf{1}_{|\mathcal{A}|} \\
&= \log \pi_n^{(t+1)}(\cdot|s) + c_n^{(t+1)}(s) \mathbf{1}_{|\mathcal{A}|},
\end{aligned}
$$

where $z_n^{(t)}$ is the normalization term (cf. line 5, Algorithm 2) and $\{c_n^{(t)}(s)\}$ are some constants. To prove the second claim, $\forall t \ge 0, \forall (s, a) \in \mathcal{S} \times \mathcal{A}$, let

$$\overline{T}^{(t)}(s, a) := \frac{1}{N} \mathbf{1}^\top \boldsymbol{T}^{(t)}(s, a).\tag{231}$$

Taking inner product with $\frac{1}{N}\mathbf{1}$ for both sides of ($U_T$) and using the double stochasticity property of $\boldsymbol{W}$, we get

$$\overline{T}^{(t+1)}(s, a) = \overline{T}^{(t)}(s, a) + \widehat{Q}_\tau^{(t+1)}(s, a) - \widehat{Q}_\tau^{(t)}(s, a).\tag{232}$$

By the choice of $\boldsymbol{T}^{(0)}$ (line 2 of Algorithm 2), we have $\overline{T}^{(0)} = \widehat{Q}_\tau^{(0)}$ and hence by induction

$$\forall t \geq 0: \quad \overline{T}^{(t)} = \widehat{Q}_\tau^{(t)}. \tag{233}$$

This implies

$$
\begin{aligned}
\log \overline{\xi}^{(t+1)}(s,a) - \alpha \log \overline{\xi}^{(t)}(s,a) &= (1-\alpha)\widehat{Q}_\tau^{(t)}(s,a)/\tau \\
&= (1-\alpha)\overline{T}^{(t)}(s,a)/\tau \\
&= \frac{1}{N}\mathbf{1}^\top \log \boldsymbol{\xi}^{(t+1)}(s,a) - \alpha \frac{1}{N}\mathbf{1}^\top \log \boldsymbol{\xi}^{(t)}(s,a).
\end{aligned}
$$

Therefore, to prove (161), it suffices to verify the claim for $t = 0$:

$$\frac{1}{N}\mathbf{1}^\top \log \boldsymbol{\xi}^{(0)}(s,a) = \log \|\exp(Q_\tau^\star(s,\cdot)/\tau)\|_1 + \frac{1}{N}\mathbf{1}^\top \log \boldsymbol{\pi}^{(0)}(a|s) - \log \left\|\exp\left(\frac{1}{N}\sum_{n=1}^N \log \pi_n^{(0)}(\cdot|s)\right)\right\|_1$$

$$= \log \|\exp(Q_\tau^\star(s,\cdot)/\tau)\|_1 + \log \overline{\pi}^{(0)}(a|s) = \log \overline{\xi}^{(0)}(s,a).$$

By taking logarithm over both sides of the definition of $\overline{\pi}^{(t+1)}$ (cf. (27)), we get

$$\log \overline{\pi}^{(t+1)}(a|s) = \alpha \log \overline{\pi}^{(t)}(a|s) + (1-\alpha)\widehat{Q}^{(t)}(s,a)/\tau - z^{(t)}(s) \tag{234}$$

for some constant $z^{(t)}(s)$, which deviate from the update rule of $\log \overline{\xi}^{(t+1)}$ by a global constant shift and hence verifies (162).

## G.2 Proof of Lemma F.3

For notational simplicity, we let $Q_\tau^{\theta'}$ and $Q_\tau^\theta$ denote $Q_\tau^{\pi_{\theta'}}$ and $Q_\tau^{\pi_\theta}$, respectively. From (6a) we immediately know that to bound $\left\|Q_\tau^{\theta'} - Q_\tau^\theta\right\|_\infty$, it suffices to control $\left|V_\tau^\theta(s) - V_\tau^{\theta'}(s)\right|$ for each $s \in \mathcal{S}$. By (4) we have

$$\left|V_\tau^\theta(s) - V_\tau^{\theta'}(s)\right| \leq \left|V^\theta(s) - V^{\theta'}(s)\right| + \tau \left|\mathcal{H}(s,\pi_\theta) - \mathcal{H}(s,\pi_{\theta'})\right|, \tag{235}$$

so in the following we bound both terms in the RHS of (235).

**Step 1: bounding** $\left|\mathcal{H}(s,\pi_\theta) - \mathcal{H}(s,\pi_{\theta'})\right|$**.** We first bound $\left|\mathcal{H}(s,\pi_\theta) - \mathcal{H}(s,\pi_{\theta'})\right|$ using the idea in the proof of Lemma 14 in [MXSS20]. We let

$$\theta^{(t)} = \theta + t(\theta' - \theta), \quad \forall t \in \mathbb{R}, \tag{236}$$

and let $h^{(t)} \in \mathbb{R}^{|\mathcal{S}|}$ be

$$\forall s \in \mathcal{S}: \quad h^{(t)}(s) := -\sum_{a\in\mathcal{A}} \pi_{\theta^{(t)}}(a|s) \log \pi_{\theta^{(t)}}(a|s). \tag{237}$$

Note that $\left\|h^{(t)}\right\|_\infty \leq \log |\mathcal{A}|$. We also denote $H^{(t)} : \mathcal{S} \to \mathbb{R}^{|\mathcal{A}|\times|\mathcal{A}|}$ by:

$$\forall s \in \mathcal{S}: \quad H^{(t)}(s) := \left.\frac{\partial \pi_\theta(\cdot|s)}{\partial\theta}\right|_{\theta=\theta^{(t)}} = \mathrm{diag}\{\pi_{\theta^{(t)}}(\cdot|s)\} - \pi_{\theta^{(t)}}(\cdot|s)\pi_{\theta^{(t)}}(\cdot|s)^\top, \tag{238}$$

then we have

$$
\begin{aligned}
\forall s \in \mathcal{S}: \quad \left|\frac{dh^{(t)}(s)}{dt}\right| &= \left|\left\langle \frac{\partial h^{(t)}(s)}{\partial\theta^{(t)}(\cdot|s)}, \theta'(s,\cdot) - \theta(s,\cdot)\right\rangle\right| \\
&= \left|\left\langle H^{(t)}(s)\log \pi_{\theta^{(t)}}(\cdot|s), \theta'(s,\cdot) - \theta(s,\cdot)\right\rangle\right| \\
&\leq \left\|H^{(t)}(s)\log \pi_{\theta^{(t)}}(\cdot|s)\right\|_1 \|\theta'(s,\cdot) - \theta(s,\cdot)\|_\infty,
\end{aligned} \tag{239}
$$

where $\frac{\partial h^{(t)}(s)}{\partial\theta^{(t)}(\cdot|s)}$ stands for $\frac{\partial h^{(t)}(s)}{\partial\theta(\cdot|s)}\big|_{\theta=\theta^{(t)}}$. The first term in (239) is further upper bounded as

$$
\begin{aligned}
\left\|H^{(t)}(s)\log \pi_{\theta^{(t)}}(\cdot|s)\right\|_1 &= \sum_{a\in\mathcal{A}} \pi_{\theta^{(t)}}(a|s)\left|\log \pi_{\theta^{(t)}}(a|s) - \pi_{\theta^{(t)}}(\cdot|s)^\top \log \pi_{\theta^{(t)}}(\cdot|s)\right| \\
&\leq \sum_{a\in\mathcal{A}} \pi_{\theta^{(t)}}(a|s)\left(\left|\log \pi_{\theta^{(t)}}(a|s)\right| + \left|\pi_{\theta^{(t)}}(\cdot|s)^\top \log \pi_{\theta^{(t)}}(\cdot|s)\right|\right) \\
&= -2\sum_{a\in\mathcal{A}} \pi_{\theta^{(t)}}(a,s)\log \pi_{\theta^{(t)}}(a|s) \leq 2\log |\mathcal{A}|.
\end{aligned}
$$

By Lagrange mean value theorem, there exists $t \in (0, 1)$ such that

$$\left| h_1(s) - h_0(s) \right| = \left| \frac{dh^{(t)}(s)}{dt} \right| \le 2 \log |\mathcal{A}| \, \|\theta'(s, \cdot) - \theta(s, \cdot)\|_\infty \,,$$

where the inequality follows from (239) and the above inequality. Combining (5) with the above inequality, we arrive at

$$\left| \mathcal{H}(s, \pi_\theta) - \mathcal{H}(s, \pi_{\theta'}) \right| \le \frac{2 \log |\mathcal{A}|}{1 - \gamma} \, \|\theta' - \theta\|_\infty \,. \tag{240}$$

**Step 2: bounding** $\left| V^\theta(s) - V^{\theta'}(s) \right|$**.** Similar to the previous proof, we bound $\left| V^\theta(s) - V^{\theta'}(s) \right|$ by bounding $\left| \frac{dV^{\theta^{(t)}}}{dt}(s) \right|$. By Bellman's consistency equation, the value function of $\pi_{\theta^{(t)}}$ is given by

$$V^{\theta^{(t)}}(s) = \sum_{a \in \mathcal{A}} \pi_{\theta^{(t)}}(a|s) r(s, a) + \gamma \sum_a \pi_{\theta_\alpha}(a|s) \sum_{s' \in \mathcal{S}} \mathcal{P}(s'|s, a) V^{\theta^{(t)}}(s') \,,$$

which can be represented in a matrix-vector form as

$$V^{\theta_\star^{(t)}}(s) = e_s^\top M_t r_t \,, \tag{241}$$

where $e_s \in \mathbb{R}^{|\mathcal{S}|}$ is a one-hot vector whose $s$-th entry is 1,

$$M_t := (I - \gamma P_t)^{-1} \,, \tag{242}$$

with $P_t \in \mathbb{R}^{|\mathcal{S}| \times |\mathcal{S}|}$ denoting the induced state transition matrix by $\pi_{\theta^{(t)}}$

$$P_t(s, s') = \sum_{a \in \mathcal{A}} \pi_{\theta^{(t)}}(a|s) \mathcal{P}(s'|s, a) \,, \tag{243}$$

and $r_t \in \mathbb{R}^{|\mathcal{S}|}$ is given by

$$\forall s \in \mathcal{S} : \quad r_t(s) := \sum_{a \in \mathcal{A}} \pi_{\theta^{(t)}}(a|s) r(s, a) \,. \tag{244}$$

Taking derivative w.r.t. $t$ in (241), we obtain [PP08]

$$\frac{dV^{\theta^{(t)}}(s)}{dt} = \gamma \cdot e_s^\top M_t \frac{dP_t}{dt} M_t r_t + e_s^\top M_t \frac{dr_t}{dt} \,. \tag{245}$$

We now calculate each term respectively.

- For the first term, it follows that

$$\left| \gamma \cdot e_s^\top M_t \frac{dP_t}{dt} M_t r_t \right| \le \gamma \left\| M_t \frac{dP_t}{dt} M_t r_t \right\|_\infty$$
$$\le \frac{\gamma}{1 - \gamma} \left\| \frac{dP_t}{dt} M_t r_t \right\|_\infty$$
$$\le \frac{2\gamma}{1 - \gamma} \|M_t r_t\|_\infty \|\theta' - \theta\|_\infty \tag{246}$$
$$\le \frac{2\gamma}{(1 - \gamma)^2} \|r_t\|_\infty \|\theta' - \theta\|_\infty$$
$$\le \frac{2\gamma}{(1 - \gamma)^2} \|\theta' - \theta\|_\infty \,. \tag{247}$$

where the second and fourth lines use the fact $\|M_t\|_1 \le 1/(1 - \gamma)$ [LWCC23a, Lemma 10], and the last line follow from

$$\|r_t\|_\infty = \max_{s \in \mathcal{S}} \left| \sum_{a \in \mathcal{A}} \pi_{\theta^{(t)}}(a|s) r(s, a) \right| \le 1.$$

We defer the proof of (246) to the end of proof.

- For the second term, it follows that

$$\left| e_s^\top M_t \frac{dr_t}{dt} \right| \le \frac{1}{1-\gamma} \left\| \frac{dr_t}{dt} \right\|_\infty \le \frac{1}{1-\gamma} \|\theta' - \theta\|_\infty . \tag{248}$$

where the first inequality follows again from $\|M_t\|_1 \le 1/(1-\gamma)$, and the second inequality follows from

$$\begin{aligned}
\left\| \frac{dr_t}{dt} \right\|_\infty &= \max_{s\in\mathcal{S}} \left| \frac{dr_t(s)}{dt} \right| = \max_{s\in\mathcal{S}} \left| \left\langle \frac{\partial \pi_{\theta^{(t)}}(\cdot|s)^\top r(s,\cdot)}{\partial \theta^{(t)}(s,\cdot)}, \theta'(s,\cdot) - \theta(s,\cdot) \right\rangle \right| \\
&\le \max_{s\in\mathcal{S}} \left\| \frac{\partial \pi_{\theta^{(t)}}(\cdot|s)^\top}{\partial \theta^{(t)}(s,\cdot)} r(s,\cdot) \right\|_1 \|\theta'(s,\cdot) - \theta(s,\cdot)\|_\infty \\
&= \max_{s\in\mathcal{S}} \left( \sum_{a\in\mathcal{A}} \pi_{\theta^{(t)}}(a|s) \left| r(s,a) - \pi_{\theta^{(t)}}(\cdot|s)^\top r(s,\cdot) \right| \right) \|\theta'(s,\cdot) - \theta(s,\cdot)\|_\infty \\
&\le \max_{s\in\mathcal{S}} \underbrace{\max_{a\in\mathcal{A}} \left| r(s,a) - \pi_{\theta^{(t)}}(\cdot|s)^\top r(s,\cdot) \right|}_{\le 1 \text{ since } r(s,a)\in[0,1]} \|\theta'(s,\cdot) - \theta(s,\cdot)\|_\infty \\
&\le \max_{s\in\mathcal{S}} \|\theta'(s,\cdot) - \theta(s,\cdot)\|_\infty = \|\theta' - \theta\|_\infty . \tag{249}
\end{aligned}$$

Plugging the above two inequalities into (245) and using Lagrange mean value theorem, we have

$$\left| V^\theta(s) - V^{\theta'}(s) \right| \le \frac{1+\gamma}{(1-\gamma)^2} \|\theta' - \theta\|_\infty . \tag{250}$$

**Step 3: sum up.** Combining (250), (240) and (235), we have

$$\forall s \in \mathcal{S}: \quad \left| V_\tau^\theta(s) - V_\tau^{\theta'}(s) \right| \le \frac{1+\gamma+2\tau(1-\gamma)\log|\mathcal{A}|}{(1-\gamma)^2} \|\log\pi - \log\pi'\|_\infty . \tag{251}$$

Combining (251) and (6a), (170) immediately follows.

**Proof of (246).** For any vector $x \in \mathbb{R}^{|\mathcal{S}|}$, we have

$$\left[ \frac{dP_t}{dt} x \right]_s = \sum_{s'\in\mathcal{S}} \sum_{a\in\mathcal{A}} \frac{d\pi_{\theta^{(t)}}(a|s)}{dt} \mathcal{P}(s'|s,a) x(s') ,$$

from which we can bound the $l_\infty$ norm as

$$\begin{aligned}
\left\| \frac{dP_t}{dt} x \right\|_\infty &\le \max_s \sum_{a\in\mathcal{A}} \sum_{s'\in\mathcal{S}} \mathcal{P}(s'|s,a) \left| \frac{d\pi_{\theta^{(t)}}(a|s)}{dt} \right| \|x\|_\infty \\
&= \max_s \sum_{a\in\mathcal{A}} \left| \frac{d\pi_{\theta^{(t)}}(a|s)}{dt} \right| \|x\|_\infty \\
&\le 2 \|\theta' - \theta\|_\infty \|x\|_\infty \tag{252}
\end{aligned}$$

as desired, where the last line follows from the following fact:

$$\begin{aligned}
\sum_{a\in\mathcal{A}} \left| \frac{d\pi_{\theta^{(t)}}(a|s)}{dt} \right| &= \sum_{a\in\mathcal{A}} \left| \left\langle \frac{\partial \pi_{\theta^{(t)}}(a|s)}{\partial \theta^{(t)}}, \theta' - \theta \right\rangle \right| \\
&= \sum_{a\in\mathcal{A}} \left| \left\langle \frac{\partial \pi_{\theta^{(t)}}(a|s)}{\partial \theta^{(t)}(s,\cdot)}, \theta'(s,\cdot) - \theta(s,\cdot) \right\rangle \right| \\
&= \sum_{a\in\mathcal{A}} \pi_{\theta^{(t)}}(a|s) \left| (\theta'(s,a) - \theta(s,a)) - \pi_{\theta^{(t)}}(\cdot|s)^\top (\theta'(s,\cdot) - \theta(s,\cdot)) \right| \\
&\le \max_a |\theta'(s,a) - \theta(s,a)| + \left| \pi_{\theta^{(t)}}(\cdot|s)^\top (\theta'(s,\cdot) - \theta(s,\cdot)) \right| \\
&\le 2 \|\theta' - \theta\|_\infty .
\end{aligned}$$

## G.3 Proof of Lemma F.4

To simplify the notation, we denote

$$\delta^{(t)} := \widehat{Q}_\tau^{(t)} - \overline{Q}_\tau^{(t)}. \tag{253}$$

We first rearrange the terms of (234) and obtain

$$-\tau \log \overline{\pi}^{(t)}(a|s) + \left(\overline{Q}_\tau^{(t)}(s,a) + \delta^{(t)}(s,a)\right) = \frac{1-\gamma}{\eta}\left(\log \overline{\pi}^{(t+1)}(a|s) - \log \overline{\pi}^{(t)}(a|s)\right) + \frac{1-\gamma}{\eta} z^{(t)}(s). \tag{254}$$

This in turn allows us to express $\overline{V}_\tau^{(t)}(s_0)$ for any $s_0 \in \mathcal{S}$ as follows

$$\begin{aligned}
\overline{V}_\tau^{(t)}(s_0) &= \mathop{\mathbb{E}}_{a_0 \sim \overline{\pi}^{(t)}(\cdot|s_0)}\left[-\tau \log \overline{\pi}^{(t)}(a_0|s_0) + \overline{Q}_\tau^{(t)}(s_0,a_0)\right]\\
&= \mathop{\mathbb{E}}_{a_0 \sim \overline{\pi}^{(t)}(\cdot|s_0)}\left[\frac{1-\gamma}{\eta} z^{(t)}(s_0)\right] + \mathop{\mathbb{E}}_{a_0 \sim \overline{\pi}^{(t)}(\cdot|s_0)}\left[\frac{1-\gamma}{\eta}\left(\log \overline{\pi}^{(t+1)}(a_0|s_0) - \log \overline{\pi}^{(t)}(a_0|s_0)\right) - \delta^{(t)}(s_0,a_0)\right]\\
&= \frac{1-\gamma}{\eta} z^{(t)}(s_0) - \frac{1-\gamma}{\eta}\mathsf{KL}\big(\overline{\pi}^{(t)}(\cdot|s_0)\,\|\,\overline{\pi}^{(t+1)}(\cdot|s_0)\big) - \mathop{\mathbb{E}}_{a_0 \sim \overline{\pi}^{(t)}(\cdot|s_0)}\left[\delta^{(t)}(s_0,a_0)\right]\\
&= \mathop{\mathbb{E}}_{a_0 \sim \overline{\pi}^{(t+1)}(\cdot|s_0)}\left[\frac{1-\gamma}{\eta} z^{(t)}(s_0)\right] - \frac{1-\gamma}{\eta}\mathsf{KL}\big(\overline{\pi}^{(t)}(\cdot|s_0)\,\|\,\overline{\pi}^{(t+1)}(\cdot|s_0)\big) - \mathop{\mathbb{E}}_{a_0 \sim \overline{\pi}^{(t)}(\cdot|s_0)}\left[\delta^{(t)}(s_0,a_0)\right],
\end{aligned} \tag{255}$$

where the first identity makes use of (6b), the second line follows from (254). Invoking (6b) again to rewrite the $z(s_0)$ appearing in the first term of (255), we reach

$$\begin{aligned}
\overline{V}_\tau^{(t)}(s_0) &\\
&= \mathop{\mathbb{E}}_{a_0 \sim \overline{\pi}^{(t+1)}(\cdot|s_0)}\left[-\tau \log \overline{\pi}^{(t+1)}(a_0|s_0) + \overline{Q}_\tau^{(t)}(s_0,a_0) + \left(\tau - \frac{1-\gamma}{\eta}\right)\left(\log \overline{\pi}^{(t+1)}(a_0|s_0) - \log \overline{\pi}^{(t)}(a|s)\right)\right]\\
&\quad - \frac{1-\gamma}{\eta}\mathsf{KL}\big(\overline{\pi}^{(t)}(\cdot|s_0)\,\|\,\overline{\pi}^{(t+1)}(\cdot|s_0)\big) - \mathop{\mathbb{E}}_{a_0 \sim \overline{\pi}^{(t)}(\cdot|s_0)}\left[\delta^{(t)}(s_0,a_0)\right] + \mathop{\mathbb{E}}_{a_0 \sim \overline{\pi}^{(t+1)}(\cdot|s_0)}\left[\delta^{(t)}(s_0,a_0)\right]\\
&= \mathop{\mathbb{E}}_{\substack{a_0 \sim \overline{\pi}^{(t+1)}(\cdot|s_0),\\ s_1 \sim P(\cdot|s_0,a_0)}}\left[-\tau \log \overline{\pi}^{(t+1)}(a_0|s_0) + r(s_0,a_0) + \gamma \overline{V}_\tau^{(t)}(s_0)\right]\\
&\quad - \left(\frac{1-\gamma}{\eta} - \tau\right)\mathsf{KL}\big(\overline{\pi}^{(t+1)}(\cdot|s_0)\,\|\,\overline{\pi}^{(t)}(\cdot|s_0)\big) - \frac{1-\gamma}{\eta}\mathsf{KL}\big(\overline{\pi}^{(t)}(\cdot|s_0)\,\|\,\overline{\pi}^{(t+1)}(\cdot|s_0)\big)\\
&\quad - \mathop{\mathbb{E}}_{a_0 \sim \overline{\pi}^{(t)}(\cdot|s_0)}\left[\delta^{(t)}(s_0,a_0)\right] + \mathop{\mathbb{E}}_{a_0 \sim \overline{\pi}^{(t+1)}(\cdot|s_0)}\left[\delta^{(t)}(s_0,a_0)\right].
\end{aligned} \tag{256}$$

Note that for any $(s_0,a_0) \in \mathcal{S} \times \mathcal{A}$, we have

$$\begin{aligned}
&- \mathop{\mathbb{E}}_{a_0 \sim \overline{\pi}^{(t)}(\cdot|s_0)}\left[\delta^{(t)}(s_0,a_0)\right] + \mathop{\mathbb{E}}_{a_0 \sim \overline{\pi}^{(t+1)}(\cdot|s_0)}\left[\delta^{(t)}(s_0,a_0)\right]\\
&= \sum_{a_0 \in \mathcal{A}}\left(\overline{\pi}^{(t+1)}(a_0|s_0) - \overline{\pi}^{(t)}(a_0|s_0)\right)\delta^{(t)}(s_0,a_0)\\
&\leq \left\|\overline{\pi}^{(t+1)}(\cdot|s_0) - \overline{\pi}^{(t)}(\cdot|s_0)\right\|_1 \left\|\delta^{(t)}\right\|_\infty \leq 2\left\|\delta^{(t)}\right\|_\infty.
\end{aligned} \tag{257}$$

To finish up, applying (256) recursively to expand $\overline{V}_\tau^{(t)}(s_i)$, $i \geq 1$ and making use of (257), we arrive at

$$
\begin{aligned}
&\overline{V}_\tau^{(t)}(s_0) \\
&\leq \sum_{i=1}^\infty \gamma^i \cdot 2 \left\| \delta^{(t)} \right\|_\infty + \underset{\substack{a_i \sim \overline{\pi}^{(t+1)}(\cdot|s_i), \\ s_{i+1} \sim P(\cdot|s_i, a_i), \forall i \geq 0}}{\mathbb{E}} \left[ \sum_{i=1}^\infty \gamma^i \left\{ r(s_i, a_i) - \tau \log \overline{\pi}^{(t+1)}(a_i|s_i) \right\} \right. \\
&\qquad \left. - \sum_{i=1}^\infty \gamma^i \left\{ \left( \frac{1-\gamma}{\eta} - \tau \right) \mathsf{KL}\big(\overline{\pi}^{(t+1)}(\cdot|s_i) \,\|\, \overline{\pi}^{(t)}(\cdot|s_i)\big) + \frac{1-\gamma}{\eta} \mathsf{KL}\big(\overline{\pi}^{(t)}(\cdot|s_i) \,\|\, \overline{\pi}^{(t+1)}(\cdot|s_i)\big) \right\} \right] \\
&= \frac{2}{1-\gamma} \left\| \delta^{(t)} \right\|_\infty + \overline{V}_\tau^{(t+1)}(s_0) \\
&\qquad - \underset{s \sim d_{s_0}^{\overline{\pi}^{(t+1)}}}{\mathbb{E}} \left[ \left( \frac{1}{\eta} - \frac{\tau}{1-\gamma} \right) \mathsf{KL}\big(\overline{\pi}^{(t+1)}(\cdot|s_i) \,\|\, \overline{\pi}^{(t)}(\cdot|s_i)\big) + \frac{1}{\eta} \mathsf{KL}\big(\overline{\pi}^{(t)}(\cdot|s_i) \,\|\, \overline{\pi}^{(t+1)}(\cdot|s_i)\big) \right],
\end{aligned}
$$
(258)

where the third line follows since $\overline{V}_\tau^{(t+1)}$ can be viewed as the value function of $\overline{\pi}^{(t+1)}$ with adjusted rewards $\overline{r}^{(t+1)}(s, a) := r(s, a) - \tau \log \overline{\pi}^{(t+1)}(s|a)$. And (188) follows immediately from the above inequality (258). By (6a) we can easily see that (189) is a consequence of (188).

### G.4 Proof of Lemma F.6

We first introduce the famous performance difference lemma which will be used in our proof.

**Lemma G.1** (Performance difference lemma). *For any policy $\pi, \pi' \in \Delta(\mathcal{A})^{\mathcal{S}}$ and $\rho \in \Delta(\mathcal{S})$, we have*

$$
V^\pi(\rho) - V^{\pi'}(\rho) = \frac{1}{1-\gamma} \mathbb{E}_{(s,a) \sim \bar{d}^\pi} \left[ A^{\pi'}(s, a) \right]
$$
(259)

$$
= \frac{1}{1-\gamma} \mathbb{E}_{s \sim d^\pi} \left[ \langle Q^{\pi'}(s), \pi(s) - \pi'(s) \rangle \right].
$$
(260)

*Proof.* See Lemma 3 in [YDG⁺22]. □

For all $t \geq 0$, we define the advantage function $\overline{A}^{(t)}$ as:

$$
\forall (s, a) \in \mathcal{S} \times \mathcal{A}: \quad \overline{A}^{(t)}(s, a) := \overline{Q}^{(t)}(s, a) - \overline{V}^{(t)}(s).
$$
(261)

Then for Alg. 1, the update rule of $\overline{\pi}$ (Eq. (234)) can be written as

$$
\log \overline{\pi}^{(t+1)}(a|s) = \log \overline{\pi}^{(t)}(a|s) + \frac{\eta}{1-\gamma} \left( \overline{A}^{(t)}(s, a) + \delta^{(t)}(s, a) \right) - \log \widehat{z}^{(t)}(s),
$$
(262)

where $\delta^{(t)}$ is defined in (253) and

$$
\begin{aligned}
\log \widehat{z}^{(t)}(s) &= \log \sum_{a' \in \mathcal{A}} \overline{\pi}^{(t)}(a'|s) \exp \left\{ \frac{\eta}{1-\gamma} \left( \overline{A}^{(t)}(s, a') + \delta^{(t)}(s, a') \right) \right\} \\
&\geq \sum_{a' \in \mathcal{A}} \overline{\pi}^{(t)}(a'|s) \log \exp \left\{ \frac{\eta}{1-\gamma} \left( \overline{A}^{(t)}(s, a') + \delta^{(t)}(s, a') \right) \right\} \\
&= \frac{\eta}{1-\gamma} \sum_{a' \in \mathcal{A}} \overline{\pi}^{(t)}(a'|s) \left( \overline{A}^{(t)}(s, a') + \delta^{(t)}(s, a') \right) \\
&= \frac{\eta}{1-\gamma} \sum_{a' \in \mathcal{A}} \overline{\pi}^{(t)}(a'|s) \delta^{(t)}(s, a') \geq -\frac{\eta}{1-\gamma} \left\| \delta^{(t)} \right\|_\infty,
\end{aligned}
$$
(263)

where the first inequality follows by Jensen's inequality on the concave function $\log x$ and the last equality uses $\sum_{a' \in \mathcal{A}} \overline{\pi}^{(t)}(a'|s) \overline{A}^{(t)}(s, a') = 0$.

For all starting state distribution $\mu$, we use $d^{(t+1)}$ as shorthand for $d_\mu^{\overline{\pi}^{(t+1)}}$, the performance difference lemma (Lemma G.1) implies:

$$
\begin{aligned}
&\overline{V}^{(t+1)}(\mu) - \overline{V}^{(t)}(\mu) \\
&= \frac{1}{1-\gamma}\mathbb{E}_{s\sim d^{(t+1)}}\sum_{a\in\mathcal{A}}\overline{\pi}^{(t+1)}(a|s)\left(\overline{A}^{(t)}(s,a)+\delta^{(t)}(s,a)\right) - \frac{1}{1-\gamma}\mathbb{E}_{s\sim d^{(t+1)}}\mathbb{E}_{a\sim\overline{\pi}^{(t+1)}(\cdot|s)}\left[\delta^{(t)}(s,a)\right] \\
&= \frac{1}{\eta}\mathbb{E}_{s\sim d^{(t+1)}}\sum_{a\in\mathcal{A}}\overline{\pi}^{(t+1)}(a|s)\log\frac{\overline{\pi}^{(t+1)}(a|s)\widehat{z}^{(t)}(s)}{\overline{\pi}^{(t)}(a|s)} - \frac{1}{1-\gamma}\mathbb{E}_{s\sim d^{(t+1)}}\mathbb{E}_{a\sim\overline{\pi}^{(t+1)}(\cdot|s)}\left[\delta^{(t)}(s,a)\right] \\
&= \frac{1}{\eta}\mathbb{E}_{s\sim d^{(t+1)}}\mathsf{KL}\big(\overline{\pi}^{(t+1)}(\cdot|s)\,\|\,\overline{\pi}^{(t)}(\cdot|s)\big) + \frac{1}{\eta}\mathbb{E}_{s\sim d^{(t+1)}}\log\widehat{z}^{(t)}(s) - \frac{1}{1-\gamma}\mathbb{E}_{s\sim d^{(t+1)}}\mathbb{E}_{a\sim\overline{\pi}^{(t+1)}(\cdot|s)}\left[\delta^{(t)}(s,a)\right] \\
&\geq \frac{1}{\eta}\mathbb{E}_{s\sim d^{(t+1)}}\left(\log\widehat{z}^{(t)}(s)+\frac{\eta}{1-\gamma}\big\|\delta^{(t)}\big\|_\infty\right) - \frac{2}{1-\gamma}\big\|\delta^{(t)}\big\|_\infty,
\end{aligned}
$$

from which we can see that

$$
\overline{V}^{(t+1)}(\mu) - \overline{V}^{(t)}(\mu) \geq -\frac{2}{1-\gamma}\big\|\delta^{(t)}\big\|_\infty, \tag{264}
$$

where we use (263), and that

$$
\overline{V}^{(t+1)}(\mu) - \overline{V}^{(t)}(\mu) \geq \frac{1-\gamma}{\eta}\mathbb{E}_{s\sim\mu}\left(\log\widehat{z}^{(t)}(s)+\frac{\eta}{1-\gamma}\big\|\delta^{(t)}\big\|_\infty\right) - \frac{2}{1-\gamma}\big\|\delta^{(t)}\big\|_\infty, \tag{265}
$$

which follows from $d^{(t+1)} = d_\mu^{\overline{\pi}^{(t+1)}} \geq (1-\gamma)\mu$ and the fact that $\log\widehat{z}^{(t)}(s)+\frac{\eta}{1-\gamma}\big\|\delta^{(t)}\big\|_\infty \geq 0$ (by (263)).

For any fixed $\rho$, we use $d^\star$ as shorthand for $d_\rho^{\pi^\star}$. By the performance difference lemma (Lemma G.1),

$$
\begin{aligned}
&V^\star(\rho) - \overline{V}^{(t)}(\rho) \\
&= \frac{1}{1-\gamma}\mathbb{E}_{s\sim d^\star}\sum_{a\in\mathcal{A}}\pi^\star(a|s)\left(\overline{A}^{(t)}(s,a)+\delta^{(t)}(s,a)\right) - \frac{1}{1-\gamma}\mathbb{E}_{s\sim d^\star}\mathbb{E}_{a\sim\pi^\star(\cdot|s)}\left[\delta^{(t)}(s,a)\right] \\
&= \frac{1}{\eta}\mathbb{E}_{s\sim d^\star}\sum_{a\in\mathcal{A}}\pi^\star(a|s)\log\frac{\overline{\pi}^{(t+1)}(a|s)\widehat{z}^{(t)}(s)}{\overline{\pi}^{(t)}(a|s)} - \frac{1}{1-\gamma}\mathbb{E}_{s\sim d^\star}\mathbb{E}_{a\sim\pi^\star(\cdot|s)}\left[\delta^{(t)}(s,a)\right] \\
&= \frac{1}{\eta}\mathbb{E}_{s\sim d^\star}\left(\mathsf{KL}\big(\pi^\star(\cdot|s)\,\|\,\overline{\pi}^{(t)}(\cdot|s)\big) - \mathsf{KL}\big(\pi^\star(\cdot|s)\,\|\,\overline{\pi}^{(t+1)}(\cdot|s)\big) + \log\widehat{z}^{(t)}(s)\right) - \frac{1}{1-\gamma}\mathbb{E}_{s\sim d^\star}\mathbb{E}_{a\sim\pi^\star(\cdot|s)}\left[\delta^{(t)}(s,a)\right] \\
&\leq \frac{1}{\eta}\mathbb{E}_{s\sim d^\star}\left(\mathsf{KL}\big(\pi^\star(\cdot|s)\,\|\,\overline{\pi}^{(t)}(\cdot|s)\big) - \mathsf{KL}\big(\pi^\star(\cdot|s)\,\|\,\overline{\pi}^{(t+1)}(\cdot|s)\big) + \left(\log\widehat{z}^{(t)}(s)+\frac{\eta}{1-\gamma}\big\|\delta^{(t)}\big\|_\infty\right)\right),
\end{aligned} \tag{266}
$$

where we use (262) in the second equality.

By applying (265) with $\mu = d^\star$ as the initial state distribution, we have

$$
\frac{1}{\eta}\mathbb{E}_{s\sim\mu}\left(\log\widehat{z}^{(t)}(s)+\frac{\eta}{1-\gamma}\big\|\delta^{(t)}\big\|_\infty\right) \leq \frac{1}{1-\gamma}\left(\overline{V}^{(t+1)}(d^\star)-\overline{V}^{(t)}(d^\star)\right) + \frac{2}{(1-\gamma)^2}\big\|\delta^{(t)}\big\|_\infty.
$$

Plugging the above equation into (266), we obtain

$$
\begin{aligned}
V^\star(\rho) - \overline{V}^{(t)}(\rho) &\leq \frac{1}{\eta}\mathbb{E}_{s\sim d^\star}\left(\mathsf{KL}\big(\pi^\star(\cdot|s)\,\|\,\overline{\pi}^{(t)}(\cdot|s)\big) - \mathsf{KL}\big(\pi^\star(\cdot|s)\,\|\,\overline{\pi}^{(t+1)}(\cdot|s)\big)\right) \\
&\quad + \frac{1}{1-\gamma}\left(\overline{V}^{(t+1)}(d^\star)-\overline{V}^{(t)}(d^\star)\right) + \frac{2}{(1-\gamma)^2}\big\|\delta^{(t)}\big\|_\infty,
\end{aligned}
$$

which gives Lemma F.6.

## G.5   Proof of Theorem E.3

The proof of Theorem E.3 could be found in Appendix C.5 in [YDG$^+$22]. We present it for completeness. To prove Theorem E.3, we need the following Theorem G.2.

**Theorem G.2** (Theorem 1 in [BM13]). *Consider the following assaumptions:*

  (i) *The observations $(\boldsymbol{a}_k, \boldsymbol{b}_k) \in \mathbb{R}^p \times \mathbb{R}^p$ are independent and identically distributed.*

  (ii) $\mathbb{E}\left[\|\boldsymbol{a}_k\|^2\right]$[8] *and* $\mathbb{E}\left[\|\boldsymbol{b}_k\|^2\right]$ *are finite. The covariance* $\mathbb{E}\left[\boldsymbol{a}_k\boldsymbol{a}_k^\top\right]$ *is invertible.*

  (iii) *The global minimum of* $g(w) = \frac{1}{2}\mathbb{E}\left[\langle\boldsymbol{w}, \boldsymbol{a}_k\rangle^2 - 2\langle\boldsymbol{w}, \boldsymbol{b}_k\rangle\right]$ *is attained at a certain* $\boldsymbol{w}^\star \in \mathbb{R}^p$. *Let* $\Delta_k = \boldsymbol{b}_k - \langle\boldsymbol{w}^\star, \boldsymbol{a}_k\rangle\boldsymbol{a}_k$ *denote the residual. We have* $\mathbb{E}[\Delta_k] = 0$.

  (iv) $\exists R > 0$ *and* $\sigma > 0$ *such that* $\mathbb{E}\left[\Delta_k\Delta_k^\top\right] \leq \sigma^2\mathbb{E}\left[\boldsymbol{a}_k\boldsymbol{a}_k^\top\right]$ *and* $\mathbb{E}\left[\|\boldsymbol{a}_k\|^2 \boldsymbol{a}_k\boldsymbol{a}_k^\top\right] \leq R^2\mathbb{E}\left[\boldsymbol{a}_k\boldsymbol{a}_k^\top\right]$.

*Consider the stochastic gradient recursion*
$$\boldsymbol{w}_{k+1} = \boldsymbol{w}_k - \eta\left(\langle\boldsymbol{w}_k, \boldsymbol{a}_k\rangle\boldsymbol{a}_k - \boldsymbol{b}_k\right)$$
*started from* $\boldsymbol{w}_0 \in \mathbb{R}^p$. *Let* $\boldsymbol{w}_{out} = \frac{1}{K}\sum_{k=1}^K \boldsymbol{w}_k$. *When* $\eta = \frac{1}{4R^2}$, *we have*

$$\mathbb{E}\left[g(\boldsymbol{w}_{out}) - g(\boldsymbol{w}^\star)\right] \leq \frac{2}{K}(\sigma\sqrt{p} + R\|\boldsymbol{w}_0 - \boldsymbol{w}^\star\|)^2. \tag{267}$$

In the proof of Theorem E.3 we'll show that for Algorithm 4, the assumptions in Theorem G.2 are all satisfied and thus we can use the result (267).

*Proof of Theorem E.3.* We let $\boldsymbol{a}_k$ and $\boldsymbol{b}_k$ in Theorem G.2 be $\phi(s,a)$ and $\widehat{Q}_\xi\phi(s,a)$ in Algorithm 4, respectively. And we let $\|\cdot\| = \|\cdot\|_2$ in Theorem G.2. Since the observations $\left(\phi(s,a), \widehat{Q}_\xi(s,a)\phi(s,a)\right) \in \mathbb{R}^p \times \mathbb{R}^p$ are i.i.d., (i) is satisfied.

As we assume $\|\phi(s,a)\|_2 \leq C_\phi$, $\mathbb{E}\left[\|\phi(s,a)\|_2^2\right]$ is finite. From Assumption 4.1 we know that $\mathbb{E}\left[\phi(s,a)\phi(s,a)^\top\right]$ is invertible.

Let $H$ be the length of trajectory for estimating $\widehat{Q}_\xi(s,a)$. Then $\left(\widehat{Q}_\xi(s,a)\right)^2$ is bounded by

$$\mathbb{E}\left[\left(\widehat{Q}_\xi(s,a)\right)^2\right] = \mathbb{E}_{(s,a)\sim\tilde{d}_\nu^{\pi_\xi}}\left[\sum_{\tau=0}^\infty Pr(H=\tau)\mathbb{E}\left[\left(\sum_{t=0}^\tau r(s_t,a_t)\right)^2 \middle| H=\tau, s_0=s, a_0=a\right]\right]$$

$$= \mathbb{E}_{(s,a)\sim\tilde{d}_\nu^{\pi_\xi}}\left[(1-\gamma)\sum_{\tau=0}^\infty \gamma^\tau\mathbb{E}\left[\left(\sum_{t=0}^\tau r(s_t,a_t)\right)^2 \middle| H=\tau, s_0=s, a_0=a\right]\right]$$

$$\leq \mathbb{E}_{(s,a)\sim\tilde{d}_\nu^{\pi_\xi}}\left[(1-\gamma)\sum_{\tau=0}^\infty \gamma^\tau(\tau+1)^2\right] \leq \frac{2}{(1-\gamma)^2}, \tag{268}$$

from which we deduce $\mathbb{E}\left[\left\|\widehat{Q}_\xi(s,a)\phi(s,a)\right\|_2^2\right] \leq C_\phi^2\mathbb{E}\left[\widehat{Q}_\xi(s,a)^2\right]$ is bounded. Thus (ii) holds.

Furthermore, we introduce the residual

$$\Delta := \left(\widehat{Q}_\xi(s,a) - \phi(s,a)^\top w^\star\right)\phi(s,a), \tag{269}$$

then from [YDG$^+$22, Lemma 7] we know that $\mathbb{E}[\Delta] = \frac{1}{2}\nabla_w\ell(\boldsymbol{w}^\star, \widehat{Q}_\xi, d_\nu^{\pi_\xi}) = 0$, which gives (iii).

To verify (iv), we let $R = C_\phi$ in Theorem G.2, then $\mathbb{E}\left[\|\phi(s,a)\|_2^2\phi(s,a)\phi(s,a)^\top\right] \leq C_\phi^2\mathbb{E}\left[\phi(s,a)\phi(s,a)^\top\right]$. Also note that

$$\boldsymbol{w}^\star = \left(\mathbb{E}_{(s,a)\sim\tilde{d}_\nu^{\pi_\xi}}\left[\phi(s,a)\phi(s,a)^\top\right]\right)^\dagger \mathbb{E}_{(s,a)\sim\tilde{d}_\nu^{\pi_\xi}}\left[\widehat{Q}_\xi(s,a)\phi(s,a)\right]$$

$$\leq \frac{1}{1-\gamma}\left(\mathbb{E}_{(s,a)\sim\nu}\left[\phi(s,a)\phi(s,a)^\top\right]\right)^\dagger \mathbb{E}_{(s,a)\sim\tilde{d}_\nu^{\pi_\xi}}\left[\widehat{Q}_\xi(s,a)\phi(s,a)\right], \tag{270}$$

---

[8]Here $\|\cdot\|$ could be any norm in $\mathbb{R}^p$.

from which we deduce

$$\|\boldsymbol{w}^\star\|_2 \le \frac{B}{\mu(1-\gamma)^2}. \tag{271}$$

$$\mathbb{E}\left[\left(\widehat{Q}_\xi(s,a) - \phi(s,a)^\top \boldsymbol{w}^\star\right)^2 | s,a\right] = \mathbb{E}\left[\left(\widehat{Q}_\xi(s,a)\right)^2 | s,a\right] - 2Q_\xi(s,a)\phi(s,a)^\top \boldsymbol{w}^\star + (\phi(s,a)^\top w^\star)^2 \tag{272}$$

$$\le \frac{2}{(1-\gamma)^2} + \frac{2C_\phi^2}{\mu(1-\gamma)^3} + \frac{C_\phi^4}{\mu^2(1-\gamma)^4}$$

$$\le \frac{2}{(1-\gamma)^2}\left(\frac{C_\phi^2}{\mu(1-\gamma)} + 1\right)^2. \tag{273}$$

The above expression implies

$$\mathbb{E}\left[\Delta\Delta^\top\right] = \mathbb{E}_{(s,a)\sim \tilde{d}_\nu^{\pi_\xi}}\left[\left(\widehat{Q}_\xi(s,a) - \phi(s,a)^\top \boldsymbol{w}^\star\right)^2 \phi(s,a)\phi(s,a)^\top | s,a\right]$$

$$= \mathbb{E}_{(s,a)\sim \tilde{d}_\nu^{\pi_\xi}}\left[\mathbb{E}\left[\left(\widehat{Q}_\xi(s,a) - \phi(s,a)^\top \boldsymbol{w}^\star\right)^2 | s,a\right]\phi(s,a)\phi(s,a)^\top\right]$$

$$\le \left(\underbrace{\frac{\sqrt{2}}{1-\gamma}\left(\frac{C_\phi^2}{\mu(1-\gamma)} + 1\right)}_{\sigma}\right)\mathbb{E}[\phi(s,a)\phi(s,a)^\top]. \tag{274}$$

Therefore, (iv) is verified.

Thus by (267), with stepsize $\beta = \frac{1}{2C_\phi^2}$, initialization $\boldsymbol{w}_0 = \boldsymbol{0}$ and $K$ steps of critic updates, we have

$$\mathbb{E}\left[\ell\left(\boldsymbol{w}_{\text{out}}, Q_\xi, \tilde{d}_\xi\right)\right] - \ell\left(\boldsymbol{w}^\star, Q_\xi, \tilde{d}_\xi\right) \le \frac{4}{K}\left(\sigma\sqrt{p} + C_\phi\|\boldsymbol{w}^\star\|_2\right)^2$$

$$\le \frac{4}{K}\left(\frac{\sqrt{2p}}{1-\gamma}\left(\frac{C_\phi^2}{\mu(1-\gamma)} + 1\right) + \frac{C_\phi^2}{\mu(1-\gamma)^2}\right)^2,$$

which gives (113). $\qquad\square$

### G.6   Proof of Lemma E.7

*Proof of Lemma E.7.* For notational simplicity we let $V^\xi, V^{\xi'}$ denote $V^{f_\xi}, V^{f_{\xi'}}$, resp. Same as in Lemma F.3, We define $\xi^{(t)} = \xi + t(\xi' - \xi)$ and define $\boldsymbol{P}_t, \boldsymbol{M}_t, r_t$ by replacing $\pi_{\xi^{(t)}}$ with $f_{\xi^{(t)}}$ in (243),(242) and (244), respectively. Define

$$\bar{\phi}_\xi(s,a) = \phi(s,a) - \mathbb{E}_{a'\sim f_{\xi^{(t)}}}[\phi(s,a')],$$

then we have

$$\frac{\partial f_\xi(a|s)}{\partial \xi} = f_\xi(a|s)\bar{\phi}_\xi(s,a). \tag{275}$$

Analogous to (252), we have

$$\left\|\frac{d\boldsymbol{P}_t}{dt}x\right\|_\infty \le \max_s \sum_{a\in\mathcal{A}}\sum_{s'\in\mathcal{S}}\mathcal{P}(s'|s,a)\left|\frac{d\pi_{\xi^{(t)}}(a|s)}{dt}\right|\|x\|_\infty$$

$$= \max_s \sum_{a\in\mathcal{A}}\left|\frac{d\pi_{\xi^{(t)}}(a|s)}{dt}\right|\|x\|_\infty$$

$$\le 2C_\phi\|\xi' - \xi\|_2\|x\|_\infty$$

where the last line follows is due to

$$\sum_{a\in\mathcal{A}}\left|\frac{df_{\xi^{(t)}}(a|s)}{dt}\right| = \sum_{a\in\mathcal{A}}\left|\left\langle\frac{\partial f_{\xi^{(t)}}(a|s)}{\partial\boldsymbol{\xi}^{(t)}}, \boldsymbol{\xi}'-\boldsymbol{\xi}\right\rangle\right|$$

$$= \sum_{a\in\mathcal{A}} f_{\xi^{(t)}}(a|s)\left|\langle\bar{\phi}_\xi(s,a), \boldsymbol{\xi}'-\boldsymbol{\xi}\rangle\right|$$

$$\leq \sum_{a\in\mathcal{A}} f_{\xi^{(t)}}(a|s)\left\|\bar{\phi}_\xi(s,a)\right\|_2\left\|\boldsymbol{\xi}'-\boldsymbol{\xi}\right\|_2$$

$$\leq 2C_\phi\left\|\boldsymbol{\xi}'-\boldsymbol{\xi}\right\|_\infty.$$

Same as (245) in Lemma F.3, we have

$$\frac{dV^{\xi^{(t)}}(s)}{dt} = \gamma\cdot e_s^\top\boldsymbol{M}_t\frac{d\boldsymbol{P}_t}{dt}\boldsymbol{M}_t r_t + e_s^\top\boldsymbol{M}_t\frac{dr_t}{dt}. \tag{276}$$

And similar to (249), we deduce

$$\left\|\frac{dr_t}{dt}\right\|_\infty = \max_{s\in\mathcal{S}}\left|\frac{dr_t(s)}{dt}\right| = \max_{s\in\mathcal{S}}\left|\left\langle\frac{\partial f_{\xi^{(t)}}(\cdot|s)^\top r(s,\cdot)}{\partial\boldsymbol{\xi}^{(t)}}, \boldsymbol{\xi}'-\boldsymbol{\xi}\right\rangle\right|$$

$$= \left|\left\langle\sum_{a\in\mathcal{A}} f_\xi(a|s)\bar{\phi}_\xi(s,a)r(s,a), \boldsymbol{\xi}'-\boldsymbol{\xi}\right\rangle\right|$$

$$= \sum_{a\in\mathcal{A}} f_\xi(a|s)r(s,a)\left|\langle\bar{\phi}_\xi(s,a), \boldsymbol{\xi}'-\boldsymbol{\xi}\rangle\right|$$

$$\leq 2C_\phi\left\|\boldsymbol{\xi}'-\boldsymbol{\xi}\right\|_2,$$

which gives

$$\left|e_s^\top\boldsymbol{M}_t\frac{dr_t}{dt}\right| \leq \frac{1}{1-\gamma}\left\|\frac{dr_t}{dt}\right\|_\infty \leq \frac{2C_\phi}{1-\gamma}\left\|\boldsymbol{\xi}'-\boldsymbol{\xi}\right\|_2. \tag{277}$$

Following the same steps in (247), we deduce

$$\left|\gamma\cdot e_s^\top\boldsymbol{M}_t\frac{d\boldsymbol{P}_t}{dt}\boldsymbol{M}_t r_t\right| \leq \frac{2\gamma C_\phi}{(1-\gamma)^2}\left\|\boldsymbol{\xi}'-\boldsymbol{\xi}\right\|_2. \tag{278}$$

Combining the above two expressions (277) and (278) with (276), we deduce

$$|V^\xi(s) - V^{\xi'}(s)| \leq \frac{2C_\phi(1+\gamma)}{(1-\gamma)^2}\left\|\boldsymbol{\xi}'-\boldsymbol{\xi}\right\|_2, \tag{279}$$

which implies

$$\forall(s,a)\in\mathcal{S}\times\mathcal{A}: \quad |Q^\xi(s,a) - Q^{\xi'}(s,a)| \leq \frac{2C_\phi\gamma(1+\gamma)}{(1-\gamma)^2}\left\|\boldsymbol{\xi}'-\boldsymbol{\xi}\right\|_2. \tag{280}$$

$\square$

### G.7 Proof of Lemma E.8

This proof is inspired by the proof of [YDG⁺22, Theorem 1]. To give the proof, we first introduce the following three-point descent lemma:

**Lemma G.3** (Three-point descent lemma Lemma 6 in [Xia22]). *Suppose that $\mathcal{C}\subset\mathbb{R}^m$ is a closed convex set, $g:\mathcal{C}\to\mathbb{R}$ is a proper, closed, convex function, $D_h(\cdot,\cdot)$ is the Bregman divergence generated by a function $h$ of Lengendre type and $\mathrm{rint}\,\mathrm{dom}h\cap\mathcal{C}\neq\emptyset$. For any $x\in\mathrm{rintdom}h$, let*

$$x^+ \in \arg\min_{u\in\mathrm{dom}h\cap\mathcal{C}}\{f(u)+D_h(u,x)\},$$

*then $x^+\in\mathrm{dom}h\cap\mathcal{C}$ and for any $u\in\mathrm{dom}h\cap\mathcal{C}$, it holds that*

$$f(x^+)+D_h(x^+,x) \leq f(u)+D_h(u,x)-D_h(u,x^+). \tag{281}$$

*Proof of Lemma E.8.* By the update rule (114) and the parameterization (24) we know know that

$$\forall (s,a) \in \mathcal{S} \times \mathcal{A}: \quad \bar{f}^{(t+1)}(a|s) = \frac{1}{Z^{(t)}(s)} f^{(t)}(a|s) \exp\left(\alpha \phi^\top(s,a) \hat{\boldsymbol{w}}^{(t)}\right),$$

where $Z^{(t)}(s)$ is a normalization coefficient to ensure $\sum_{a \in \mathcal{A}} f^{(t+1)}(s,a) = 1$ for each $s \in \mathcal{S}$. Note that the above $\pi^{(t+1)}$ could also be obtained by a mirror descent update:

$$\forall s \in \mathcal{S}: \quad f^{(t+1)}(\cdot|s) = \arg\min_{g \in \Delta(\mathcal{A})} \left\{ -\alpha \langle \Phi(s)\hat{\boldsymbol{w}}^{(t)}, g \rangle + D(g, f^{(t)}(\cdot|s)) \right\}, \qquad (282)$$

where $\Phi(s) \in \mathbb{R}^{|\mathcal{A}| \times p}$ is a matrix with rows $\phi^\top(s,a) \in \mathbb{R}^p$ for $a \in \mathcal{A}$, and $D(\cdot, \cdot)$ denotes the KL divergence defined in (109).

We apply the three-point descent lemma—Lemma G.3 with $\mathcal{C} = \Delta(\mathcal{A})$, $f = -\alpha \langle \Phi(s)\hat{\boldsymbol{w}}^{(t)}, \cdot \rangle$ and $h : \Delta(\mathcal{A}) \to \mathbb{R}$ is the negative entropy with $h(q) = \sum_{a \in \mathcal{A}} q(a) \log q(a)$ and deduce that for any $q \in \Delta(\mathcal{A})$, we have

$$-\alpha \langle \Phi(s)\hat{\boldsymbol{w}}^{(t)}, \bar{f}^{(t+1)}(\cdot|s) \rangle + D\left(\bar{f}^{(t+1)}(\cdot|s), \bar{f}^{(t)}(\cdot|s)\right) \le -\alpha \langle \Phi(s)\hat{\boldsymbol{w}}^{(t)}, q \rangle + D\left(q, \bar{f}^{(t)}(\cdot|s)\right) - D\left(q, \bar{f}^{(t+1)}(\cdot|s)\right).$$

Rearranging terms and dividing both sides by $-\alpha$, we obtain

$$\langle \Phi(s)\hat{\boldsymbol{w}}^{(t)}, \bar{f}^{(t+1)}(\cdot|s) - q \rangle - \frac{1}{\alpha} D\left(\bar{f}^{(t+1)}(\cdot|s), \bar{f}^{(t)}(\cdot|s)\right) \ge -\frac{1}{\alpha} D\left(q, \bar{f}^{(t)}(\cdot|s)\right) + \frac{1}{\alpha} D\left(q, \bar{f}^{(t+1)}(\cdot|s)\right). \qquad (283)$$

Let $q = \bar{f}^{(t)}(\cdot|s)$ and $\pi^\star(\cdot|s)$,resp., we have the following two inequalities:

$$\langle \Phi(s)\hat{\boldsymbol{w}}^{(t)}, \bar{f}^{(t+1)}(\cdot|s) - \bar{f}^{(t)}(\cdot|s) \rangle \ge \frac{1}{\alpha} D\left(\bar{f}^{(t+1)}(\cdot|s), \bar{f}^{(t)}(\cdot|s)\right) + \frac{1}{\alpha} D\left(\bar{f}^{(t)}(\cdot|s), \bar{f}^{(t+1)}(\cdot|s)\right) \ge 0. \qquad (284)$$

$$\langle \Phi(s)\hat{\boldsymbol{w}}^{(t)}, \bar{f}^{(t+1)}(\cdot|s) - \bar{f}^{(t)}(\cdot|s) \rangle + \langle \Phi(s)\hat{\boldsymbol{w}}^{(t)}, \bar{f}^{(t)}(\cdot|s) - \pi^\star(\cdot|s) \rangle$$
$$\ge -\frac{1}{\alpha} D\left(\pi^\star(\cdot|s), \bar{f}^{(t)}(\cdot|s)\right) + \frac{1}{\alpha} D\left(\pi^\star(\cdot|s), \bar{f}^{(t+1)}(\cdot|s)\right). \qquad (285)$$

Taking expectation w.r.t. distribution $d^\star$ on both sides of (285), we arrive at

$$\mathbb{E}_{s \sim d^\star}\left[\langle \Phi(s)\hat{\boldsymbol{w}}^{(t)}, \bar{f}^{(t+1)}(\cdot|s) - \bar{f}^{(t)}(\cdot|s) \rangle\right] + \mathbb{E}_{s \sim d^\star}\left[\langle \Phi(s)\hat{\boldsymbol{w}}^{(t)}, \bar{f}^{(t)}(\cdot|s) - \pi^\star(\cdot|s) \rangle\right] \ge \frac{1}{\alpha}(D_\star^{(t+1)} - D_\star^{(t)}). \qquad (286)$$

To simplify the notation we let $\bar{Q}^{(t)}$ and $\bar{V}^{(t)}$ denote $Q^{\bar{f}^{(t)}}$ and $V^{\bar{f}^{(t)}}$, respectively. Note that the first expectation in the above expression (286) could be upper bounded as follows:

$$\mathbb{E}_{s \sim d^\star}\left[\langle \Phi(s)\hat{\boldsymbol{w}}^{(t)}, \bar{f}^{(t+1)}(\cdot|s) - \bar{f}^{(t)}(\cdot|s) \rangle\right]$$
$$= \sum_{s \in \mathcal{S}} d^\star(s) \langle \Phi(s)\hat{\boldsymbol{w}}^{(t)}, \bar{f}^{(t+1)}(\cdot|s) - \bar{f}^{(t)}(\cdot|s) \rangle$$
$$= \sum_{s \in \mathcal{S}} \frac{d^\star(s)}{d^{\bar{f}^{(k+1)}}(s)} d^{\bar{f}^{(k+1)}}(s) \langle \Phi(s)\hat{\boldsymbol{w}}^{(t)}, \bar{f}^{(t+1)}(\cdot|s) - \bar{f}^{(t)}(\cdot|s) \rangle$$
$$\le \vartheta_\rho \sum_{s \in \mathcal{S}} d^{\bar{f}^{(k+1)}}(s) \langle \Phi(s)\hat{\boldsymbol{w}}^{(t)}, \bar{f}^{(t+1)}(\cdot|s) - \bar{f}^{(t)}(\cdot|s) \rangle$$
$$= \vartheta_\rho \sum_{s \in \mathcal{S}} d^{\bar{f}^{(k+1)}}(s) \langle \bar{Q}^{(t)}(s,\cdot), \bar{f}^{(t+1)}(\cdot|s) - \bar{f}^{(t)}(\cdot|s) \rangle + \vartheta_\rho \sum_{s \in \mathcal{S}} d^{\bar{f}^{(k+1)}}(s) \langle \bar{\Phi}(s)\hat{\boldsymbol{w}}^{(t)} - \bar{Q}^{(t)}(s,\cdot), \bar{f}^{(t+1)}(\cdot|s) - \bar{f}^{(t)}(\cdot|s) \rangle$$
$$= \vartheta_\rho (1-\gamma)\left(\bar{V}^{(t+1)}(\rho) - \bar{V}^{(t)}(\rho)\right) + \vartheta_\rho \sum_{s \in \mathcal{S}} d^{\bar{f}^{(k+1)}}(s) \langle \bar{\Phi}(s)\hat{\boldsymbol{w}}^{(t)} - \bar{Q}^{(t)}(s,\cdot), \bar{f}^{(t+1)}(\cdot|s) - \bar{f}^{(t)}(\cdot|s) \rangle, \qquad (287)$$

where the first inequality uses (**??**) and the definition of $\vartheta_\rho$ (107) and the last line follows from (260) in Lemma G.1. We separate the second term of the last line into four terms as follows:

$$
\sum_{s\in\mathcal{S}} d^{\bar{f}^{(t+1)}}(s)\langle\bar{\Phi}(s)\hat{\boldsymbol{w}}^{(t)} - \bar{Q}^{(t)}(s,\cdot), \bar{f}^{(t+1)}(\cdot|s) - \bar{f}^{(t)}(\cdot|s)\rangle
$$
$$
= \underbrace{\sum_{s\in\mathcal{S}}\sum_{a\in\mathcal{A}} d^{\bar{f}^{(t+1)}}(s)\bar{f}^{(t+1)}(a|s)\phi^\top(s,a)(\hat{\boldsymbol{w}}^{(t)} - \hat{\boldsymbol{w}}_\star^{(t)})}_{(I)} + \underbrace{\sum_{s\in\mathcal{S}}\sum_{a\in\mathcal{A}} d^{\bar{f}^{(t+1)}}(s)\bar{f}^{(t+1)}(a|s)\left(\phi^\top(s,a)\hat{\boldsymbol{w}}_\star^{(t)} - \bar{Q}^{(t)}(s,a)\right)}_{(II)}
$$
$$
+ \underbrace{\sum_{s\in\mathcal{S}}\sum_{a\in\mathcal{A}} d^{\bar{f}^{(t+1)}}(s)\bar{f}^{(t)}(a|s)\phi^\top(s,a)(\hat{\boldsymbol{w}}_\star^{(t)} - \hat{\boldsymbol{w}}^{(t)})}_{(III)} + \underbrace{\sum_{s\in\mathcal{S}}\sum_{a\in\mathcal{A}} d^{\bar{f}^{(t+1)}}(s)\bar{f}^{(t)}(a|s)\left(\bar{Q}^{(t)}(s,a) - \phi^\top(s,a)\hat{\boldsymbol{w}}_\star^{(t)}\right)}_{(IV)} .
$$

$$(288)$$

Applying again Lemma G.1, we deduce the equivalent form of the second expectation in (286) as follows:

$$
\mathbb{E}_{s\sim d^\star}\left[\langle\Phi(s)\hat{\boldsymbol{w}}^{(t)}, \bar{f}^{(t)}(\cdot|s) - \pi^\star(\cdot|s)\rangle\right]
$$
$$
= \mathbb{E}_{s\sim d^\star}\left[\langle\bar{Q}^{(t)}(s,\cdot), \bar{f}^{(t)}(\cdot|s) - \pi^\star(\cdot|s)\rangle\right] + \mathbb{E}_{s\sim d^\star}\left[\langle\Phi(s)\hat{\boldsymbol{w}}^{(t)} - \bar{Q}^{(t)}(s,\cdot), \bar{f}^{(t)}(\cdot|s) - \pi^\star(\cdot|s)\rangle\right]
$$
$$
= (1-\gamma)\left(\bar{V}^{(t)}(\rho) - V^{\pi^\star}(\rho)\right) + \mathbb{E}_{s\sim d^\star}\left[\langle\Phi(s)\hat{\boldsymbol{w}}^{(t)} - \bar{Q}^{(t)}(s,\cdot), \bar{f}^{(t)}(\cdot|s) - \pi^\star(\cdot|s)\rangle\right] , \quad (289)
$$

where the second term of the last line could be decomposed into the following terms:

$$
\mathbb{E}_{s\sim d^\star}\left[\langle\Phi(s)\hat{\boldsymbol{w}}^{(t)} - \bar{Q}^{(t)}(s,\cdot), \bar{f}^{(t)}(\cdot|s) - \pi^\star(\cdot|s)\rangle\right]
$$
$$
= \underbrace{\sum_{s\in\mathcal{S}}\sum_{a\in\mathcal{A}} d^\star(s)\bar{f}^{(t)}(a|s)\phi^\top(s,a)(\hat{\boldsymbol{w}}^{(t)} - \hat{\boldsymbol{w}}_\star^{(t)})}_{(A)} + \underbrace{\sum_{s\in\mathcal{S}}\sum_{a\in\mathcal{A}} d^\star(s)\bar{f}^{(t)}(a|s)\left(\phi^\top(s,a)\hat{\boldsymbol{w}}_\star^{(t)} - \bar{Q}^{(t)}(s,a)\right)}_{(B)}
$$
$$
+ \underbrace{\sum_{s\in\mathcal{S}}\sum_{a\in\mathcal{A}} d^\star(s)\pi^\star(a|s)\phi^\top(s,a)(\hat{\boldsymbol{w}}_\star^{(t)} - \hat{\boldsymbol{w}}^{(t)})}_{(C)} + \underbrace{\sum_{s\in\mathcal{S}}\sum_{a\in\mathcal{A}} d^\star(s)\pi^\star(a|s)\left(\bar{Q}^{(t)}(s,a) - \phi^\top(s,a)\hat{\boldsymbol{w}}_\star^{(t)}\right)}_{(D)} .
$$

$$(290)$$

Plugging (288), (290) into (287) and (289), resp., and making use of (286), we have

$$
\vartheta_\rho(1-\gamma)\left(\bar{V}^{(t+1)}(\rho) - \bar{V}^{(t)}(\rho)\right) + (1-\gamma)\left(\bar{V}^{(t)}(\rho) - V^{\pi^\star}(\rho)\right)
$$
$$
+ \vartheta_\rho\bigg(\underbrace{\sum_{s\in\mathcal{S}}\sum_{a\in\mathcal{A}} d^{\bar{f}^{(t+1)}}(s)\bar{f}^{(t+1)}(a|s)\phi^\top(s,a)(\hat{\boldsymbol{w}}^{(t)} - \hat{\boldsymbol{w}}_\star^{(t)})}_{(I)} + \underbrace{\sum_{s\in\mathcal{S}}\sum_{a\in\mathcal{A}} d^{\bar{f}^{(t+1)}}(s)\bar{f}^{(t+1)}(a|s)\left(\phi^\top(s,a)\hat{\boldsymbol{w}}_\star^{(t)} - \bar{Q}^{(t)}(s,a)\right)}_{(II)}
$$
$$
+ \underbrace{\sum_{s\in\mathcal{S}}\sum_{a\in\mathcal{A}} d^{\bar{f}^{(t+1)}}(s)\bar{f}^{(t)}(a|s)\phi^\top(s,a)(\hat{\boldsymbol{w}}_\star^{(t)} - \hat{\boldsymbol{w}}^{(t)})}_{(III)} + \underbrace{\sum_{s\in\mathcal{S}}\sum_{a\in\mathcal{A}} d^{\bar{f}^{(t+1)}}(s)\bar{f}^{(t)}(a|s)\left(\bar{Q}^{(t)}(s,a) - \phi^\top(s,a)\hat{\boldsymbol{w}}_\star^{(t)}\right)}_{(IV)}\bigg)
$$
$$
+ \underbrace{\sum_{s\in\mathcal{S}}\sum_{a\in\mathcal{A}} d^\star(s)\bar{f}^{(t)}(a|s)\phi^\top(s,a)(\hat{\boldsymbol{w}}^{(t)} - \hat{\boldsymbol{w}}_\star^{(t)})}_{(A)} + \underbrace{\sum_{s\in\mathcal{S}}\sum_{a\in\mathcal{A}} d^\star(s)\bar{f}^{(t)}(a|s)\left(\phi^\top(s,a)\hat{\boldsymbol{w}}_\star^{(t)} - \bar{Q}^{(t)}(s,a)\right)}_{(B)}
$$
$$
+ \underbrace{\sum_{s\in\mathcal{S}}\sum_{a\in\mathcal{A}} d^\star(s)\pi^\star(a|s)\phi^\top(s,a)(\hat{\boldsymbol{w}}_\star^{(t)} - \hat{\boldsymbol{w}}^{(t)})}_{(C)} + \underbrace{\sum_{s\in\mathcal{S}}\sum_{a\in\mathcal{A}} d^\star(s)\pi^\star(a|s)\left(\bar{Q}^{(t)}(s,a) - \phi^\top(s,a)\hat{\boldsymbol{w}}_\star^{(t)}\right)}_{(D)}
$$
$$
\geq \frac{1}{\alpha}(D_\star^{(t+1)} - D_\star^{(t)}).
$$

$$(291)$$

Below we upper bound $|(I)|$-$|(IV)|$ and $|(A)|$-$|(D)|$.

For any $t \in \mathbb{N}$ and $n \in [N]$, we define matrix $\Sigma_{\tilde{d}_n^{(t)}} \in \mathbb{R}^{p \times p}$ as

$$\Sigma_{\tilde{d}_n^{(t)}} := \mathbb{E}_{(s,a) \sim \tilde{d}_n^{(t)}} \left[ \phi(s,a) \phi^\top(s,a) \right] , \tag{292}$$

and we define

$$\varepsilon_{\text{stat},n}^{(t)} := \ell \left( \boldsymbol{w}_n^{(t)}, Q_n^{(t)}, \tilde{d}_n^{(t)} \right) - \ell \left( \boldsymbol{w}_{\star,n}^{(t)}, Q_n^{(t)}, \tilde{d}_n^{(t)} \right) , \tag{293}$$

$$\varepsilon_{\text{approx},n}^{(t)} := \ell \left( \boldsymbol{w}_{\star,n}^{(t)}, Q_n^{(t)}, \tilde{d}_n^{(t)} \right) , \tag{294}$$

then for all $n \in [N]$, by Assumption E.2 and Assumption 4.2 we have

$$\mathbb{E} \left[ \varepsilon_{\text{stat},n}^{(t)} \right] \le \varepsilon_{\text{stat}}^n , \quad \text{and} \quad \mathbb{E} \left[ \varepsilon_{\text{approx},n}^{(t)} \right] \le \varepsilon_{\text{approx}}^n . \tag{295}$$

We let $\bar{\varepsilon}_{\text{stat}}^{(t)} := \frac{1}{N} \sum_{n=1}^N \varepsilon_{\text{stat},n}^{(t)}$ and $\bar{\varepsilon}_{\text{approx}}^{(t)} := \frac{1}{N} \sum_{n=1}^N \varepsilon_{\text{approx},n}^{(t)}$. By Cauchy-Schwartz's inequality we have

$$|(I)| \le \frac{1}{N} \sum_{n=1}^N \sum_{(s,a) \in \mathcal{S} \times \mathcal{A}} d^{\bar{f}^{(t+1)}}(s) \bar{f}^{(t+1)}(a|s) |\phi^\top(s,a)(\boldsymbol{w}_n^{(t)} - \boldsymbol{w}_{\star,n}^{(t)})|$$

$$\le \frac{1}{N} \sum_{n=1}^N \sqrt{ \sum_{(s,a) \in \mathcal{S} \times \mathcal{A}} \frac{\left( d^{\bar{f}^{(t+1)}}(s) \right)^2 \left( \bar{f}^{(t+1)}(a|s) \right)^2}{\tilde{d}_n^{(t)}(s,a)} \cdot \sum_{(s,a) \in \mathcal{S} \times \mathcal{A}} \tilde{d}_n^{(t)}(s,a) \left( \phi^\top(s,a)(\boldsymbol{w}_n^{(t)} - \boldsymbol{w}_{\star,n}^{(t)}) \right)^2 }$$

$$= \frac{1}{N} \sum_{n=1}^N \sqrt{ \mathbb{E}_{(s,a) \sim \tilde{d}_n^{(t)}} \left[ \left( \frac{\left( d^{\bar{f}^{(t+1)}}(s) \right) \left( \bar{f}^{(t+1)}(a|s) \right)}{\tilde{d}_n^{(t)}(s,a)} \right)^2 \right] \left\| \boldsymbol{w}_n^{(t)} - \boldsymbol{w}_{\star,n}^{(t)} \right\|_{\Sigma_{\tilde{d}_n^{(t)}}}^2 }$$

$$\le \frac{1}{N} \sum_{n=1}^N \sqrt{ C_\nu \left\| \boldsymbol{w}_n^{(t)} - \boldsymbol{w}_{\star,n}^{(t)} \right\|_{\Sigma_{\tilde{d}_n^{(t)}}}^2 }$$

$$\le \frac{1}{N} \sum_{n=1}^N \sqrt{ C_\nu \varepsilon_{\text{stat},n}^{(t)} } \le \sqrt{ C_\nu \bar{\varepsilon}_{\text{stat}}^{(t)} } , \tag{296}$$

where the third inequality follows from Assumption 4.3, the last inequality uses Jensen's inequality, and the penultimate inequality by Assumption E.2 and by noticing that for all $\boldsymbol{w} \in \mathbb{R}^p$, we have

$$\ell \left( \boldsymbol{w}, Q_n^{(t)}, \tilde{d}_n^{(t)} \right) - \ell \left( \boldsymbol{w}_{\star,n}^{(t)}, Q_n^{(t)}, \tilde{d}_n^{(t)} \right)$$

$$= \mathbb{E}_{(s,a) \sim \tilde{d}_n^{(t)}} \left[ \left( \phi^\top(s,a)\boldsymbol{w} - \phi^\top(s,a)\boldsymbol{w}_{\star,n}^{(t)} + \phi^\top(s,a)\boldsymbol{w}_{\star,n}^{(t)} - Q_n^{(t)}(s,a) \right)^2 \right] - \ell \left( \boldsymbol{w}_{\star,n}^{(t)}, Q_n^{(t)}, \tilde{d}_n^{(t)} \right)$$

$$= \mathbb{E}_{(s,a) \sim \tilde{d}_n^{(t)}} \left[ \left( \phi^\top(s,a)\boldsymbol{w} - \phi^\top(s,a)\boldsymbol{w}_{\star,n}^{(t)} \right)^2 \right] + 2 \left( \boldsymbol{w} - \boldsymbol{w}_{\star,n}^{(t)} \right)^\top \mathbb{E}_{(s,a) \sim \tilde{d}_n^{(t)}} \left[ \left( \phi^\top(s,a)\boldsymbol{w}_{\star,n}^{(t)} - Q_n^{(t)}(s,a) \right) \phi(s,a) \right]$$

$$= \left\| \boldsymbol{w} - \boldsymbol{w}_{\star,n}^{(t)} \right\|_{\Sigma_{\tilde{d}_n^{(t)}}} + \left( \boldsymbol{w} - \boldsymbol{w}_{\star,n}^{(t)} \right)^\top \nabla_w \ell \left( \boldsymbol{w}_{\star,n}^{(t)}, Q_n^{(t)}, \tilde{d}_n^{(t)} \right)$$

$$\ge \left\| \boldsymbol{w} - \boldsymbol{w}_{\star,n}^{(t)} \right\|_{\Sigma_{\tilde{d}_n^{(t)}}} , \tag{297}$$

where the last line follows from the first-order optimality condition for the minimum point $\boldsymbol{w}_{\star,n}^{(t)} \in \arg\min_w \ell \left( \boldsymbol{w}, Q_n^{(t)}, \tilde{d}_n^{(t)} \right)$:

$$\forall \boldsymbol{w} \in \mathbb{R}^p : \quad \left( \boldsymbol{w} - \boldsymbol{w}_{\star,n}^{(t)} \right)^\top \nabla_w \ell \left( \boldsymbol{w}_{\star,n}^{(t)}, Q_n^{(t)}, \tilde{d}_n^{(t)} \right) \ge 0.$$

$\square$

Analogous to bounding $|(I)|$, by simply substituting $\bar{f}^{(t+1)}$ with $\bar{f}^{(t)}$ or $\pi^\star$ or substituting $d^{\bar{f}^{(t+1)}}$ into $d^\star$, we obtain the same upper bound for $|(III)|$, $|(A)|$ and $|(C)|$, i.e.,

$$|(III)|, |(A)|, |(C)| \le \sqrt{ C_\nu \bar{\varepsilon}_{\text{stat}}^{(t)} } . \tag{298}$$

Now we upper bound $|(II)|$ as follows:

$$|(II)| \leq \frac{1}{N} \sum_{n=1}^{N} \sum_{(s,a) \in \mathcal{S} \times \mathcal{A}} d^{\bar{f}^{(t+1)}}(s) \bar{f}^{(t+1)}(a|s) \left( |\phi^{\top}(s,a) \boldsymbol{w}_{\star,n}^{(t)} - Q_n^{(t)}(s,a)| + |Q_n^{(t)}(s,a) - \bar{Q}^{(t)}(s,a)| \right)$$

$$\leq \frac{1}{N} \sum_{n=1}^{N} \sqrt{\sum_{(s,a) \in \mathcal{S} \times \mathcal{A}} \frac{\left(d^{\bar{f}^{(t+1)}}(s)\right)^2 \left(\bar{f}^{(t+1)}(a|s)\right)^2}{\tilde{d}_n^{(t)}(s,a)}} \cdot$$

$$\cdot \sqrt{2 \sum_{(s,a) \in \mathcal{S} \times \mathcal{A}} \tilde{d}_n^{(t)}(s,a) \left( \left(\phi^{\top}(s,a) \boldsymbol{w}_{\star,n}^{(t)} - Q_n^{(t)}(s,a)\right)^2 + \left(Q_n^{(t)}(s,a) - \bar{Q}^{(t)}(s,a)\right)^2 \right)}$$

$$= \frac{1}{N} \sum_{n=1}^{N} \sqrt{\mathbb{E}_{(s,a) \sim \tilde{d}_n^{(t)}} \left[ \left( \frac{\left(d^{\bar{f}^{(t+1)}}(s)\right)\left(\bar{f}^{(t+1)}(a|s)\right)}{\tilde{d}_n^{(t)}(s,a)} \right)^2 \right] \cdot 2 \left( \varepsilon_{\text{approx},n}^{(t)} + L_Q^2 \left\| \boldsymbol{\xi}_n^{(t)} - \bar{\boldsymbol{\xi}}^{(t)} \right\|_2^2 \right)}$$

$$\leq \sqrt{2 C_\nu \left( \bar{\varepsilon}_{\text{approx}}^{(t)} + \frac{L_Q^2}{N} \left\| \boldsymbol{\xi}^{(t)} - \mathbf{1}(\bar{\boldsymbol{\xi}}^{(t)})^{\top} \right\|_F^2 \right)}, \tag{299}$$

where $L_Q$ is defined in Lemma E.7, the second line uses Cauchy-Schwartz's inequality and Young's inequality (117) and the last inequality uses Assumption 4.3 and Jensen's inequality.

Analogous to bounding $|(II)|$, by simply substituting $\bar{f}^{(t+1)}$ with $\bar{f}^{(t)}$ or $\pi^\star$ or substituting $d^{\bar{f}^{(t+1)}}$ into $d^\star$, we obtain the same upper bound for $|(IV)|, |(B)|$ and $|(D)|$, i.e.,

$$|(IV)|, |(B)|, |(D)| \leq \sqrt{2 C_\nu \left( \bar{\varepsilon}_{\text{approx}}^{(t)} + \frac{L_Q^2}{N} \left\| \boldsymbol{\xi}^{(t)} - \mathbf{1}(\bar{\boldsymbol{\xi}}^{(t)})^{\top} \right\|_F^2 \right)}. \tag{300}$$

Plugging (296),(298),(299),(300) into (291) and dividing both sides by $(1 - \gamma)$ yield

$$\vartheta_\rho \left( \delta^{(t+1)} - \delta^{(t)} \right) + \delta^{(t)} \leq \frac{D_\star^{(t)}}{(1-\gamma)\alpha} - \frac{D_\star^{(t+1)}}{(1-\gamma)\alpha} + \frac{2\sqrt{C_\nu}(\vartheta+1)}{1-\gamma} \left( \sqrt{\bar{\varepsilon}_{\text{stat}}^{(t)}} + \sqrt{2 \left( \bar{\varepsilon}_{\text{approx}}^{(t)} + \frac{L_Q^2}{N} \left\| \boldsymbol{\xi}^{(t)} - \mathbf{1}(\bar{\boldsymbol{\xi}}^{(t)})^{\top} \right\|_F^2 \right)} \right).$$

Taking expectation on both sides of the above expression and making use of the simple fact that

$$\mathbb{E}\left[ \sqrt{x} \right] \leq \sqrt{\mathbb{E}[x]},$$

we reach the conclusion (123).

## G.8 Proof of Lemma E.9

*Proof of Lemma E.9.* For any $\zeta > 0$, by the actor update rule (34) and (114) we have that

$$\left\| \boldsymbol{\xi}^{(t+1)} - \mathbf{1}_N \bar{\boldsymbol{\xi}}^{(t+1)\top} \right\|_F^2 = \left\| \boldsymbol{W}(\boldsymbol{\xi}^{(t)} + \alpha \boldsymbol{h}^{(t)}) - \mathbf{1}_N (\bar{\boldsymbol{\xi}}^{(t)} + \alpha \hat{\boldsymbol{w}}^{(t)})^{\top} \right\|_F^2$$

$$\leq (1+\zeta)\sigma^2 \left\| \boldsymbol{\xi}^{(t)} - \mathbf{1}_N \bar{\boldsymbol{\xi}}^{(t)\top} \right\|_F^2 + \alpha^2(1+1/\zeta)\sigma^2 \left\| \boldsymbol{h}^{(t)} - \mathbf{1}_N \hat{\boldsymbol{w}}^{(t)\top} \right\|_F^2, \tag{301}$$

where the last line follows from Young's inequality (116) and (11). By the gradient tracking step (33), Young's inequality (116) and (11), we have

$$\left\| \boldsymbol{h}^{(t+1)} - \mathbf{1}\hat{\boldsymbol{w}}^{(t+1)\top} \right\|_F^2 = \left\| \boldsymbol{W}(\boldsymbol{h}^{(t)} + \boldsymbol{w}^{(t+1)} - \boldsymbol{w}^{(t)}) - \mathbf{1}\hat{\boldsymbol{w}}^{(t)\top} + \mathbf{1}(\hat{\boldsymbol{w}}^{(t)\top} - \hat{\boldsymbol{w}}^{(t+1)\top}) \right\|_F^2$$

$$= \left\| \boldsymbol{W}\boldsymbol{h}^{(t)} - \mathbf{1}\hat{\boldsymbol{w}}^{(t)\top} + \boldsymbol{W}(\boldsymbol{w}^{(t+1)} - \boldsymbol{w}^{(t)}) - \mathbf{1}(\hat{\boldsymbol{w}}^{(t+1)\top} - \hat{\boldsymbol{w}}^{(t)\top}) \right\|_F^2$$

$$\leq (1+\zeta)\sigma^2 \left\| \boldsymbol{h}^{(t)} - \mathbf{1}_N \hat{\boldsymbol{w}}^{(t)\top} \right\| + (1+1/\zeta)\sigma^2 \left\| \boldsymbol{w}^{(t+1)} - \boldsymbol{w}^{(t)} - \mathbf{1}(\hat{\boldsymbol{w}}^{(t+1)\top} - \hat{\boldsymbol{w}}^{(t)\top}) \right\|_F^2$$

$$\leq (1+\zeta)\sigma^2 \left\| \boldsymbol{h}^{(t)} - \mathbf{1}_N \hat{\boldsymbol{w}}^{(t)\top} \right\| + (1+1/\zeta)\sigma^2 \left\| \boldsymbol{w}^{(t+1)} - \boldsymbol{w}^{(t)} \right\|_F^2, \tag{302}$$

where the last inequality follows from the fact

$$\left\| \boldsymbol{w}^{(t+1)} - \boldsymbol{w}^{(t)} - \mathbf{1}(\hat{\boldsymbol{w}}^{(t+1)\top} - \hat{\boldsymbol{w}}^{(t)\top}) \right\|_{\mathrm{F}}^2$$

$$= \left\| \boldsymbol{w}^{(t+1)} - \boldsymbol{w}^{(t)} \right\|_{\mathrm{F}}^2 + N \left\| \hat{\boldsymbol{w}}^{(t+1)} - \hat{\boldsymbol{w}}^{(t)} \right\|_2^2 - 2 \sum_{n=1}^{N} \langle \boldsymbol{w}_n^{(t+1)} - \boldsymbol{w}_n^{(t)}, \hat{\boldsymbol{w}}^{(t+1)} - \hat{\boldsymbol{w}}^{(t)} \rangle$$

$$= \left\| \boldsymbol{w}^{(t+1)} - \boldsymbol{w}^{(t)} \right\|_{\mathrm{F}}^2 - N \left\| \hat{\boldsymbol{w}}^{(t+1)} - \hat{\boldsymbol{w}}^{(t)} \right\|_2^2$$

$$\leq \left\| \boldsymbol{w}^{(t+1)} - \boldsymbol{w}^{(t)} \right\|_{\mathrm{F}}^2 . \tag{303}$$

Then for any $n \in [N]$, $t \in \mathbb{N}$ and $\boldsymbol{w} \in \mathbb{R}^p$, we have

$$\ell(\boldsymbol{w}, Q_n^{(t)}, \tilde{d}_n^{(t)}) - \ell(\boldsymbol{w}_{\star,n}^{(t)}, Q_n^{(t)}, \tilde{d}_n^{(t)})$$

$$= \mathbb{E}_{(s,a)\sim\tilde{d}_n^{(t)}} \left[ \left( \phi^\top(s,a)\boldsymbol{w} - \phi^\top(s,a)\boldsymbol{w}_{\star,n}^{(t)} + \phi^\top(s,a)\boldsymbol{w}_{\star,n}^{(t)} - Q_n^{(t)}(s,a) \right)^2 \right] - \ell(\boldsymbol{w}_{\star,n}^{(t)}, Q_n^{(t)}, \tilde{d}_n^{(t)})$$

$$= \mathbb{E}_{(s,a)\sim\tilde{d}_n^{(t)}} \left[ \left( \phi^\top(s,a)\boldsymbol{w} - \phi^\top(s,a)\boldsymbol{w}_{\star,n}^{(t)} \right)^2 \right] + 2(\boldsymbol{w} - \boldsymbol{w}_{\star,n}^{(t)})^\top \mathbb{E}_{(s,a)\sim\tilde{d}_n^{(t)}} \left[ \left( \phi^\top(s,a)\boldsymbol{w}_{\star,n}^{(t)} - Q_n^{(t)}(s,a) \right) \phi(s,a) \right]$$

$$= \left\| \boldsymbol{w} - \boldsymbol{w}_{\star,n}^{(t)} \right\|_{\Sigma_{\tilde{d}_n^{(t)}}}^2 + (\boldsymbol{w} - \boldsymbol{w}_{\star,n}^{(t)})^\top \nabla_w \ell(\boldsymbol{w}_{\star,n}^{(t)}, Q_n^{(t)}, \tilde{d}_n^{(t)})$$

$$\geq \left\| \boldsymbol{w} - \boldsymbol{w}_{\star,n}^{(t)} \right\|_{\Sigma_{\tilde{d}_n^{(t)}}}^2$$

$$\geq (1-\gamma)\mu \left\| \boldsymbol{w} - \boldsymbol{w}_{\star,n}^{(t)} \right\|_2^2, \tag{304}$$

where the penultimate line follows from the first-order optimality conditions for the optima $\boldsymbol{w}_{\star,n}^{(t)}$:

$$\forall \boldsymbol{w} \in \mathbb{R}^p : \quad (\boldsymbol{w} - \boldsymbol{w}_{\star,n}^{(t)})^\top \nabla_w \ell(\boldsymbol{w}_{\star,n}^{(t)}, Q_n^{(t)}, \tilde{d}_n^{(t)}) \geq 0 \tag{305}$$

and the last line is by Assumption 4.1 and (**??**).

Note that

$$\ell(\boldsymbol{w}_{\star,n}^{(t)}, Q_n^{(t+1)}, \tilde{d}_n^{(t+1)})$$

$$= \mathbb{E}_{(s,a)\sim\tilde{d}_n^{(t+1)}} \left[ (\phi^\top(s,a)\boldsymbol{w}_{\star,n}^{(t)} - Q_n^{(t+1)}(s,a))^2 \right]$$

$$\leq 2 \sum_{(s,a)\in\mathcal{S}\times\mathcal{A}} \tilde{d}_n^{(t)}(s,a) \frac{\tilde{d}_n^{(t+1)}(s,a)}{\tilde{d}_n^{(t)}(s,a)} (\phi^\top(s,a)\boldsymbol{w}_{\star,n}^{(t)} - Q_n^{(t)}(s,a))^2 + 2\mathbb{E}_{(s,a)\sim\tilde{d}_n^{(t+1)}} (Q_n^{(t+1)}(s,a) - Q_n^{(t)}(s,a))^2$$

$$\leq 2C_\nu \mathbb{E}_{(s,a)\sim\tilde{d}_n^{(t)}} (\phi^\top(s,a)\boldsymbol{w}_{\star,n}^{(t)} - Q_n^{(t)}(s,a))^2 + 2L_Q \left\| \boldsymbol{\xi}_n^{(t+1)} - \boldsymbol{\xi}_n^{(t)} \right\|_2^2$$

$$\leq 2C_\nu \varepsilon_{\text{approx}}^n + 2L_Q^2 \left\| \boldsymbol{\xi}_n^{(t+1)} - \boldsymbol{\xi}_n^{(t)} \right\|_2^2, \tag{306}$$

where the second inequality uses Assumption 4.3 and Lemma E.7, and the last line uses Assumption 4.2.

The above equation (306) together with (304) gives

$$\left\| \boldsymbol{w}_\star^{(t+1)} - \boldsymbol{w}_\star^{(t)} \right\|_{\mathrm{F}}^2 = \sum_{n=1}^{N} \left\| \boldsymbol{w}_{\star,n}^{(t+1)} - \boldsymbol{w}_{\star,n}^{(t)} \right\|_2^2$$

$$\leq \frac{1}{(1-\gamma)\mu} \sum_{n=1}^{N} \left( \ell(\boldsymbol{w}_{\star,n}^{(t)}, Q_n^{(t+1)}, \tilde{d}_n^{(t+1)}) - \ell(\boldsymbol{w}_{\star,n}^{(t+1)}, Q_n^{(t+1)}, \tilde{d}_n^{(t+1)}) \right)$$

$$\leq \frac{2}{(1-\gamma)\mu} \left( C_\nu \sum_{n=1}^{N} \varepsilon_{\text{approx}}^n + L_Q^2 \left\| \boldsymbol{\xi}^{(t+1)} - \boldsymbol{\xi}^{(t)} \right\|_{\mathrm{F}}^2 \right). \tag{307}$$

where $\boldsymbol{w}_\star^{(t)} := (\boldsymbol{w}_1^{(t)}, \cdots, \boldsymbol{w}_N^{(t)})^\top$, $\forall t$.

Also note that by Assumption E.2 and (304) we have

$$\forall t \in \mathbb{N}: \quad \left\| \boldsymbol{w}^{(t)} - \boldsymbol{w}_\star^{(t)} \right\|_{\mathrm{F}}^2 \leq \frac{\sum_{n=1}^N \varepsilon_{\mathrm{stat}}^n}{(1-\gamma)\mu}. \tag{308}$$

Therefore, by (306) and (308) we have

$$\left\| \boldsymbol{w}^{(t+1)} - \boldsymbol{w}^{(t)} \right\|_{\mathrm{F}}^2 \leq 3 \left( \left\| \boldsymbol{w}^{(t+1)} - \boldsymbol{w}_\star^{(t+1)} \right\|_{\mathrm{F}}^2 + \left\| \boldsymbol{w}_\star^{(t+1)} - \boldsymbol{w}_\star^{(t)} \right\|_{\mathrm{F}}^2 + \left\| \boldsymbol{w}^{(t)} - \boldsymbol{w}_\star^{(t)} \right\|_{\mathrm{F}}^2 \right)$$

$$\leq \frac{6}{(1-\gamma)\mu} \left( N(C_\nu \bar{\varepsilon}_{\mathrm{approx}} + \bar{\varepsilon}_{\mathrm{stat}}) + L_Q^2 \left\| \boldsymbol{\xi}^{(t+1)} - \boldsymbol{\xi}^{(t)} \right\|_{\mathrm{F}}^2 \right). \tag{309}$$

where the first inequality uses Young's inequality (117).

Note that by the update rule (34), the double stochasticity of the mixing matrix $\boldsymbol{W}$ and the consensus property (11) we have

$$\left\| \boldsymbol{\xi}^{(t+1)} - \boldsymbol{\xi}^{(t)} \right\|_{\mathrm{F}}^2$$

$$= \left\| \boldsymbol{W}(\boldsymbol{\xi}^{(t)} + \alpha \boldsymbol{h}^{(t)}) - \boldsymbol{\xi}^{(t)} \right\|_{\mathrm{F}}^2$$

$$= \left\| (\boldsymbol{W} - \boldsymbol{I})(\boldsymbol{\xi}^{(t)} - \mathbf{1}_N \bar{\boldsymbol{\xi}}^{(t)\top}) + \alpha(\boldsymbol{W}\boldsymbol{h}^{(t)} - \mathbf{1}_N \hat{\boldsymbol{w}}^{(t)\top}) + \alpha\mathbf{1}(\hat{\boldsymbol{w}}^{(t)} - \hat{\boldsymbol{w}}_\star^{(t)})^\top + \mathbf{1}(\hat{\boldsymbol{w}}_\star^{(t)})^\top \right\|_{\mathrm{F}}^2$$

$$\leq 16 \left\| \boldsymbol{\xi}^{(t)} - \mathbf{1}_N \bar{\boldsymbol{\xi}}^{(t)\top} \right\|_{\mathrm{F}}^2 + 4\alpha^2 \sigma^2 \left\| \boldsymbol{h}^{(t)} - \mathbf{1}_N \hat{\boldsymbol{w}}^{(t)\top} \right\|_{\mathrm{F}}^2 + 4\alpha^2 N \left\| \hat{\boldsymbol{w}}^{(t)} - \hat{\boldsymbol{w}}_\star^{(t)} \right\|_2^2 + 4\alpha^2 N \left\| \hat{\boldsymbol{w}}_\star^{(t)} \right\|_2^2$$

$$\leq 16 \left\| \boldsymbol{\xi}^{(t)} - \mathbf{1}_N \bar{\boldsymbol{\xi}}^{(t)\top} \right\|_{\mathrm{F}}^2 + 4\alpha^2 \sigma^2 \left\| \boldsymbol{h}^{(t)} - \mathbf{1}_N \hat{\boldsymbol{w}}^{(t)\top} \right\|_{\mathrm{F}}^2 + 4\alpha^2 \sum_{n=1}^N \left\| \boldsymbol{w}_n^{(t)} - \boldsymbol{w}_{\star,n}^{(t)} \right\|_2^2 + 4\alpha^2 \sum_{n=1}^N \left\| \boldsymbol{w}_{\star,n}^{(t)} \right\|_2^2$$

$$\leq 16 \left\| \boldsymbol{\xi}^{(t)} - \mathbf{1}_N \bar{\boldsymbol{\xi}}^{(t)\top} \right\|_{\mathrm{F}}^2 + 4\alpha^2 \sigma^2 \left\| \boldsymbol{h}^{(t)} - \mathbf{1}_N \hat{\boldsymbol{w}}^{(t)\top} \right\|_{\mathrm{F}}^2 + \frac{4\alpha^2 N \bar{\varepsilon}_{\mathrm{stat}}}{(1-\gamma)\mu} + \frac{4\alpha^2 N C_\phi^2}{\mu^2 (1-\gamma)^4}, \tag{310}$$

where the penultimate line uses Jensen's inequality and the last line follows from (304), Assumption E.2 and (271).

Combining (310) and (309) with (302), we deduce

$$\left\| \boldsymbol{h}^{(t+1)} - \mathbf{1}\hat{\boldsymbol{w}}^{(t+1)\top} \right\|_{\mathrm{F}}^2$$

$$\leq (1 + 1/\zeta)\frac{96\sigma^2 L_Q^2}{(1-\gamma)\mu} \left\| \boldsymbol{\xi}^{(t)} - \mathbf{1}\bar{\boldsymbol{\xi}}^{(t)\top} \right\|_{\mathrm{F}}^2 + \sigma^2 \left( 1 + \zeta + (1 + 1/\zeta)\frac{24L_Q^2 \alpha^2}{(1-\gamma)\mu} \right) \left\| \boldsymbol{h}^{(t)} - \mathbf{1}\hat{\boldsymbol{w}}^{(t)\top} \right\|_{\mathrm{F}}^2$$

$$+ (1 + 1/\zeta)\frac{6\sigma^2}{(1-\gamma)\mu} \left( N(\bar{\varepsilon}_{\mathrm{stat}} + C_\nu \bar{\varepsilon}_{\mathrm{approx}}) + 4L_Q^2 \left( \frac{\alpha^2 N \bar{\varepsilon}_{\mathrm{stat}}}{(1-\gamma)\mu} + \frac{\alpha^2 N C_\phi^2}{\mu^2 (1-\gamma)^2} \right) \right). \tag{311}$$

Finally, (124) follows from taking expectations on both sides of (301) and (311). $\quad\square$

## H   Numerical experiments

**Experimental setup.** We study the empirical performance of FedNPG (Algorithm 1) and entropy-regularized FedNPG (Algorithm 2) on a $K \times K$ GridWorld problem. To be specific, the collective goal of $N$ agents is to learn a global optimal policy to follow a predetermined path which starts at the top left corner and ends at the bottom right corner. However, each agent only has access to partial information about the whole map: in figure 1 (where we take $N = 3$ and $K = 9$ as an example), agent $n$ explores on map $n$, $n \in [N]$. After taking an action, only when the agent is at the shaded positions can it get reward 1, otherwise it gets 0. We stipulate the action space of all agents to be $\mathcal{A} = \{\mathrm{right}, \mathrm{down}\}$, i.e. movement is allowed only to the right or down. If an agent takes an action that will lead it out of the boarder of the map, we stipulate the agent's state doesn't change and receive reward 0. Each agent starts at the top left corner. To learn a shared policy to follow the path, we aim to maximize the average value function of all agents.

**Results.** In the following we discuss the empirical results of our algorithms. In all the experiments, we fix the discounted factor $\gamma = 0.99$. In our experiments, we also don't require the mixing matrix to

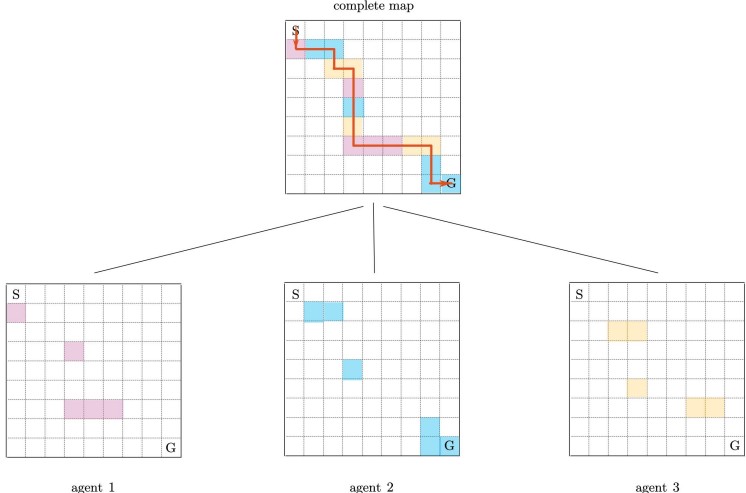

Figure 1: Gridworld experiement. $N$ agents ($N = 3$ here) aim to learn a shared policy to follow a predetermined path, which is the red dashed line in the complete map. Each agent only has access to partial information about the path and gets reward 1 only at the shaded positions and 0 at other positions. Each agent starts at the top left corner.

strictly adhere to Assumption 3.1. In Figure 2, we validate the effectiveness of vanilla FedNPG and entropy-regularized FedNPG across different map size $K$, where we set $\tau = 0, 0.005, 0.05, \eta = 0.1$, $N = 10$, and use a *standard ring graph* where agent $n$ receives information from agent $n + 1$ for $n \in [N-1]$, and agent $N$ receives information from agent 1, and we set all the weights on each edge of the communication graph to be 0.5. The corresponding mixing matrix of the standard ring graph is as follows:

$$\boldsymbol{W} = \begin{pmatrix} 0.5 & 0.5 & 0 & 0 & \cdots & 0 & 0 \\ 0 & 0.5 & 0.5 & 0 & \cdots & 0 & 0 \\ 0 & 0 & 0.5 & 0.5 & \cdots & 0 & 0 \\ \vdots & \vdots & \vdots & \vdots & & \vdots & \vdots \\ 0 & 0 & 0 & 0 & \cdots & 0.5 & 0.5 \\ 0.5 & 0 & 0 & 0 & \cdots & 0 & 0.5 \end{pmatrix}. \tag{312}$$

Here, $\boldsymbol{W}$ in (312) satisfies the double stochasticity assumption but is not symmetric.

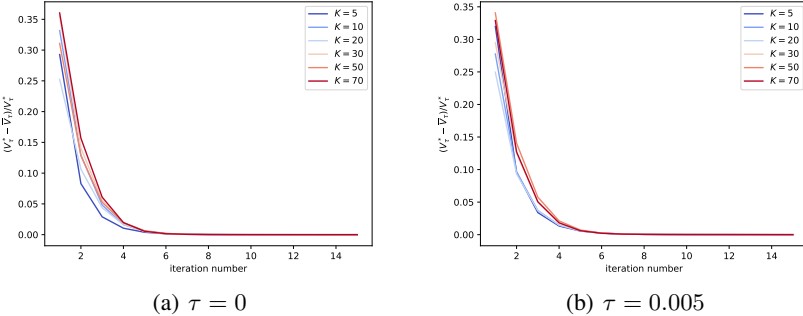

(a) $\tau = 0$        (b) $\tau = 0.005$

Figure 2: **Changing map size $K$.** we let $\tau = 0, 0.005$ and change $K$ for each $\tau$. We plot the curves of $(V_\tau^\star - \overline{V}_\tau^{(t)})/V_\tau^\star$ changing with the iteration number. We can see that both vanilla and entropy-regularized NPG converges to the optimal value function in a few iterations, and the convergence speed is almost the same across different $K$.

Figure 2 illustrates the normalized sub-optimality gap $(V_\tau^\star - \overline{V}_\tau^{(t)})/V_\tau^\star$ with respect to the iteration number. It can be seen that both vanilla and entropy-regularized NPG converge to the optimal value function in a few iterations, and the convergence speed is almost the same across different $K$, i.e. the impact of $K$ on the convergence speed is minimal.

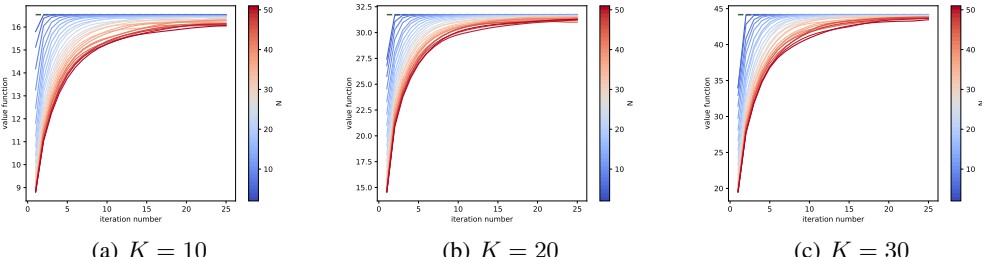

(a) $K = 10$  (b) $K = 20$  (c) $K = 30$

Figure 3: **Changing number of agents $N$.** we let $K = 10, 20, 30$ and change $N$ for each $K$. We plot the curves of value functions changing with the iteration number. The green dashed line represents the optimal value. We can see that the convergence speed decreases as $N$ increases. Same as before, the convergence speed is insensitive to the change of $K$.

In Figure 3, we study the performance of our algorithms when the number of agents $N$ varies. We set $K = 10, 20, 30$, $\tau = 0.005$, $\eta = 0.1$ and the communication graph to be the standard ring graph. We can see that the convergence speed decreases as $N$ increases. Same as before, the convergence speed is insensitive to the change of $K$.

In Figure 4, we illustrate the effect of the communication network topology to our algorithms. To be specific, we change the number of neighbors of each agent (i.e., the number of non-zero entries in each row of $\boldsymbol{W}$) and (i) randomly generalize the weights of the graph such that each row of $\boldsymbol{W}$ sum up to 1, i.e., $\boldsymbol{W}\boldsymbol{1} = \boldsymbol{1}$, see Figure 4(a); (ii) set the non-zero entries in each row of $\boldsymbol{W}$ all to be $\frac{1}{\text{number of neighbors}}$, see Figure 4(b). We fix $\eta = 0.1$, $K = 10$, $\tau = 0.005$. We plot the curves of value functions changing with the iteration number. The green dashed line represents the optimal value. For both 4(a) and 4(b), the convergence speed increase as number of neighbors of each agent increases. FedNPG performs better when using equal weights.

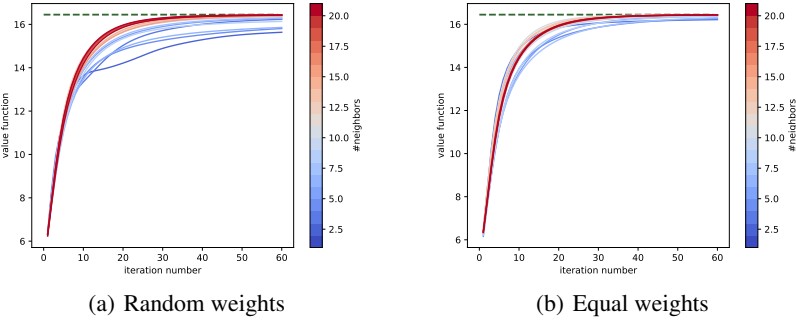

(a) Random weights  (b) Equal weights

Figure 4: **Changing communication network topology.** We change the number of neighbors of each agent. (i) In Figure 4(a), we randomly generalize the weights of the graph such that each row of $\boldsymbol{W}$ sum up to 1; (ii) In Figure 4(b), we set the non-zero entries in each row of $\boldsymbol{W}$ all to be $\frac{1}{\text{number of neighbors}}$. We plot the curves of value functions changing with the iteration number. The green dashed line represents the optimal value. For both 4(a) and 4(b), the convergence speed increase as number of neighbors increases. FedNPG performs better when using equal weights.

### H.1 Discussion on the Experiments

Note that even though there are many existing works in federated RL, none of the existing works, to the best of our knowledge, studies federated multi-tasks RL in the decentralized setting. Therefore,

we are not able to compare our work with existing works. However, here we include a comparison between FedNPG and a naïve baseline without the Q-tracking technique (line 6 in Algorithm 1).

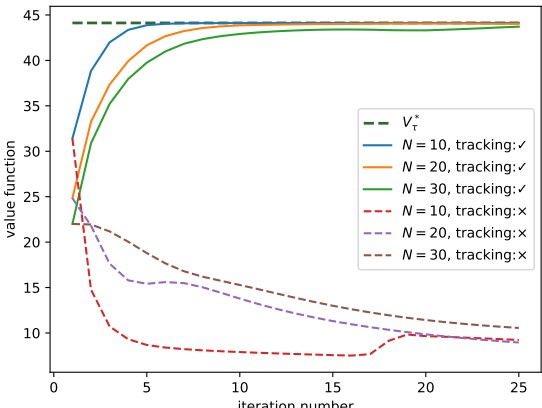

Figure 5: Comparison between FedNPG and a naïve baseline without the Q-tracking technique. The plot shows that while FedNPG converges within a few iterates, the algorithm without Q-tracking diverges, confirming the positive role of Q-tracking in ensuring convergence.

For this plot, we use the standard ring graph (Eq. 312). We fix the size of the maze $K = 30$, learning rate $\eta = 0.1$, and regularity coefficient $\tau = 0.005$. We experiment on different number of agents $N$ and plot the curves of value function changing with the iteration number. The plot shows that while FedNPG converges within a few iterates, the algorithm without Q-tracking diverges, confirming the positive role of Q-tracking in ensuring convergence.

