# OpenReview forum: "Federated Natural Policy Gradient and Actor Critic Methods for Multi-task Reinforcement Learning"
_NeurIPS.cc/2024/Conference — NeurIPS 2024 poster_

### Official Review · Reviewer_vHsi · 2024-07-09

**Soundness:** 3
**Presentation:** 3
**Contribution:** 3
**Rating:** 7
**Confidence:** 4

**Summary:**

This paper studies RL in the federated setting, where each agent only receives a local reward and communicates with other agents in a networked graph. The authors develop federated natural actor-critic in the tabular setting and the linear function approximation setting, with exact and inexact policy evaluation. In all cases, convergence rate guarantees are provided.

**Strengths:**

The presentation is mostly clear. The theoretical results are solid and seem to be the first in the federated setting.

**Weaknesses:**

(1) Calling the setting "multi-task RL" is a bit confusing. Since each agent has its own reward function, which can be misaligned with other agents' reward functions, assuming that all agents are willing to collaborate to reach a common goal (maximizing the averaged reward) can be unreasonable. For example, suppose there are only three agents, and $r_1=r_2=-r_3$. In this case, the optimal policy for the averaged reward is the same as the local optimal policy for agents 1 and 2, but it is the exact opposite of what agent 3 wants. Therefore, there is no reason to assume that agent 3 is willing to collaborate. Perhaps directly calling the setting "cooperative RL" is more suitable.

(2) The convergence rate of federated NAC seems to be much worse (especially in terms of the dependence on $1/(1-\gamma)$) compared with that of federated Q-learning. Is there a fundamental reason behind it?

(3) Suppose that we do not change the goal (which is to have global optimality with respect to the original model not the regularized one) but use regularization as a means for algorithm design. By letting $\tau$ go to zero at an appropriate rate, or choosing $\tau$ based on the required accuracy, what is the iteration and sample complexity of entropy-regularized NAC? Do we get $\epsilon^{-2}$ or something worse?

**Questions:**

See the section above.

**Limitations:**

Yes.

---

> ### Author Rebuttal · Authors · 2024-08-07
>
> ## Response to Reviewer vHsi
>
> Thank you for your time in reviewing our paper. We appreciate your positive feedback. Below We address your points. If our responses adequately address your concerns, we would be grateful if you could consider increasing your current score. And we are happy to answer your additional questions.
>
> >**(W1) regarding the name "multi-task RL"**
>
> We understand your concern about the potential confusion between "multi-task RL" and the cooperative nature of our setting. However, we believe that "multi-task RL" and "cooperative RL" are not mutually exclusive terms, as they describe different aspects of the problem:
>
> 1. "Multi-task RL" refers to the fact that each agent has a different reward function, corresponding to different tasks or objectives within the same environment.
> 2. "Cooperative" describes the agents' willingness to work together towards a common goal, in this case maximizing the averaged reward
>
> We have explicitly specified in our paper that we are considering a cooperative setting within the multi-task framework. This approach is consistent with existing literature in the field. For example, [1] and [2] consider respectively the cooperative and adversarial settings within multi-task RL, demonstrating that the term "multi-task" does not inherently imply either cooperation or competition.
>
> >**(W2) The convergence rate of federated NAC seems to be much worse (especially in terms of the dependence on $1/(1-\gamma)$) compared with that of federated Q-learning. Is there a fundamental reason behind it?**
>
> We want to clarify that our setting is different that of Federated Q-learning (e.g, Woo et al. 2023). Woo et al. 2023 considers the server-client setting, where all the agents share the same environment (both transition kernel and reward function), so the local value functions are in fact the same and no environment heterogeneity is tackled.
>
> >**Suppose that we do not change the goal (which is to have global optimality with respect to the original model not the regularized one) but use regularization as a means for algorithm design. By letting $\tau$ go to zero at an appropriate rate, or choosing $\tau$ based on the required accuracy, what is the iteration and sample complexity of entropy-regularized NAC? Do we get $\varepsilon^{-2}$ or something worse?**
>
> Thank you for your question.
> Setting $\tau=\frac{(1-\gamma)\varepsilon}{4\log |A|}$ guarantees $V^\star_\tau=V^\star$ and ensures we do not change the goal (see (25) in [3]). Here we adopt this choice of $\tau$ and let $\eta=\eta_0$ defined in Theorem 3.6. Then similar as in Remark D.1 in our paper, by employing fresh samples for the policy evaluation of each agent at every iteration, set $\varepsilon_q$ in Theorem 3.8 to be $\varepsilon_q = \mathcal{O}\left(\frac{\tau^2\varepsilon}{\gamma^2N\sigma^2+(1-\sigma)^2\tau^2(1-\gamma)^6}\right),$ and invoke the union bound over the iteration complexity $\widetilde{\mathcal{O}} \left(\frac{N\sigma^2}{\tau^2}+\frac{1}{1-\gamma}\right)$, we could give a loose upper bound of sample complexity of entropy-regularized FedNPG as
> $$\widetilde{\mathcal{O}} \left(\frac{N(N\sigma^2+\tau^2)^2}{\tau^4\varepsilon^2}\left(\frac{N\sigma^2}{\tau^2}+\frac{1}{1-\gamma}\right)\right).$$
>
> Here we only highlight $\varepsilon$-dependency to simplify the expressions.
>
> - When $\sigma=0$ (the server-client setting), the above complexity becomes $\widetilde{\mathcal{O}} \left(\varepsilon^{-2}\right)$, recovering that of single-agent entropy-regularized NPG ([3]).
>
> - When $\sigma$ is close to 1, the above complexity becomes $\widetilde{\mathcal{O}} \left(\varepsilon^{-8}\right)$, which is worse than the bound for inexact vanilla FedNPG $\widetilde{\mathcal{O}} \left(\varepsilon^{-3.5}\right)$ presented in (19). Therefore, FedNAC focused on unregularized case as it has better rates when the goal is to solve the unregulairzed problem.
>
> - The reason why the bound for the entropy-regularized case is worse may be due to proof artifacts as we need to address the complicated interplay between consensus errors and optimization errors (recall that simiar to the single-agent case, in our federated setting, vanilla FedNPG and entropy-regularzied FedNPG also require different proof frameworks), and we leave tightening the bound for future work.
>
> ---
>
> [1] S Zeng et al. (2021) A Decentralized Policy Gradient Approach to Multi-Task Reinforcement Learning.
>
> [2] A Anwar and A Raychowdhury (2021). Multi-task federated reinforcement learning with adversaries.
>
> [3] S Cen et al. (2023). Fast Global Convergence of Natural Policy Gradient Methods with Entropy Regularization.

---

> > ### Author Response · Authors · 2024-08-10
> >
> > Dear Reviewer vHsi,
> >
> > We've taken your initial feedback into careful consideration in our response. Could you kindly confirm whether our responses have appropriately addressed your concerns?
> >
> > If you find that we have properly addressed your concerns, could you please kindly consider increasing your initial score accordingly? Please let us know if you have further comments.
> >
> > Thank you for your time and effort in reviewing our work!
> >
> > Many thanks, Authors

---

> > > ### Comment · Reviewer_vHsi · 2024-08-11
> > > **Rebuttal Feedback**
> > >
> > > Thank the authors for the response. I have raised my score.
> > >
> > > W1: I could not find the word "cooperative" in the paper. Please make it explicit in the paper.
> > >
> > > W2: The authors address my concerns.
> > >
> > > W3: Out of curiosity, do the authors expect the $O(1/\epsilon^{-2})$ sample complexity to be achievable for federated Q-learning and NAC?

---

> > > > ### Author Response · Authors · 2024-08-13
> > > >
> > > > Dear Reviewer vHsi,
> > > >
> > > > Thank you very much for raising the score! For W1, we'll make it clear that we consider the cooperative setting in our updated paper. Thank you for your suggestion. For W3, we currently don't know if $O(\epsilon^{-2})$ sample complexity is achievable in the fully decentralized multi-task setting we consider and will leave exploring it as future work. Thank you for your interests in this!
> > > >
> > > > Many thanks, Authors

---

### Official Review · Reviewer_LgxC · 2024-07-10

**Soundness:** 3
**Presentation:** 3
**Contribution:** 2
**Rating:** 6
**Confidence:** 3

**Summary:**

This paper studied federated multi-task reinforcement learning (RL), where multiple learning agents interact with different RL problems (different rewards) and communicate through an arbitrary communication graph. This paper proposed a federated natural policy gradient algorithm and a federated natural actor-critic algorithm. Both methods are proved to converge to a global optimum with favorable convergence rate.

**Strengths:**

1. The paper is technically sound.
2. The studied problem is well-motivated
4. The results are solid and provide insightful relationship between the number of tasks and convergence rate.

**Weaknesses:**

1. The learning objective function is less general. When the rewards are different, a better learning goal is to find local optimal policies for different tasks (agents).
2. The technical novelty is less clear. For example, compared to the single task counterpart (Cen et al, 2021), is there any technical difficulties?

Cen et al., Fast Global Convergence of Natural Policy Gradient Methods with Entropy Regularization, 2021

**Questions:**

Please see the weaknesses part.

**Limitations:**

No further limitations to be addressed.

---

> ### Author Rebuttal · Authors · 2024-08-07
>
> ## Response to Reviewer LgxC
>
> Thank you for your review and positive comments. Below we address you questions point-by-point. If our responses resolve your concerns, we'd appreciate your consideration of increasing your current score. Certainly, please also let us know if you have further questions.
>
> >**regarding learning objective**
>
> While we appreciate the perspective that finding local optimal policies for different tasks could be an alternative objective, we believe the problem of finding a single global policy that maximizes the average performance across all tasks is also valuable and necessary in many scenarios, especially when the goal is to collectively maximize the global performance (e.g., for fairness). Please refer to Appendix B for details. We also provide a few examples here:
>
> - The GridWorld experiment in our paper (Appendix H), where agents collectively learn to navigate a global map with partial information. This problem could be seen as a simplified version of the Unmanned Aerial Vehicle (UAV) Patrol Mission, each UAV patrols only in a specific area, and they need to collectively train a strategy utilizing information from the entire patrol range.
>
> - Multi-task robot control, where a single robot needs to perform well across various tasks.
>
>
> Note that our objective is well-established in the federated/decentralized RL literature, see [1-3] for example.
>
> >**regarding technical novelty**
>
> We have a detailed discussion on our technical novelty in Appendix B.2. We also summarize our key points here to give you a better understanding of the technique difficulties we conquer:
>
> One key difficulty is to estimate the global Q-functions using only neighboring information and local data. To address this issue, we invoke the "Q-tracking" step (see Algorithm 1,2), which is inspired by the gradient tracking method in decentralized optimization. Note that this generalization is highly non-trivial: to the best of our knowledge, the utility of gradient tracking has not been exploited in policy optimization, and the intrinsic nonconcavity issue, together with the use of natural gradients, prevents us from directly using the results from decentralized optimization. It is thus of great value to study if the combination of NPG and gradient tracking could lead to fast globally convergent algorithms as in the standard decentralized optimization literature despite the nonconcavity.
>
> Besides, due to the lack of global information sharing, care needs to be taken to judiciously balance the use of neighboring information (to facilitate consensus) and local data (to facilitate learning) when updating the policy. Compared to the centralized version of our proposed algorithms, a much more delicate theoretical analysis is required to prove our convergence results. For example, the key step to establish the convergence rate of the single-agent exact entropy-regularized NPG is to form the 2nd-order linear system in Eq.(47) in Cen et al., 2021, while in our corresponding analysis, a 4th-order linear system in Lemma 1 is needed, where the inequality in each line is non-trivial and requires the introduction of some intricate and novel auxiliary lemmas, see appendix D.
>
> ---
>
> [1] S Zeng et al. (2021) A Decentralized Policy Gradient Approach to Multi-Task Reinforcement Learning.
>
> [2] K Zhang et al. (2018) Fully Decentralized Multi-Agent Reinforcement Learning with Networked Agents.
>
> [3] T Chen et al. (2022) Communication-Efficient Policy Gradient Methods for Distributed Reinforcement Learning.

---

> ### Author Response · Authors · 2024-08-10
>
> Dear Reviewer LgxC,
>
> We've taken your initial feedback into careful consideration in our response. Could you kindly confirm whether our responses have appropriately addressed your concerns?
>
> If you find that we have properly addressed your concerns, could you please kindly consider increasing your initial score accordingly? Please let us know if you have further comments.
>
> Thank you for your time and effort in reviewing our work!
>
> Many thanks, Authors

---

> > ### Comment · Reviewer_LgxC · 2024-08-13
> >
> > Thanks for the response and addressing my concerns. The newly provided examples seems reasonable. I decide to remain my score.

---

### Official Review · Reviewer_tByy · 2024-07-12

**Soundness:** 3
**Presentation:** 3
**Contribution:** 2
**Rating:** 6
**Confidence:** 4

**Summary:**

A decentralized policy-gradient algorithm is introduced.  The setting is that multiple agents are operating in environments with identical states, actions, and dynamics but different reward functions; the goal is for the agents to collaboratively find a common policy that maximimzes the aggregate value across all tasks.  Convergence results are presented both with and without entropy regularization.

**Strengths:**

The convergence results are strong in that the dependence on $T$ is reasonable and the dependence on the sizes of the state and action spaces is weak.

I appreciate the high-level discussion in Appendix B and the over-arching research goal of trying to understand multi-task learning from a mathematical point of view.

**Weaknesses:**

The decision to defer a careful comparison to related work to the Supplementary Material is questionable.

Lines 100-101:
"To the best of our knowledge, the proposed federated NPG and NAC methods are the first policy
101 optimization methods for multi-task RL that achieve explicit non-asymptotic global convergence
102 guarantees in terms of iteration and sample complexities, allowing for fully decentralized communi-
103 cation without any need to share local reward/task information."

I am not sure that I agree with this statement.  Theorem 2 in [ZAD+21] seems to do exactly this.  The algorithm is different (but still fully decentralized) and the results are not quite as strong, but it does provide non-asymptotic global convergence guarantees in terms of iteration and sample complexities.


Line 215: You have $\sigma=0$ for a fully connected graph ... is "client-server" the best way to describe this?

While I really do appreciate the discussion in Appendix B.1, none of the points brought up there are addressed by the results in the paper.  Nothing in the theoretical results shows that the agents learn any faster because they are communicating then they would just learning their own tasks independently; nothing in the results speaks to the generalization of the common policy learned to a new task; there may be something to scalability here, but that would be the same for any federated learning setting.


Also, let me just nitpick the way a few of the examples relate to the paper:
It is unclear why healthcare providers or people training large ChatGPT models would be restricting themselves to communicating on a mesh network.  The problem of UAVs patrolling different areas then combining their results is probably more appropriately modeled as different state spaces with the same reward functions, rather than same state space with different rewards.  It is hard to imagine that having a common policy across different characters in a video game captures the spirit of learning to play video games; many times the whole point of having different characters is that they have different optimal control policies.

**Questions:**

None,

**Limitations:**

None.

---

> ### Author Rebuttal · Authors · 2024-08-07
>
> ## Response to Reviewer tByy
>
> Thank you for your comments. Below we answer your questions point-by-point. If these clarifications address your primary concerns, we'd appreciate your consideration of increasing your score. Certainly, please don't hesitate to request any further clarification.
>
> > **defer related work to the Supplementary Material**
>
> Thank you for your feedback. Our decision was driven by the page limitation and the extensive technical content in the main paper. We'll put the related work to the main paper if this work is accepted and an extra page is allowed.
>
> >**regarding our claim: federated NPG and NAC methods are the first policy optimization methods for multi-task RL that achieve explicit non-asymptotic global convergence guarantees in terms of iteration and sample complexity.**
>
> - We thank the reviewer for bringing this up, and apologize for missing the early literature.
>
> - We will adjust our claim to the following:
>
>    *the proposed federated NPG and NAC methods are the first natural policy gradient based methods for multi-task RL that achieve explicit non-asymptotic global convergence guarantees in terms of iteration and sample complexity.*
>
> - We want to point out that [ZAD+21] studied decentralized PG and did not provide sample complexity analysese, as they assume the advantage function is obtained by oracle. Besides, our entropy-regularized FedNPG has last-iterate convergece guarantee while their algorithm doesn't, even though they also use entropy regularization, and the iteration complexity of ours is better than [ZAD+21] as we leverage NPG-based updates. We'll add a brief comparison with [ZAD+21] in our paper to highlight the distinctions of our results.
>
> > **You have $\sigma$ for a fully connected graph ... is "client-server" the best way to describe this?**
>
> - We appreciate your attention to this detail and will add the following clarification in our updated paper:
>
> *In our paper, we use the term "server-client" to refer to the centralized learning setting where there exists a central server that aggregates information from all agents (clients) and the distributes the aggregated information back to all agents. The mixing matrix $W$ of the server-client setting is $W=1/N \boldsymbol{1}_N\boldsymbol{1}_N^\top$ and by Definition 3.2, the spectral radius $\sigma$ is 0 for this case.*
>
> - The use of "server-client" to describe this centralized architecture is common in the decentralized learning literature ([1-3]).
>
> >**points in Appendix B.1 not addressed**
>
> We want to clarify that our goal is to learn an optimal policy that maximizes the total values over all the agents, rather than hoping to maximize the individual value functions at each agent.
>
> In line 575, by "accelerating learning and improving performance by leveraging experiences gained from one task to another," we mean that our federated approach allows agents to collectively learn a policy that performs well across all tasks, which would not be possible if agents learned independently.
>
> > **concerns regarding the examples**
>
> We acknowledge that real-world scenarios are often more complex. Our model represents a meaningful simplification to enable rigorous mathematical analysis, while still capturing essential features of federated multi-task RL problems. While current practice of some applications we envisioned may not be completely aligned with the setting of our federated RL framework, we hope that our algorithm designs might inspire their future adoption into practice.
>
> ---
>
> [1] J Ma et al. (2021). Adaptive distillation for decentralized learning from heterogeneous clients.
>
> [2] R Gu et al. (2021). From server-based to client-based machine learning: A comprehensive survey.
>
> [3] H Kasyap et al. (2021). Privacy-preserving decentralized learning framework for healthcare system.

---

> > ### Author Response · Authors · 2024-08-10
> >
> > Dear Reviewer tByy,
> >
> > We've taken your initial feedback into careful consideration in our response. Could you kindly confirm whether our responses have appropriately addressed your concerns?
> >
> > If you find that we have properly addressed your concerns, could you please kindly consider increasing your initial score accordingly? Please let us know if you have further comments.
> >
> > Thank you for your time and effort in reviewing our work!
> >
> > Many thanks, Authors

---

> > > ### Comment · Reviewer_tByy · 2024-08-13
> > >
> > > Yes, thanks.  The response does address at least some of my concerns.  I will move the score up.

---

### Official Review · Reviewer_bXwg · 2024-07-15

**Soundness:** 3
**Presentation:** 2
**Contribution:** 3
**Rating:** 6
**Confidence:** 3

**Summary:**

The paper studies the federated multi-task reinforcement learning in a decentralized setting. In the problem, the agents share the same transition kernel but have different reward functions. The communications of agents are defined on a prescribed graph topology. The authors first consider tabular setting and develop a federated natural policy gradient (FedNPG) to solve the problem and prove the sublinear convergence. Besides, the authors extend FedNPG to function approximation setting and propose a federated natural actor critic with its theoretical convergence guarantee. The proposed algorithms are evaluated on a $K\times K$ GridWorld problem.

**Strengths:**

The strengths of the paper are summarized as below.
+ The paper considers federated reinforcement learning in a decentralized setting, which is challenging and closely related to the concerns of the community.
+ The paper develops algorithms for different cases with tabular settings and function approximation settings.
+ The paper gives the first rigorous analysis on the convergence rate of the FedNPG and FedNAC, which illustrates some insights including the impact of the spectral radius $\sigma$ on the performance.

**Weaknesses:**

The proposed algorithm aggregates the neighborhood information by a mixing matrix $W$. However, it is unclear how to obtain the matrix $W$ in the practical decentralized setting. It would be better to have more explanations on this.

**Questions:**

In the considered setting, the agents learn their own policies in an online decentralized manner while the convergence analysis is about the aggregated policy $\bar{\pi}^t$.  The calculation of $\bar{\pi}^t$ needs the results of all $N$ policies. This seems to be contradictory with the decentralized setting.

Can the authors give some examples about the mixing matrix $W$ and show how to obtain the matrix $W$?

I am happy to change the score according to the answers.

**Limitations:**

The authors need to more clearly address the limitations on the required information of the proposed algorithms.

---

> ### Author Rebuttal · Authors · 2024-08-07
>
> ## Response to Reviewer bXwg
>
> Thank you for your feedback. Below We address your concerns. If our responses resolve your questions, we'd appreciate your consideration in raising the score. For any remaining issues, please don't hesitate to let us know.
>
> > **(Weaknesses) how to obtain the mixing matrix $W$ in practice**
>
> There are several standard designs for $W$, such as uniform weights (where each agent assigns equal weight to itself and all its neighbors), Metropolis-Hasting weights (based on the degree of each node) [1], and Laplacian-based weights [2], with the aim that the designed $W$ agrees with the graph topology while mixing fast. Some papers also propose optimization methods to find $W$, see [3] for example.
>
> > **examples of the mixing matrix $W$**
>
> We are happy to provide the following examples:
>
> - In the server-client setting, the mixing matrix is a rescaled all-one matrix.
>
>
> - In our experiment we assume the commucation network is a standard ring graph whose corresponding mixing matrix is given in Eq.(304) in Appendix H.
>
> - In the star network [2], one central agent communicates with all others and peripheral agents only communicate with the center. It's corresponding mixing matrix is given by
> [[1/4  1/4  1/4  1/4],
> [1/2   1/2  0  0],
> [1/2   0  1/2  0],
> [1/2   0  0  1/2]]
> where we assume $N=4$.
>
> - In random network [1], the connections of the nodes are random.
>
> > **regarding the aggregated policy $\bar \pi^{(t)}$**
>
> We want to clarify that in the formal version of all our theorems, we provide consensus error bounds which demonstrate the local policies converge to the aggregated policy $\bar \pi^{(t)}$, i.e., there is no need to construct the aggregated policy $\bar \pi^{(t)}$ in practice.  Please see (47) (exact entropy-regularized FedNPG), (65) (inexact entropy-regularized FedNPG), (72) (exact vanilla FedNPG), (93) (inexact vanilla FedNPG) and (129) (FedNAC). They show that analyzing the convergence of $\bar \pi^{(t)}$ is meaningful and directly relevant to the behavior of the local policies in the decentralized setting.
>
> > **The authors need to more clearly address the limitations on the required information of the proposed algorithms.**
>
> We speculate that the reviewer is inquiring about the specific information each agent needs to know and share for the algorithms to function properly. Please let us know if our interpretation is incorrect. To address this, we will clarify in the paper that each agent $i$ only needs to know its local reward function $r_i$, its neighbors and the corresponding weights $w_{ij}$. No global information or centralized computation is required during the algorithm execution, and agents only share local policy and Q-function estimates with neighbors.
>
> ---
>
> Reference:
>
> [1] Y Dandi et al. (2022). Data-heterogeneity-aware mixing for decentralized learning.
>
> [2] CC Chiu et al. (2023). Laplacian matrix sampling for communication-efficient decentralized learning.
>
> [3] L Xiao, S Boyd (2004). Fast linear iterations for distributed averaging.
>
> [4] Qu G, Li N (2017). Harnessing smoothness to accelerate distributed optimization[J].
>
> [5] Zhao et al (2022). BEER: Fast O(1/T) Rate for Decentralized Nonconvex Optimization with Communication Compression.

---

> > ### Author Response · Authors · 2024-08-10
> >
> > Dear Reviewer bXwg,
> >
> > We've taken your initial feedback into careful consideration in our response. Could you kindly confirm whether our responses have appropriately addressed your concerns?
> >
> > If you find that we have properly addressed your concerns, could you please kindly consider increasing your initial score accordingly? Please let us know if you have further comments.
> >
> > Thank you for your time and effort in reviewing our work!
> >
> > Many thanks, Authors

---

> > > ### Comment · Reviewer_bXwg · 2024-08-10
> > >
> > > Thank you for your responses which address my concerns.
> > >
> > > I believe that the convergence of the local policies to the aggregated policy is important to understand the results, so please consider to move the consensus error bounds to the main paper.
> > >
> > > I'm happy to increase the score.

---

> > > > ### Author Response · Authors · 2024-08-11
> > > >
> > > > Dear Reviewer bXwg,
> > > >
> > > > Thank you very much for raising the score! We'll move the consensus error bounds to the main paper in our updated version.

---

### Decision · Program_Chairs · 2024-09-25

**Decision:**

Accept (poster)

**Comment:**

This paper develops federated reinforcement learning algorithms for the setting where clients’ transition kernels are the same but their reward functions differ. The authors consider both the tabular and function approximation settings and derive (last iterate / consensus) convergence rates for natural policy gradient and actor critic approaches under general client connectivity conditions. In general, the reviewers found the analysis rigorous and comprehensive. Areas for improvement mostly lie in the motivations given for the problem setting (discussed in Appendix B.1) and the mismatch with developed theory:
- Optimizing for average reward: ideally, in a hospital setting, one would want a bespoke treatment for each patient, not a one-size-fits-all treatment. Reviewer vHsi gives an illustrative example of the impact of optimizing average reward on individuals in their weakness (1).
- Acceleration / knowledge transfer: the convergence rates do not reveal a dependence on the number of clients / tasks that implies speedup as is suggested by the authors’ references to  [KSJM22, WJC23] in the paragraph on *Distributed and federated RL*.

As a suggestion, the authors might better motivate the setting by assuming each client samples a reward function from the same reward distribution. One would like to learn a policy that is optimal w.r.t. the expected reward function, but a single client / agent only observes a single sampled reward function. In this setting, federated learning allows many clients to leverage each other's sampled reward functions to indirectly learn a policy that is individually optimal. Overall, the reviewers are generally positive about the contribution of the work, and so we recommend accepting this paper.